EMBO
Molecular Medicine

# Tenascin-C orchestrates radiotherapy-induced head and neck tumor regression

Thomas Loustau [1,2,3,17]✉, Ioanna Mitrentsi [1,2,17], Nuohan Wang[1,2], Caroline Spenlé[1,2,4], Alexia Pavlidaki[1,2], Thibaud Tranchant[1,2], Gilles Riegel[1,2], Akhil Venu[1,2], Rime Oueidat[1,2], Manuel Koch [5], Marion Dumas[1,2], Fanny Wack[1,2], Aurelie Hirschler[6], Christine Carapito[6], Nicodème Paul [7], Raphael Carapito [7], Matthias Mörgelin[8], Uwe Hansen[9], Joyce Azzi[10,11], Lucie Aubergeon[12], Nathalie Salomé [1,2], Sayda Dhaouadi[13], Pierre Grenot [14], Balkiss Bouhaouala-Zahar [13], Simona La Cioppa[15], Philippe Oertle[15], Valerio Izzi [16], Marija Plodinec[15], Georges Noel [10,11], Hélène Burckel[10,11] & Gertraud Orend [1,2]✉

## Abstract

Given that head and neck squamous cell carcinoma (HNSCC) patients have poor survival outcomes, a better understanding of the therapeutic benefits of ionizing irradiation (IR), the major treatment modality besides surgery, is needed. A confounding factor is the immunosuppressive tumor microenvironment determined by tenascin-C (TNC), a highly abundant extracellular matrix molecule upregulated by IR. We investigated the roles of TNC on radio-induced tumor regression in a murine oral HNSCC model expressing or lacking TNC. While tumors in a TNC-expressing host were radiosensitive, they were radioresistant in TNC genetically depleted mice. We identified fibroblast reticular cells (FRCs) as critical regulators. TNC plays a compartmentalized and dual role in regulating tumor radiosensitivity with a detrimental role in the tumor stroma opposed to an essential role in the tumor-draining lymph nodes. This is relevant as a high FRC signature and high TNC levels together correlate with shorter HNSCC patient survival. TNC-expressing FRCs may be an excellent novel target to improve radiotherapy-induced tumor eradication, as our TNC targeting MAREMO peptide reduced tumor cell numbers and plasticity upon IR.

**Keywords** Tenascin-C; Radiotherapy; Fibroblast Reticular Cells; Tumor Draining Lymph Nodes; MAREMO Peptide
**Subject Category** Cancer

## Introduction

Head and neck squamous cell carcinoma (HNSCC) is an aggressive malignancy with poor patient survival despite advances in therapy (reviewed in Chinn and Myers, 2015). Ionizing radiation (IR) remains a key treatment strategy, effectively inducing tumor and stromal cell death and activating anti-tumor immunity through the tumor-draining lymph nodes (TdLNs), which serve as critical sites for T-cell priming and expansion (Buchwald et al, 2020; Koukourakis and Giatromanolaki, 2022). However, IR paradoxically fosters an immunosuppressive tumor microenvironment (IS-TME), which limits its therapeutic efficacy (Menon et al, 2019; Guo et al, 2023). A central component of the IS-TME is tenascin-C (TNC), an extracellular matrix (ECM) glycoprotein implicated in immune modulation and tumor progression across various cancers (Yilmaz et al, 2022). In HNSCC, fibroblast reticular cells (FRCs) are a major source of TNC (Spenlé et al, 2020). TNC promotes immune evasion by inducing tolerogenic molecules, skewing macrophages toward an M2 phenotype, enhancing regulatory T cell (Treg) infiltration, and dampening CD8 + T cell responses (Deligne et al, 2020; Spenlé et al, 2020; Jachetti et al, 2015). Moreover, TNC induces the expression of ECM molecules that form immunosuppressive niches, so-called tumor matrix tracks (TMT) (Spenlé et al,

[1]INSERM UMR_S 1109, The Tumor Microenvironment Laboratory, Hôpital Civil, Institut d'Hématologie et d'Immunologie, Strasbourg, France. [2]University of Strasbourg, Strasbourg, France. [3]University of Strasbourg, UPR CNRS 9002, ARN, IUT Louis Pasteur, 1 allée d'Athènes, Schiltigheim, France. [4]University of Strasbourg, ESBS, INSERM-ERL1321, groupe Biothérapie peptidique, Pôle Api, Bâtiment D, 300 Boulevard Sébastien Brant, Illkirch, France. [5]University Cologne, Faculty of Medicine and University Hospital Cologne, Center for Dental, Oral and Maxillofacial Medicine (central facilities), Cologne, Germany. [6]Laboratoire de Spectrométrie de Masse BioOrganique, Institut Pluridisciplinaire Hubert Curien (IPHC), UMR 7178, Université de Strasbourg, CNRS, Infrastructure Nationale de Protéomique ProFi - UAR2048, Strasbourg, France. [7]Laboratoire d'ImmunoRhumatologie Moléculaire, INSERM UMR_S 1109, Plateforme GENOMAX, ITI Médecine de Précision de Strasbourg, Transplantex NG, Faculté de Médecine, Fédération Hospitalo-Universitaire OMICARE, Strasbourg, France. [8]Colzyx AB, Lund, Sweden. [9]Institute for Musculoskeletal Medicine (IMM), University Hospital Muenster, Muenster, Germany. [10]Institut de Cancérologie Strasbourg Europe (ICANS), UNICANCER, Radiobiology Laboratory, Paul Strauss Comprehensive, Cancer Center, Strasbourg, France. [11]ICube, UMR7357, Equipe Imagerie Multimodale Intégrative en Santé, Université de Strasbourg, Strasbourg, France. [12]University of Strasbourg, UPR CNRS 3572, I2CT, 2 allée Konrad Roentgen, Strasbourg, France. [13]Pasteur Institute of Tunis, NanoBioMedika research Team, 13 Place Pasteur - B.P. 74 - 1002, Tunis, Tunisia. [14]Laboratoire d'ImmunoRhumatologie Moléculaire, INSERM UMR_S 1109, Plateforme CYTOMAX, Institut d'Hématologie et d'Immunologie, Strasbourg, France. [15]Artidis, Hochbergerstrasse 60c, Basel, Switzerland. [16]Faculty of Biochemistry and Molecular Medicine at University of Oulu, Oulu, Finland. [17]These authors contributed equally as first authors: Thomas Loustau, Ioanna Mitrentsi. ✉E-mail: thomas.loustau@unistra.fr; gertraud.orend@inserm.fr

2020; Murdamoothoo et al, 2021; Fonta et al, 2023). TMT might phenocopy immune-regulating reticular matrix networks or so-called conduits in TdLNs as well as other lymphoid tissues, where TNC is an integral component (Spenlé et al, 2015; Huang et al, 2018; Drumea-Mirancea et al, 2006; Panocha et al, 2025). By sequestering soluble factors that attract immune cells, TNC creates an adhesive substratum that retains dendritic cells (DCs), macrophages (M-phages), and CD8 + T cells in the tumor stroma. This physical restriction prevents these immune cells from interacting with the tumor cells, thereby inhibiting both innate and adaptive immune responses (Spenlé et al, 2020; Deligne et al, 2020; Murdamoothoo et al, 2021). As IR induces TNC in human and mouse models (Spenlé et al, 2021; Omori et al, 2024; Toyomasu et al, 2022) we aimed to determine whether TNC plays a role in IR-induced tumor regression. To address this possibility, we applied the previously used carcinogen 4-NQO (4-Nitroquinoline 1-oxide)-induced tongue oral squamous cell carcinoma (OSCC) model, recapitulating the genetic alterations of human HNSCC (Wang et al, 2019) in mice, that express TNC (wild-type, WT) or are genetically depleted of TNC (TNC knockout, TNCKO). We investigated these tumors and the TdLNs at the cellular and molecular levels, including the immune cell infiltrate and FRCs, and observed that tumors expressing TNC were radiosensitive in contrast to TNC-depleted tumors, which were radioresistant. We discovered that TNC exerts opposing roles, promoting tumor regrowth after irradiation, while supporting immune functions within the TdLNs, where TNC maintains immune-supportive functions. Our findings indicate that the balance between these opposing roles critically shapes radiotherapy outcomes. We identified both FRCs and TNC as promising targets to enhance radiotherapeutic efficacy. High intratumoral levels of FRCs with strong TNC expression are associated with poorer survival in irradiated HNSCC patients. Moreover, therapeutic inhibition of TNC using the MAREMO peptide enhances tumor radiosensitivity, highlighting its potential clinical relevance.

## Results

### Expression of TNC impacts tumor radiosensitivity, and immune cell infiltration in the TdLNs of a murine model of OSCC

To investigate the effect of IR on tumors, we used tumor-bearing mice with 4NQO-induced tongue OSCC in both WT and TNCKO hosts. The tumors were then locally irradiated with a single dose of 2 Gray (Gy), a dose that represents the standard daily fraction used in conventional clinical fractionation protocols for head and neck cancer ($5 \times 2$ Gy over 7 weeks), thereby enabling direct translational comparison to human regimens (Combs et al, 2005). We selected single-dose rather than fractionated irradiation to minimize confounding variables inherent to multi-fraction scheduling while maintaining a clinically relevant dose per fraction. This single-dose approach provided a controlled and reproducible perturbation of the tumor–stroma–immune ecosystem that induced tumor regression while preserving sufficient tissue for comprehensive downstream analysis. We collected the tumors and TdLNs 6 weeks later for further investigation in order to determine their longer-term consequences (Fig. EV1A). As previously reported, WT mice

developed an average of two tumors per tongue, in contrast to TNCKO mice, which showed an average of one tumor per tongue (Fig. EV1B; Appendix Fig. S1A (Spenlé et al, 2020)). While the number and size of the tumors regressed upon 2 Gy IR in WT mice expressing TNC, this was not observed in TNCKO mice, indicating that tumors are radiosensitive in the presence of TNC, but not in its absence (Figs. 1A and EV1B). Whereas most IR-treated tumors in TNCKO mice were in situ carcinomas, TNC-expressing tumors presented a lower proportion of in situ carcinomas, accompanied by a higher prevalence of invasive carcinomas and in situ carcinomas expressing keratin. This suggests a tumor progression-promoting effect of IR, however, only in TNC-expressing tumors (Appendix Fig. S1B). As increased stiffness counteracts IR tumor remission (Mottareale et al, 2024), we applied atomic force microscopy (AFM) measurements and noticed that the TNC-depleted tumors showed no difference in stiffness in comparison to the WT tumors (Appendix Fig. S1C, Batasheva et al, 2024). By staining for Ki67 and subsequent signal quantification, we demonstrate a reduced proliferation rate upon 2 Gy IR in agreement with a reduced number and size of tumors in a TNC expression context (Fig. 1B,C). However, the already lowered proliferation in tumors depleted of TNC showed no further decrease upon irradiation (Figs. 1A,C and EV1B). By TNC staining, we confirmed higher TNC expression in the WT tumors and proved its absence in the TNCKO tumors (Fig. 1B). As TNC orchestrates an immune suppressive TME in this model (Spenlé et al, 2020), we used the leukocyte marker CD45 and the myeloid/dendritic (DC) cell marker CD11c, imaged immune cells using flow cytometry (all cells) or tissue staining (tumor islet cells) (Figs. 1D–F and EV1C; Appendix Fig. S1D–O). Whereas the number of leukocytes and DCs increased in the TNCKO tumors, their numbers were lower in WT tumors. Moreover, while these cells further decreased inside the tumor islets in the irradiated WT tumors, they did not in the irradiated TNCKO tumors (Fig. 1D,E; Appendix Fig. S1E, Spenlé et al, 2020). As TNC impacted CCR7+ expressing DCs (Spenlé et al, 2020), we used flow cytometry to analyze CCR7+ cells and showed TNC-dependent differences upon IR, in a similar direction. Whereas CCR7+ cells comprising DCs, macrophages, CD4 + T cells, Tregs, and CD8 + T cells were highly abundant in TNC-expressing IR tumors, they were much less present in the TNC-depleted IR tumors (Fig. 1E; Appendix Fig. S1D–I).

By flow cytometry, we observed that FRCs are more numerous upon IR in the TNC-expressing tumors, whereas TNC-depleted tumors presented much lower FRC levels (Figs. 1G and EV1D). In order to understand whether the FRC abundance influences the expression of CCL21, one of the ligands of CCR7 (Nagira et al, 1997), we applied ELISA. CCL21 expression was significantly higher in the WT tumors after IR, which was not the case in the TNCKO context, where CCL21 levels remained at already reduced levels (Fig. 1H). As there were more DCs in the TNCKO tumors and tumor islets, potentially acting as antigen-presenting cells (APCs), we investigated whether DCs were also more abundant in the TdLNs of these mice. Indeed, DCs were more numerous in these lymph nodes, although again their numbers remained unaffected by IR (Fig. EV1E). While CCR7+ and activated DCs (CD86 + /CD80 + ) were more abundant, immune suppressive Tregs were less numerous in the TdLNs and tumors of mice lacking TNC, presumably reflecting a deregulated adaptive immunity

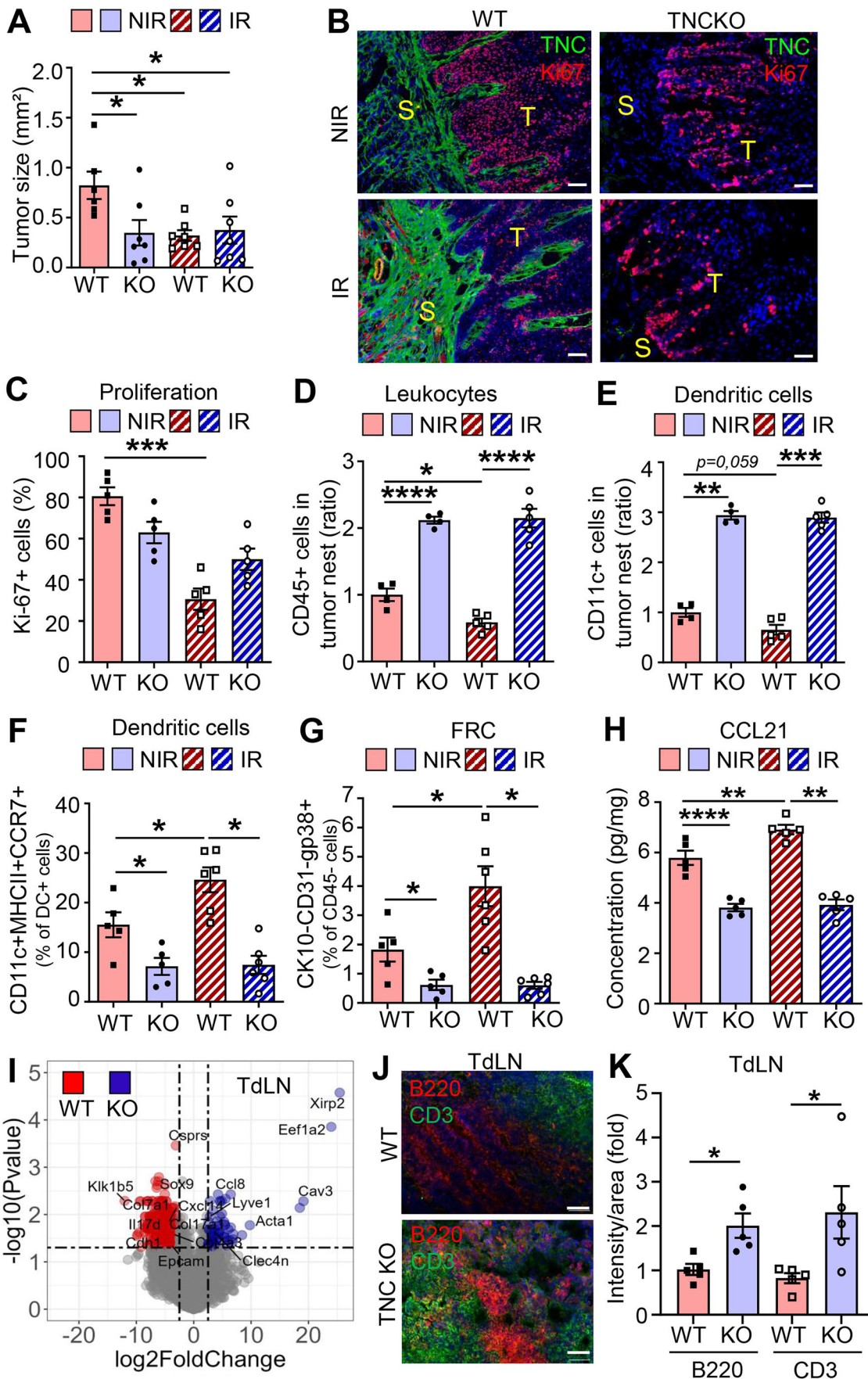

**Figure 1. Expression of TNC impacts tumor radiosensitivity, and immune cell infiltration in the TdLNs of a murine model of OSCC.**

(A) Quantification of the tongue tumor size in 4NQO-treated WT and TNCKO (KO) mice non-irradiated (NIR) or irradiated (IR), $n = 5$ per group. (B) Representative IF images for TNC and the proliferation marker Ki67 in NIR and IR 4NQO tumors of WT and TNCKO mice (T, tumor islet, S, Stroma). Scale bar, 200 μm. (C) Quantification of Ki67-positive cells (%) in the tumor per image. Four images per tumor, $n = 5$ mice per group. (D, E) Quantification of the immunostaining signal to evaluate the spatial distribution of the CD45+ (D) and CD11c+ (E) cells present in the tumor nest as a ratio of positive cells over the total of cells per image. (F, G) FACS analysis of CCR7+ dendritic cells (DC) expressed as a ratio (%) of the total DC population (CD45+, CD11c+, MHCII+) (F) and FRCs (GP38+, CK10-, CD31−) expressed as a ratio (%) of the total CD45-negative cells (G). (H) ELISA for CCL21 in NIR and IR 4NQO tumors of WT and TNCKO mice. $N = 5$–6 per group. (I) Volcano plot of deregulated genes, after RNA sequencing of the TdLN of WT and TNC KO mice. $N = 3$ WT and 3 TNCKO mice. (J, K) Representative IF images for CD3 and B220 in the TdLN of WT or TNC KO mice (J) and quantification of their intensity/area in $n = 2$–4 sections of $N = 5$ TdLNs per group (K). Scale bar, 50 μm. Mean ± SEM; Kruskal–Wallis test and Dunn post-test, *$P < 0.05$, **$P < 0.01$, ***$P < 0.005$. The exact $P$ values are listed in Appendix Table S5. Source data are available online for this figure.

(Fig. EV1F–H; Appendix Fig. S1E–O). The levels of CCL21, needed not only as a chemoattractant but also for cell activation (Marsland et al, 2005), were similar in the TdLNs of both genotypes and remained unchanged upon IR in the WT and TNC-depleted conditions (Appendix Fig. S1P). This altogether suggests that TNC might play a role in sustaining adaptive immunity responses against the tumor cells, in particular in response to IR, a function that seems to be impaired in the TdLNs of TNCKO mice. This is supported by a higher number of immature CCR7+ cells comprising CD4 + T and CD8 + T cells in the tumors of the TNC-depleted mice (Appendix Fig. S1M,N).

Next, we determined gene expression differences in the TdLNs between genotypes using RNA seq analysis with a cutoff of log2 fold change of 2.5 and an adjusted $P$ value of 0.05, and noticed profound differences in 493 genes to be downregulated and 62 genes to be upregulated in WT in comparison to the TNCKO TdLNs. Gene Set Enrichment Analysis (GSEA) demonstrated that the most deregulated categories showed downregulation of processes linked to the structural organization of the reticular networks (epithelial tube formation, cell–cell junctions, namely *Cdh1*, *Epcam*) in the TNC-depleted TdLNs, while processes involved in adaptive immunity (immunoglobulin production, B cell mediated immunity) were the most upregulated categories in the same TdLNs (Figs. 1I and EV1I; Appendix Table S1). Moreover, markers for lymphatic endothelial cells (LEC) such as Lyve 1 and Acta 1 (for cytoskeletal reorganization) were higher in the TNCKO than WT TdLNs, indicating an altered cellular composition and function in the TdLNs in the absence of TNC (Appendix Table S1). As expected, TNC was not expressed in the TdLNs of the TNCKO mice, whereas TNC overlaps with glycoprotein 38 (GP38) staining, supporting that FRCs express TNC in the WT TdLNs (Appendix Fig. S1Q). Staining for CD3 and B220 and subsequent quantification showed an excessive number of positive cells in the TNCKO TdLNs, suggesting a deregulated and impaired immunity (Fig. 1J,K; Appendix S1U). Gene expression was largely unaffected by 2 Gy IR in both WT and TNCKO TdLNs and showed similar patterns in direct comparisons of irradiated TdLNs from the two genotypes. However, the profound difference in the non-irradiated TdLNs between genotypes is lost upon IR. This suggests a major role of TNC in regulating the TdLN function and little or no direct effect of 2 Gy IR on gene expression (Appendix Fig. S1R–T). A heatmap representation reveals a significantly deregulated expression of around 200 genes comprising GO terms related to inflammatory and adaptive immune responses, including *Ccl28*, *Cx3cl1*, and *Il34* downregulated and, *Ifng*, *Granzyme B*, *Ifnar*, *Il17*, *Ctla4*, and *Cd4*, as well as several *Igh* and *Igk* genes upregulated in the TdLNs from TNCKO mice (Appendix S1V,W; Table S1).

Altogether, these results demonstrate that TNC plays a profound role in OSCC radiosensitivity, the IR-induced stroma responses, the immune cell infiltration, and TdLN functionality. Importantly, a deregulated and likely impaired TdLN immunity in the TNC-depleted mice may contribute to the radioresistance of the tumors in these mice. In the tumors, we discovered an increased abundance of FRCs upon IR that play an instrumental role in enforcing an IS-TME.

## TNC regulates the FRC phenotype

To further explore the impact of TNC on the FRC phenotype, FRCs were isolated from the lymph nodes of two naïve 10-week-old TNCKO mice and were investigated as previously done for FRCs isolated from two age-matched naïve WT mice (Spenlé et al, 2020). While FRCs expressing TNC had an elongated fibroblastic cell shape with parallel aligned actin stress fibers, FRCs depleted of TNC were less well spread with poorly formed actin stress fibers, thus appearing smaller (Figs. 2A and EV2A). To better understand the FRC phenotypes we conducted RNA sequencing (RNA seq) (Appendix Table S2) and mass spectrometry (Mass spec) analysis (Appendix Table S3), followed by IF staining and ELISA, and observed profound TNC-dependent differences (however not between cell isolates) (Figs. 2B and EV2B). Expression of well-established lymph node FRC markers (Fletcher et al, 2015; Ferreira et al, 2021; Förster and Moschovakis, 2013), such as GP38 (podoplanin), VCAM-1, and Collagen VI (ColVI, ER-TR7 epitope), was largely reduced in TNCKO FRC isolates at both protein and mRNA levels (Fig. 2C; Appendix Table S3). As all lymph nodes were collected for FRC isolation, we wanted to know whether there are genotype-specific differences in the abundance of distinct FRC subtypes (Rodda et al, 2018). However, by comparing the abundance of subtype-specific gene expression markers, this was not the case (Appendix Fig. S2A). Expression of *Gp38*, *Vcam1*, *Col6a1*, *Ccl19*, and *Ccl21*, which are normally expressed by mature lymph node FRCs, was lower in the absence of TNC, highlighting that TNC is an important regulator of the FRC identity (Fig. 2C,D). Among the 12,456 expressed genes, 9% (1245) were upregulated, and 8% (1021) were downregulated in the two duplicates of TNCKO compared to the two duplicates of the WT FRCs (Appendix Table S2). From all 2717 proteins detected by mass spectrometry analysis, expression of 1310 proteins also showed a significant difference in the same samples. GSEA demonstrated downregulation of processes related to "fibroblast proliferation", "ECM expression and organization", "collagen production", "regulation of cell signaling", "cytokine activity/chemokine production" and "TGFβ signaling" at both mRNA and protein level (Fig. EV2D,E; Appendix Fig. S2B–E). Also, chemokine and cytokine expression was altered at the transcriptional level in the absence of TNC with *Tgfb1*, *Ccl21*,

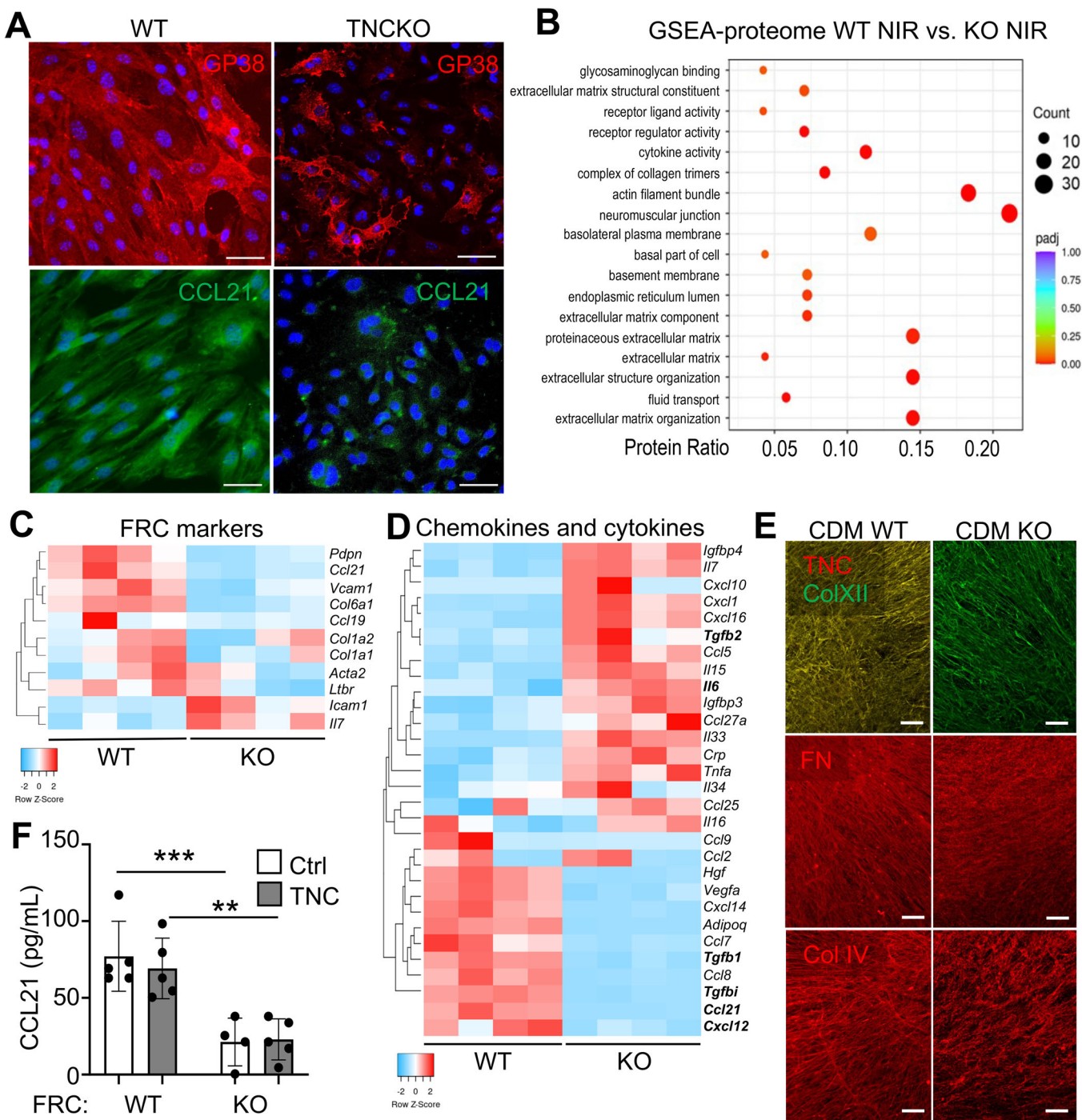

**Figure 2. TNC expression plays a pivotal role in defining the FRC phenotype.**

(A) Representative IF images for GP38 (podoplanin) and CCL21 in non-irradiated (NIR) FRCs isolated from lymph nodes of naïve WT or TNCKO mice. The same microscopic field of the image showing DAPI (blue) and GP38 (red) in the TNCKO FRCs was co-stained for the marker VCAM1 (green), which is displayed separately in Fig. EV2A. Scale bar, 50 µm. (B) Display of differentially expressed proteins in WT and TNCKO FRCs applying the GSEA analysis tool. The color code indicates the adjusted P value range and the black circles the approximate number of proteins that belong to each GO term. RNA seq gene expression analysis of WT and TNCKO FRCs represented as heatmaps for deregulated genes according to FRC markers (C) and chemokine and cytokine molecules (D). Bold text indicates genes with established roles in the respective category that are discussed in the text. (E) Representative image of Col XII, TNC, FN, and Col IV expression in the CDM of WT or TNCKO FRCs. Scale bar, 200 µm. (F) Quantification of CCL21 by ELISA in WT or TNCKO FRCs, either non-treated control (Ctrl) or treated with soluble TNC (10 µg/mL) for 24 h. $N = 4$ experiments. Mean ± SD, Kruskal–Wallis test and Dunn post-test, *$P < 0.05$, **$P < 0.01$ ***$P < 0.001$. The exact $P$ values are listed in Appendix Table S5. Source data are available online for this figure.

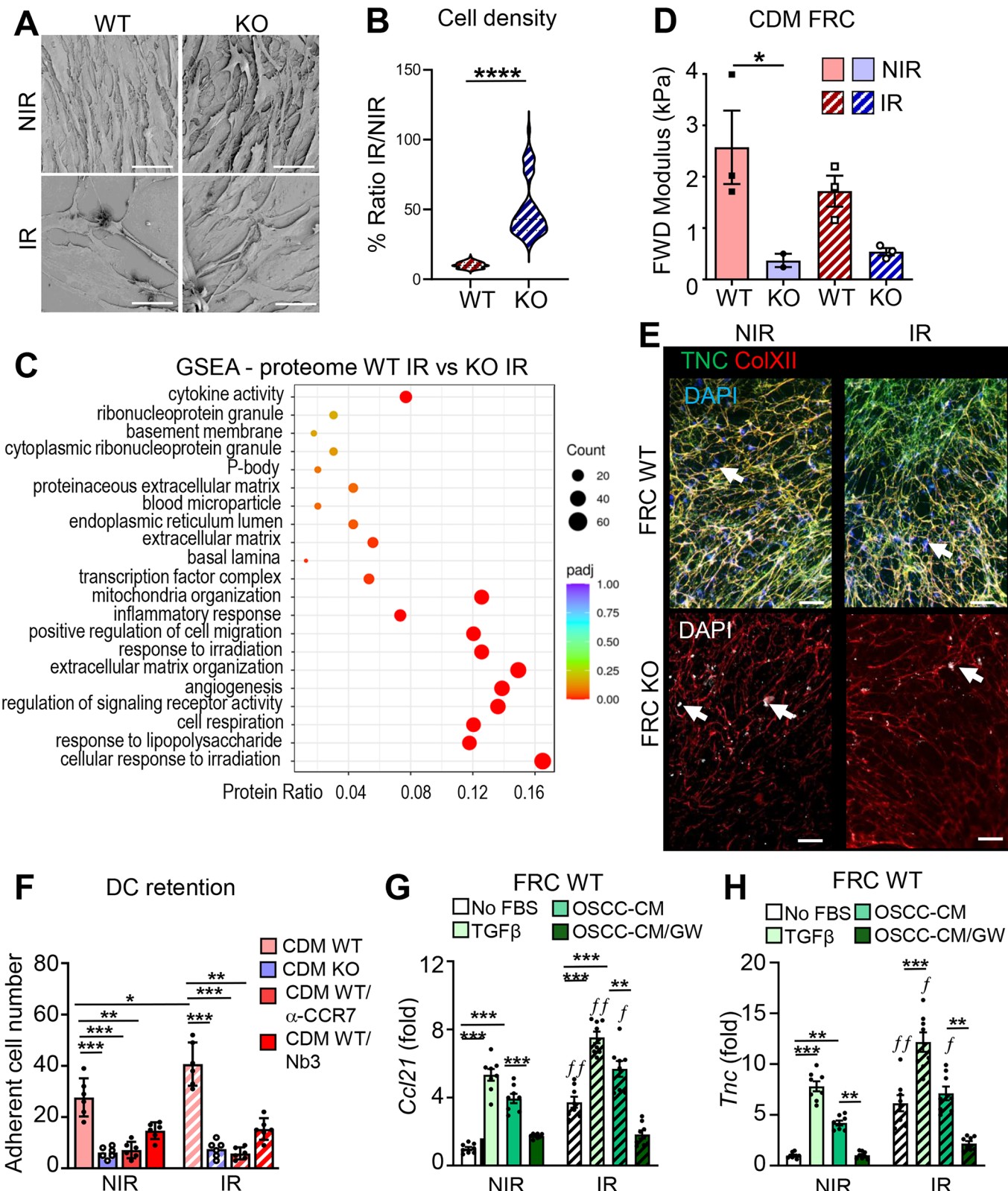

*Cxcl12* to be lower and *Il7, Cxcl10* and *Il6* higher in the TNC-depleted tumors (Figs. 2D and EV2D; Appendix Fig. S2D,H; Appendix Tables S2 and S3). Next, we investigated the cell-derived matrix (CDM) that had been deposited by the FRCs over 10 days and

observed a complete absence of TNC in the TNC-depleted FRCs, as expected, accompanied by reduced levels and less fibrillar or dense patterns of fibronectin (FN), Col IV and Collagen XII (Col XII) (Figs. 2E and EV2F). CCL21 expression was also significantly reduced

 

**Figure 3. FRC cell responses to irradiation are dependent on TNC expression.**

(A) Representative Scanning Electron Microscopy (SEM) images of irradiated (IR) or non-irradiated (NIR) WT and TNCKO FRCs. Scale bar, 50 μm. (B) Cell density (analysis of 100 SEM images for each condition) of WT and TNCKO FRCs expressed as a ratio of IR to NIR cells. Mean ± SD; Student T test, ****P < 0.0001. (C) GSEA map display of differentially expressed protein categories in WT or TNCKO FRCs after irradiation. The color code indicates the adjusted P value range, and the black circles the approximate number of proteins that belong to each GO term. (D) Forward (FWD) E-Modulus average (kPA) of the corresponding Atomic Force Microscopy measurement of the CDM of WT or TNCKO FRCs before (NIR) or after irradiation (IR). N = 3. Mean ± SEM (E, F) Representative IF images of DC2.4 cells adherent on the CDM from the FRCs (WT or KO) after the chemoretention assay. DC2.4 nuclei are stained with DAPI (arrows point at the nuclei) and were counted on the CDM in (F) upon migration toward CCL21, pre-treatment with the anti-CCR7 neutralizing antibody (αCCR7), the TNC inhibitory nanobody Nb3, or no treatment control. Scale bar, 200 μm. Mean ± SEM; n = 6 per condition; Kruskal–Wallis test and Dunn post-test, *P < 0.05, **P < 0.01, ***P < 0.005. (G, H) Expression of Ccl21 (G) and Tnc (H) was determined by qRTPCR after treatment of FRCs (WT, KO, NIR, IR) with TGFβ (10 ng/mL for 24 h) or with the CM from the OSCC13 cells in the presence or absence of GW788388 (GW). Mean ± SEM; n = 6–9 per condition; Kruskal–Wallis test and Dunn post-test, *P < 0.05, **P < 0.01. The exact P values are listed in Appendix Table S5. Source data are available online for this figure.

in the TNC-depleted FRCs (Fig. 2F; Appendix Fig. S2E). Consistent with what was previously shown in the WT FRCs, TNC addition did not upregulate CCL21 in the TNC-depleted FRCs, suggesting that TNC is not directly inducing CCL21 in either genotype cells (Fig. 2F; Appendix Fig. S2E, see below). Although lymphatic endothelial cells (LECs) are another important source of CCL21, their abundance, as determined by flow cytometry, did not differ between the two genotypes or upon IR, thus likely not contributing to the observed changes in CCL21 expression (Appendix Fig. S2F).

These results reveal that TNC has a key role in defining the FRC identity in both tumors and lymph nodes.

## Irradiation enhances an immunosuppressive phenotype in FRCs in a TNC-dependent manner

By using electron microscopy (EM), we analyzed the phenotype of the irradiated FRCs and observed differences in cell shape and density. We observed a higher cell density and number of vesicles in the irradiated WT FRCs compared to the irradiated TNCKO cells, indicating an elevated cell death upon IR in the presence of TNC (Fig. 3A,B; Appendix Fig. S3A,B). Indeed, while only 10% of the TNC-expressing FRCs survived IR, a fivefold higher population of TNC-depleted FRCs (50%) remained viable after 2 Gy IR exposure (Fig. 3B).

RNA seq and proteomic analyses also demonstrated strong differences upon IR in both genotypes and in dependence of TNC (Fig. 3C; Appendix Fig. S3C). Upon IR, 22% (2842) of all 13,025 genes were differentially expressed with 1369 (11%) genes upregulated and 1473 (11%) genes downregulated in the irradiated WT FRCs compared to the irradiated TNCKO FRCs (Appendix Table S2). At the protein level, 530 proteins were deregulated in the TNC-expressing irradiated FRCs, whereas only 126 proteins in the TNC-depleted irradiated FRCs were impacted, again reflecting the lower radiosensitivity of the TNCKO FRCs (Fig. 3C; Appendix Fig. S3C and Appendix Table S3).

The most significantly deregulated category between genotypes upon IR, was "Cellular response to irradiation" where the WT cells showed induction by IR in contrast to the TNC-depleted FRCs that showed mostly reduced expression upon IR (Figs. 3C and EV3A; Appendix Fig. S3D and Appendix Table S3). In the irradiated WT FRCs these molecular categories comprised "Regulation of cell proliferation", "Cellular response to cytokines", "Regulation of leukocyte migration" (e.g., *Cxcl12, RhoA, Rac*), and "ECM organization" (e.g., LOX, LOXL3, MMPs 2, 11, 14), altogether indicating that IR induced an activated FRC phenotype in the fewer surviving TNC-expressing cells (Appendix Fig. S3E–I). While

almost all gene ontology and GSEA proteome categories were downregulated in the irradiated TNC-depleted compared to the irradiated TNC-expressing FRCs, several proteins in the "Cellular respiration" pathway were upregulated in the TNC-depleted FRCs, potentially increasing metabolic activity contributing to their enhanced survival after IR (Appendix Fig. S3I).

To investigate potential differences in matrix stiffness in dependence of TNC and IR, we used AFM and noticed that despite an elevated expression of ECM molecules upon IR, the stiffness of the CDM deposited by the irradiated TNC-expressing FRCs was surprisingly not increased (Fig. 3D). Potentially altered crosslinking and/or degradation of the ECM networks upon IR may explain this phenotype, as e.g., LOX, TGM2, MMP2, MMP11 and MMP14 were upregulated upon IR (Appendix Fig. S3J,K). In contrast, stiffness was much reduced in the CDM from the TNC-depleted FRCs and remained unchanged upon IR, reinforcing the lack of IR responsiveness in the TNCKO FRCs (Fig. 3D). In support of an impact of IR on the matrix, EM image analysis of the tumors revealed spaces filled with loose matrix in the irradiated WT tumors (Appendix Fig. S3L).

As the proteome analysis revealed an increased expression of molecules involved in leukocyte migration and ECM organization in irradiated TNC-expressing FRCs, we wondered whether IR impacts migration and retention of DCs and performed a modified Tumor Infiltrating Leucocyte (TIL)-matrix-chemoretention assay on CDM deposited by the irradiated FRCs in the lower chamber in the context of CCl21 (Fig EV3F). First, staining of the CDM deposited by the TNCKO cells revealed a lowered Col XII expression and assembly in the absence of TNC, not only in the non-irradiated (NIR) but also in the IR condition (Fig. 3E). Although the CDM from TNC-depleted FRCs is still comprised of abundant fibrillar FN and Col IV and likely other matrix molecules, this matrix was impaired in retaining the DCs (Figs. 2E, 3F, and EV3C). In contrast, DC retention was elevated on CDM derived from the irradiated TNC-expressing FRCs and was significantly reduced with an anti-CCR7 antibody and with the TNC-specific nanobody 3 (Nb3), previously shown to block cell retention of DCs on TNC/CCL21 (Figs. 3F and EV3D, Dhaouadi et al, 2021). This suggests that IR enforces FRCs to retain DCs via a sticky TNC/CCL21 substratum. Taking into account the important role of TGFβ signaling in radio responses (Massagué, 2008) and given the increased TGFβ signaling in the TNC-expressing WT FRCs (Fig. EV2E; Appendix Table S3) and an increased expression of *Tgfb1* in these cells upon IR (Fig. EV3E), we addressed whether tumor cells can impact the FRC phenotype upon IR, potentially through TGFβ signaling. We collected conditioned medium (CM)

 

from OSCC13 cells and investigated gene expression in the irradiated FRCs by qRTPCR 48 h after exposure to either the CM or TGFβ (Fig. EV3F). This analysis revealed that both the OSCC13-derived CM and TGFβ itself induced expression of *Acta2, Col1a2, Tnc*, and *Ccl21* in the irradiated FRCs (more than in the non-irradiated counterparts), however, only when the cells were derived from a WT but not a TNCKO mouse (Fig. 3G,H; Appendix Fig. S3M–P). Most importantly, this effect was TGFβ-dependent as the TGFβRI inhibitor GW788388 (GW) impaired gene induction by IR. This result establishes an IR-stimulated crosstalk between OSCC13 cells and FRCs through TGFβ (Fig. EV3F; Appendix Table S3).

Altogether, we identified a profound effect of IR on FRC survival and phenotype. The surviving cells enforce the deposition of ECM and the expression of chemoattracting factors. TNC is crucial in defining the FRC identity, facilitating ECM deposition and promoting CCL21 expression, a process dependent on TGFβ signaling, a major pathway elicited by IR (Massagué, 2008). Moreover, TNC is essential for FRCs to establish a DC-retaining substratum, which again occurs in a TGFβ-dependent manner. Our results suggest that TGFβ signaling is largely impaired in the TNC-depleted FRCs by an unknown mechanism, which may contribute to the reduced response of FRCs to IR and subsequent tumor radioresistance.

## Irradiation enforces a TNC-dependent tumor cell-fibroblast crosstalk, promoting activation of FRCs and tumor radioresistance

Having identified TGFβ secreted by the tumor cells to impact FRC responses upon IR, we wanted to know how the contact of the two cell types is impacted by IR and TNC. Therefore, we established FRCs with stable TNC knockdown (shTNC) and confirmed effective TNC repression by western blot (Appendix Fig. S4A). Notably, neither TNC knockdown nor complete knockout affected FRC expansion in vitro (Appendix Fig. S4B). Our RNA seq and mass spectrometry analysis revealed that the FRCs express integrin α7β1, typically restricted to muscle tissue (Song et al, 1993, 1992; Liu et al, 2008; Tomatis et al, 1999), which was marginally affected by TNC depletion (Appendix Fig. S4C,D). We subsequently used, integrin α7β1 along with GP38, to identify FRCs by flow cytometry and staining. Upon a OSCC13/FRC coculture we observed sorting of cells into tumor cell islets surrounded by the FRCs, which generated a 3D-like TNC-rich stroma, resembling the tumor islet-stroma segregation seen in human HNSCC (Figs. 4A and EV4A,B, Spenlé et al, 2020). In contrast, TNC-deficient FRCs failed to segregate and instead intermingled with tumor cells, suggesting a loss of structural organization (Figs. 4A and EV4A,B).

As OSCC13 cells are radiosensitive (Spenlé et al, 2021), we applied 10 Gy which is in a similar range of 5 ×2 Gy applied in the clinic (Combs et al, 2005) to reach an optimal effect. To dissect cell-type-specific responses to IR, the two cell populations were magnetically sorted using antibodies against GP38 and integrin α7β1. Sorted FRCs (GP38⁺/α7β1⁺) expressed GP38 and integrin α7β1 but lacked *Cdh1* encoding for E-cadherin (Ecad), whereas the negative fraction (GP38⁻/α7β1⁻) expressed high levels of *Cdh1*, identifying them as tumor cells (Appendix Fig. S4E,F). Subsequently, we compared gene expression in the irradiated cocultures to that of the monocultures. In the irradiated cocultures, FRCs

became more activated by IR than in the monocultures as indicated by higher levels of *Ccl21*, TNC, GP38, α7β1, and *Col1a2* expression (Figs. 4B and EV4C–F; Appendix Fig. S4G–L). Except for IR-induced TNC mRNA changes, no other expression changes were observed in the TNC-depleted FRCs for all investigated genes, supporting an altered, potentially immature FRC phenotype in the absence of TNC (Figs. 4B and EV4D–F; Appendix Fig. S4G–L). Induction of TNC and GP38 in the irradiated FRCs was also confirmed by immunofluorescence staining of the cultured cells (Figs. 4C and EV4C).

In order to investigate the resistance of these cells to irradiation, we measured the RNA and protein levels of Rad51, the main factor of homologous recombination repair (Shinohara et al, 1992). Interestingly, Rad51 was elevated in the cocultures of OSCC13 and FRCs after IR, in a TNC-dependent manner, with this effect more pronounced in the tumor cells than in the FRCs (Fig. 4C,D; Appendix Fig. S4M,P).

One process that, along with its other effects, is well known to contribute to the radioresistance of the cancer cells, is epithelial to mesenchymal transition (EMT) (Theys et al, 2011; Nantajit et al, 2015). This phenomenon was also observed in our study. Specifically, in the irradiated cocultures, the expression of various markers analyzed apart from TNC itself, described as early EMT marker (Lüönd et al, 2021), also *Tgfb1, Vim* and *Twist* were elevated in the OSCC13 cells with a greater increase in comparison to the monocultures, again in a TNC-dependent manner (Figs. 4E,F and EV4C,D,G; Appendix Fig. S4Q–S). In parallel, *Cdh1* expression levels were decreased after IR, suggesting that upon IR, OSCC13 cells cocultured with TNC-expressing FRCs start undergoing EMT (Fig. EV4H; Appendix Fig. S4T). To further investigate the underlying mechanism, and since *Tgfb1* was increased in the OSCC13 tumor cells after irradiation (Fig. 4E), we used the TGFβRI inhibitor GW to abolish its action and investigated whether TGFβ signaling has an effect on the protein levels of Vim and the phosphorylated nuclear Smad3 (pSMAD3). Indeed, while both Vim and pSMAD staining increased in the cocultured OSCC13 cells after irradiation, this effect was diminished with the inhibitor, suggesting TNC-dependent pathway activation by IR (Fig. 4G–I). Also, *Cxcl12* (induced by IR in the FRCs, Appendix Fig. S3D) was increased in the cocultures with WT FRCs upon IR, whereas *Il6* remained unchanged, indicating a potential role for CXCL12 that has to be examined in the future (Appendix Fig. S4J,K).

Collectively, these results suggest that the coculture promotes efficient Double Strand Break (DSB) repair, specifically homologous recombination and plasticity in the tumor cells upon IR, thereby enhancing their radioresistance. In turn, the tumor cells impact the FRC phenotype through TGFβ signaling; upon irradiation, FRC numbers and activation increase, enforcing the formation of an IS-TME. TNC is a driving force of this intercellular communication, as TNC-depleted FRCs fail to respond to IR and to trigger the induction of radioresistance and plasticity in the tumor cells.

## TNC expressing FRCs generate an IS-TME and promote OSCC13 tumor growth

To determine whether FRCs play a role in tumor growth and associated IR responses, we engrafted OSCC13 cells together with

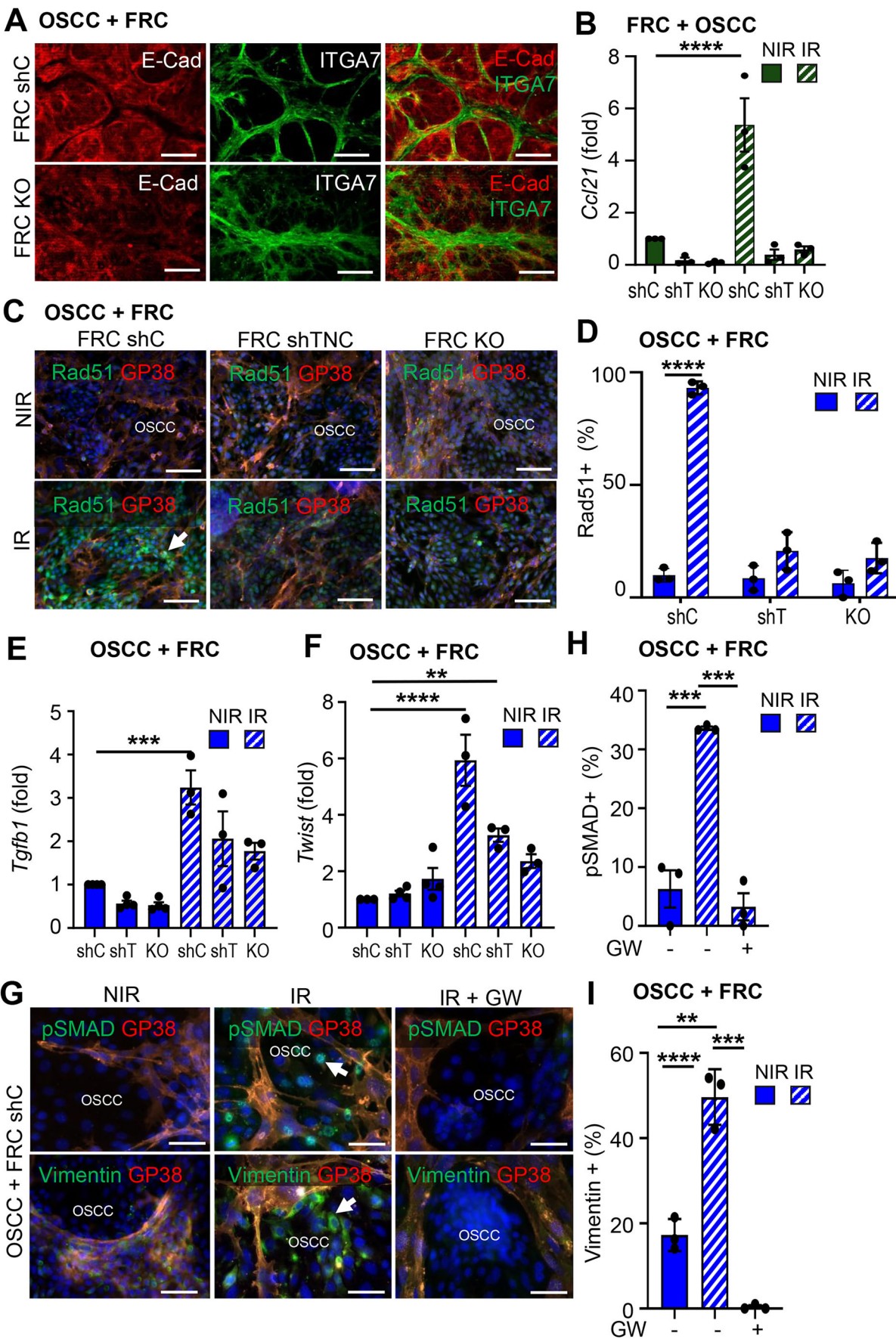

◀

**Figure 4. The FRC-OSCC crosstalk impacts irradiation responses in a TNC-dependent manner.**

(A) immunofluorescence images of OSCC13 (OSCC) and FRC (WT, shC, TNCKO) after coculture in a 2:1 ratio. Scale bar, 200 μm. (B) Gene expression (qRTPCR) analysis of Ccl21 expression in the cocultured FRCs after MACS, before or 2 days after exposure to 10 Gy IR. (C, D) GP38 and Rad51 immunofluorescence images of the cocultured OSCC and FRC (shC, shTNC,TNCKO) cells before or 2 days after exposure to 10 Gy IR (C) and quantification (%) of Rad51 positive OSCC cells (D). Arrows indicate the OSCC cells expressing Rad51. Scale bar, 200 μm. (E, F) qRTPCR analysis of *Tgfb1* (E) and *Twist* (F), in the cocultured OSCCs before or 2 days after exposure to 10 Gy IR. (G) Immunofluorescence images of OSCCs and FRCs, cocultured in a 2:1 ratio for a total of 4 days, 2 days prior to and 2 days after exposure to 10 Gy IR, under GW treatment for the indicated molecules. Arrows indicate the OSCC cells that undergo EMT. (H, I) Signal quantification in the OSCC cells expressed as % of all tumor cells. Scale bar, 50 μm, N = 3 experiments; Error bars represent mean ± STDEV; ordinary one-way ANOVA test, **P < 0.01. ***P < 0.005. ****P < 0.0001. The exact P values are listed on Appendix Table S5. Source data are available online for this figure.

the FRCs. Single tumor cell engraftments showed a very heterogeneous tumor latency or no engraftment in the given 15-week time frame. In contrast, co-engraftment with the FRCs reduced the tumor latency, resulting in consistent tumor growth across engraftments (Fig. 5A). Tissue staining and qRTPCR analysis revealed more abundant FRCs in the coengrafted tumors (Figs. 5B,C and EV5A). However, also in the single OSCC grafted tumors, ERTR7+ FRCs were present, suggesting that host-derived FRCs have infiltrated the tumors. We further demonstrated that in the co-engrafted OSCC13/FRCs tumors TMTs were enforced characterized by prominent expression and alignments of collagen VI (ColVI, ERTR7), laminin (LM) and TNC (Figs. 5B and EV5A; Appendix Fig. S5A). This was associated with recruitment of leukocytes into the stromal compartment (imaged by CD45 staining) and the establishment of an IS-TME, as evidenced by an increased expression of *Col1a2, Tnc, Acta2, Ccr7, Ccl21, Tgfb1, Il10,* and *Il17* (Fig. 5C). To address whether TNC expressed by the FRCs plays a role in shaping the TME, we compared co-engraftments of tumor cells with either WT FRCs or TNCKO FRCs. Indeed, tumors with WT FRCs showed faster growth during the first 18 days than their TNCKO counterparts as well as the tumor cells engrafted alone (Figs. 5D and EV5B). After this initial delay in tumor growth, TNCKO FRC-engrafted tumors started to catch up, which could be explained by substantial host FRC infiltration highly expressing TNC. This can be seen by the presence of host-derived GP38+ cells expressing TNC, next to the engrafted TNCKO FRCs (only GP38 +) (Fig. EV5C; Appendix Fig. S5B). Nevertheless, at the endpoint, the WT FRC-engrafted tumors revealed clearly established TMTs and higher TNC expression levels, confirming major TNC-instructed differences implemented by the FRCs (Figs. 5E and EV5D; Appendix Fig. S5B,C).

In summary, FRCs co-engraftment promotes a TME in the neck with similarities to those seen in HNSCC, exemplified by enforced TNC and *Ccl21* expression, TMT formation and an IS-TME. Our findings indicate that TNC plays a key role in supporting the ability of FRCs to build an IS-TME and, to generate a functional milieu in the TdLNs, in particular relevant upon IR.

## Targeting TNC with MAREMO peptide MP5 reduces tumor cell proliferation and plasticity upon IR

Having established that TNC expressing FRCs promote an IS-TME, we wanted to know whether tumor targeting of TNC would impact IR responses. For that, we used 3D cocultures of OSCC13 cells grown together with FRCs, forming spheroids (Fig. 6A; Appendix Fig. S6A,B). Irradiation of these spheroids with 10 Gy IR led to a

decrease of cell numbers with a similar tendency upon 5 ×2 Gy IR (Fig. 6B; Appendix Fig. S6C). This effect was even further pronounced by the addition of MP5 post-IR for 3 days, indicating reduced proliferation by MP5 in the given time frame (Fig. 6B). To confirm the delivery of MP5 inside the spheroids, we used its labeled surrogate Cy5-MP5, that we have already shown to colocalize with TNC in other tumors (Li et al, 2024). Consistently, Cy5-MP5 also colocalized with TNC-rich regions in the spheroids (Fig. 6A; Appendix Fig. S6A,B) and was similarly active in reducing DC retention as the untagged peptide (Fig. EV6A). Both peptides also impaired binding of the fibronectin type III (FNIII) 4-6 (FN4-6) molecule to the TNC FNIII 1-5 (TN1-5) molecule in a dose-dependent manner as demonstrated by a competitive ELISA with an IC50 of 3 μM (Cy5-MP5) and 5 μM (MP5), respectively (Fig. EV6B). A similar effect on reducing cell numbers upon IR in combination with MP5 was also seen in human CAL33 spheroids cocultured with human immortalized fibroblasts (TIF) where Cy5-MP5 also colocalized with TNC (Fig. 6C,D).

By staining, we observed enhanced Vim expression upon 10 Gy IR (and 5 × 2 Gy), indicative of EMT in the OSCC13 spheroids, which was blocked by Cy5-MP5 addition, revealing a similar effect on plasticity as previously reported for MP5 (Fig. 6E,F; Appendix Fig. S6B,D,E (Li et al, 2024).

Next, we addressed the TNC specificity of MP5 using two approaches. First, TNC-depleted mixed OSCC13 (shTNC)/FRC (TNCKO) spheroids were exposed to IR and MP5 as described and showed no reduction in cell numbers, unlike TNC-expressing spheroids (Fig. 6B; Appendix Fig. S6F). Second, we compared gene-expression profiles from the NT193 breast cancer cells exhibiting low TNC expression with those obtained upon MP5 treatment (Murdamoothoo et al, 2021; Li et al, 2024). Principal component analysis (PCA) revealed that MP5-treated cells cluster with shTNC cells, whereas PBS-treated controls cluster with shC cells (Fig. EV6C). This pattern indicates that MP5 induces a transcriptional shift that partially phenocopies TNC depletion. In line with this observation, MP5-treated tumor cells displayed transcriptional patterns more similar to shTNC than to control cells (Fig. EV6D). Importantly, several immune-related pathways were consistently enriched in both conditions. Gene Ontology categories linked to "adaptive immune response," "antigen presentation," and "cytokine response" were similarly upregulated following TNC knockdown and MP5 exposure (Fig. EV6E; Appendix Fig. S6G,H), suggesting that MP5 triggers downstream immune-regulatory programs typically suppressed by TNC.

In summary, our results suggest that targeting TNC with the MAREMO peptide MP5 after IR can enforce a reduction in cell numbers and reduce plasticity, two key events towards an improved

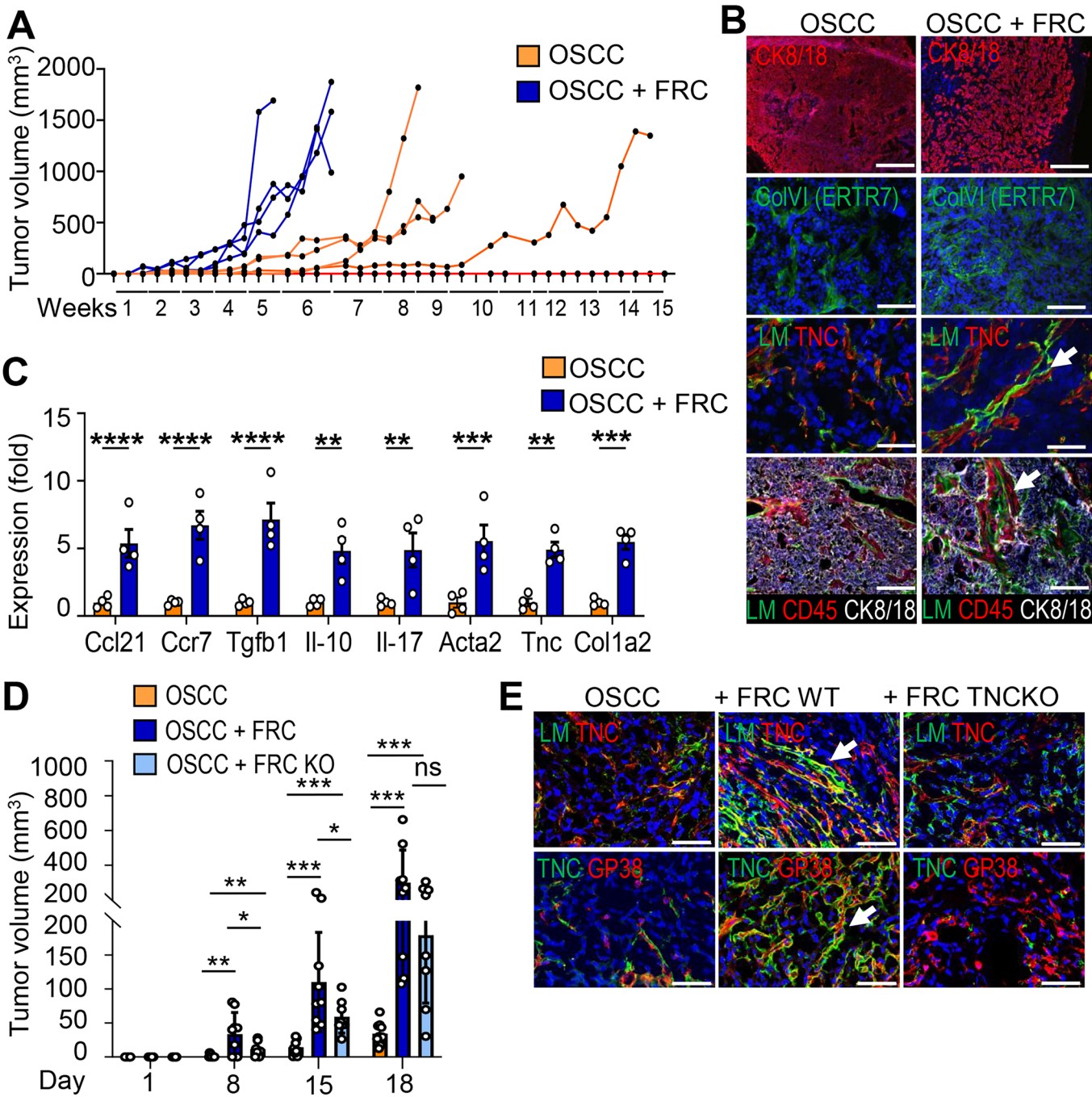

**Figure 5. Coengraftment of FRCs promotes OSCC tumorigenesis and an immunosuppressive TME.**

(A) Monitoring of tumor volumes upon engraftment of $3 \times 10^6$ OSCC13 (OSCC) cells alone (orange, $N = 5$ mice) or OSCC cells combined with WT FRC cells (blue, $N = 4$ mice), in a 5:1 ratio into the neck of C57Bl6 mice during the indicated time frame (1 to 15 weeks). (B) Representative immunofluorescence images for CK8/18, ERTR7 (Col VI), pan-laminin (LM), TNC, and CD45 in the neck coengrafted OSCC/FRC tumors or OSCC tumors. Arrows indicate the tumor matrix tracks (TMTs). The CK8/18 IF images shown are also presented with their corresponding H&E staining in Appendix Fig. S5A. The LM/TNC co-stained panel in OSCC/FRC tumors represents a cropped region of the image shown uncropped in Fig EV5A, tumor #6. Scale bar, 200 μm. (C) Gene expression analysis (qRTPCR) for the indicated genes in the engrafted tumors displayed in (A). Mean ± SEM; Kruskal–Wallis test, *$P < 0.05$. (D) Monitoring of tumor volume upon engraftment of OSCC cells ($3 \times 106$) alone (orange, $N = 10$ mice) or OSCC cells combined with WT FRC cells (dark blue, $N = 10$ mice), or OSCC cells combined with FRC TNCKO cells (light blue, $N = 10$ mice), in a 5:1 ratio into the neck of C57Bl6 mice during the indicated time frame (days 1–18). Mean ± STDEV. Ordinary one-way ANOVA with *$P < 0.05$, **$P < 0.01$, ***$P < 0.005$. (E) Representative IF images for LN/TNC and GP38/TNC in the engrafted tumors displayed in (D). Arrows indicate the tumor matrix tracks (TMTs). Scale bars, 50 μm. The exact $P$ values are listed in Appendix Table S5. Source data are available online for this figure.

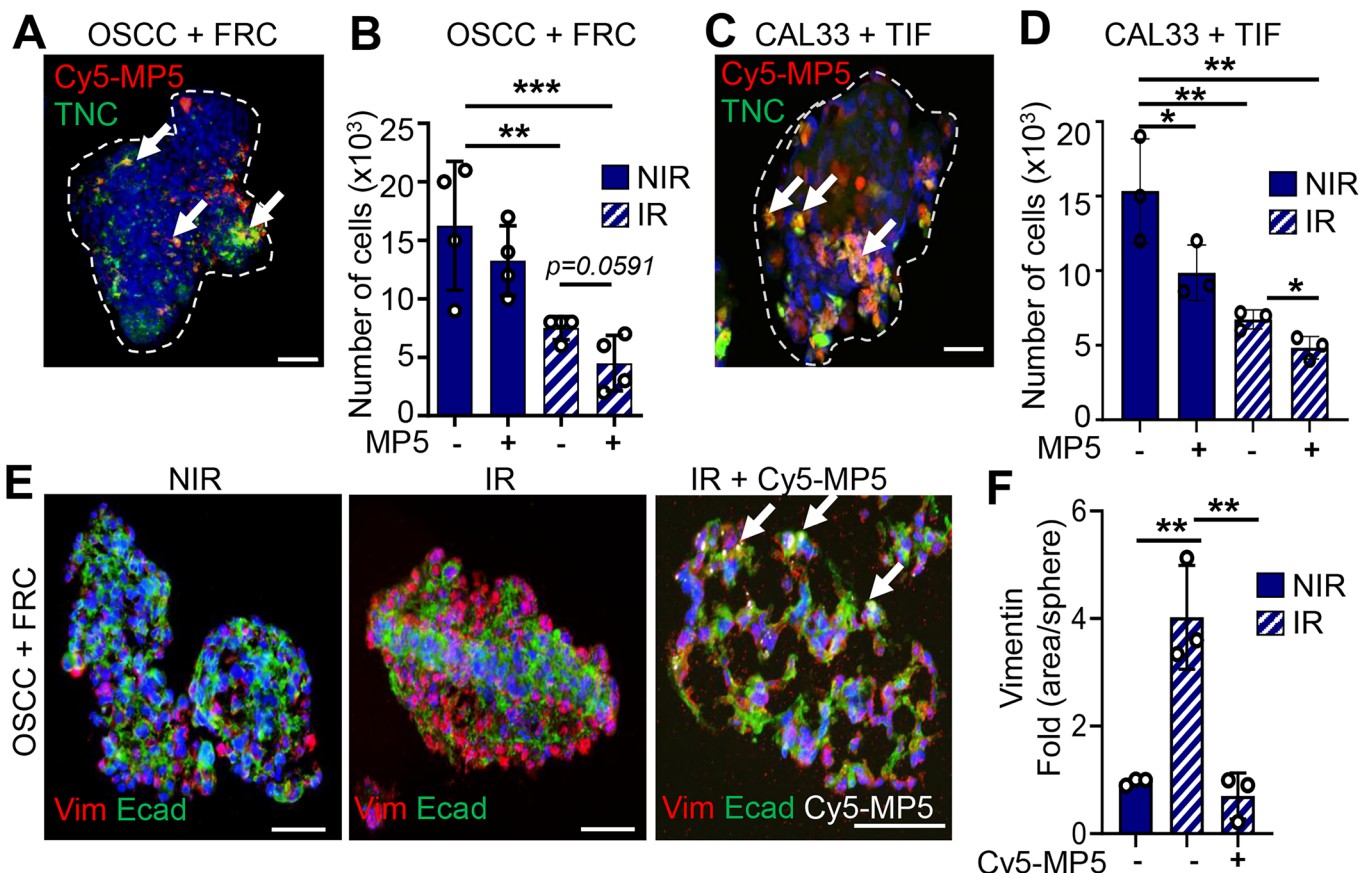

**Figure 6.   Targeting TNC with MAREMO peptide MP5 reduces tumor cell proliferation and plasticity upon IR.**

After 7 days OSCC13 (OSCC)/FRC spheroid cocultures (2:1 ratio) (**A, B, E, F**) and CAL33/TIF spheroid cocultures (1:1 ratio) (**C, D**) cells were exposed to 10 Gy and subsequently MP5 (**B, D**) or Cy5-MP5 at 50 µg/ml each for another 3 days (**E, F**) before IF imaging for the indicated molecules (**A, C, E**), quantification of cells (**B, D**) and quantification of vimentin expression (**E**). Representative IF images ($N = 3$ experiments with 5–10 spheroids per condition) are shown for the indicated molecules. The white dotted line indicates the spheroid border. Arrows indicate colocalization of Cy5-MP5 with TNC. Individual fluorescence channels of (**A**) and the IR+Cy5-MP5 condition in (**E**) are shown separately in Appendix Fig. S6A,B. Scale bars, 50 µm. Ordinary one-way ANOVA test with *$P < 0.05$, **$P < 0.01$, ***$P < 0.005$. The exact $P$ values are listed in Appendix Table S5. Source data are available online for this figure.

tumor remission by IR. Moreover, the MP5 effect is TNC-dependent.

## Patients with irradiated HNSCC have a shorter survival when having FRCs highly expressing TNC

Since FRCs became more abundant following IR and played a critical role in promoting tumor proliferation, we hypothesized that they may serve as promising biomarkers for predicting survival in HNSCC patients. By using publicly available expression and survival data, we determined whether expression of a 9-gene signature comprising *CCL19*, *CCL21*, *TNC*, *VCAM1*, *PDPD*, *COL6A1*, *COL1A1*, *ACTA2*, and *LTBR* (Fletcher et al, 2015; Ferreira et al, 2021; Förster and Moschovakis, 2013; Spenlé et al, 2021), and downregulated in FRCs lacking TNC (Fig. 2C), is correlated with patients' overall survival (OS) in a total of 267 TCGA-HNSCC patients who had undergone radiotherapy. Individual gene expression did not correlate with survival. Given the key role of TNC in regulating FRCs, patients were initially stratified into TNC-high and TNC-low groups based on the median TNC

expression. Each group was then further subdivided based on the median FRC signature expression into signature-high and signature-low subgroups. Kaplan–Meier survival analysis demonstrated a significantly lower median overall survival (OS) in the TNC-high group ($P = 0.039$, Fig. 7A). In contrast, the prognostic value of the FRC signature was reduced in patients with lower TNC expression ($P = 0.8$, Fig. 7B). In addition, Univariate Cox regression analysis identified T stage, N stage, and FRC signature expression as factors associated with OS, which were subsequently included in a multivariate Cox regression analysis. The Cox model revealed that higher FRC signature expression was an independent adverse factor for OS (HR = 1.863, 95% CI: 1.022–3.395, $P = 0.042$, Appendix Table S4). In summary, these findings suggest that an increased FRC signature expression is associated with a poorer prognosis in HNSCC patients who received RT, suggesting that tumor FRCs highly expressing TNC may represent a promising target to improve tumor eradication by radiotherapy.

In summary, our data demonstrate that while IR induces cell death in the majority of the tumor cells, it also increases the abundance of FCRs following an initial reduction in cell numbers.

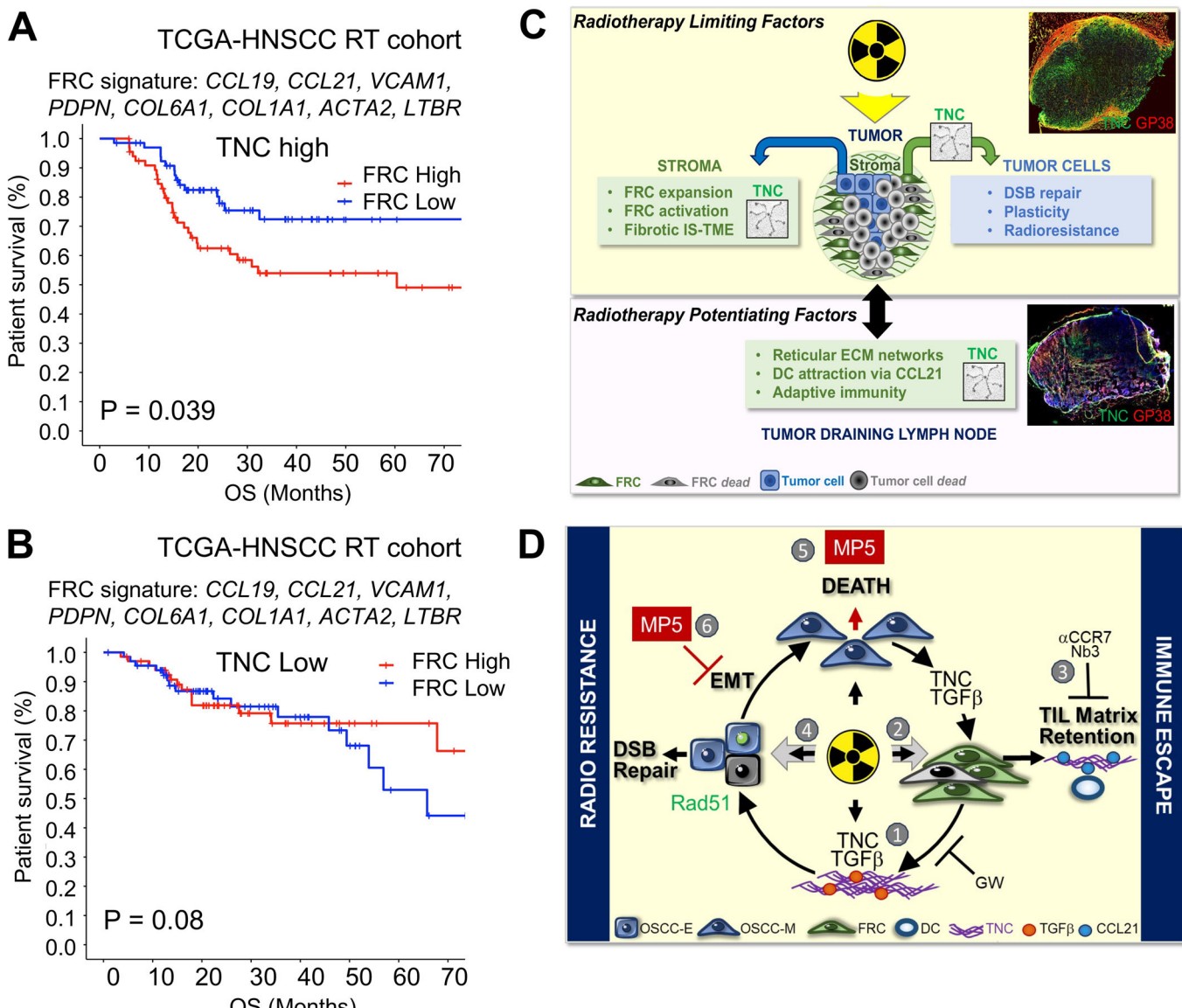

Crosstalk with tumor cells promotes FRC expansion and activation upon IR, leading to increased secretion of TNC, CCL21, and other molecules, thereby reinforcing an IS-TME that potentially protects the tumor cells from immune-mediated cell death. In parallel, crosstalk with the FRCs enforces plasticity and radioresistance in the surviving tumor cells (Fig. 7C). TNC plays a critical role in the tumors by facilitating EMT in the tumor cells and determines the FRC phenotype, which involves TGFβ signaling. While TNCKO FRCs demonstrate enhanced cellular radioresistance, this FRC survival does not seem directly to translate into tumor radio-resistance. Instead, TNC-depleted FRCs fail to establish an IS-TME and to maintain functional TdLN-mediated adaptive immunity, thereby likely contributing to radioresistance. Last, FRCs generate TNC-containing reticular fibers and ECM networks and conduits regulating adaptive immunity (Song et al, 2023), and most importantly, express CCL21, as we have shown, involving TGFβ signaling induced by the irradiated tumor cells (Fig. 7D).

## Discussion

This study identifies TNC as a central and context-dependent determinant of radiosensitivity in HNSCC, exerting distinct and even opposing effects in the tumor and in the TdLNs. Our findings reveal a dual and compartmentalized role: first, in the TdLNs, TNC provides essential baseline structural and immunoregulatory functions that are required for efficient IR-induced tumor regression, yet these functions seem independent of irradiation itself. Second, within the tumor, TNC instead contributes to the establishment of an IS-TME that undermines the therapeutic benefit of IR. In both settings, fibroblastic reticular cells (FRCs) emerge as key mediators of TNC biology. This duality reflects the compartmentalized nature of TNC and FRC functions. In the TdLNs, TNC supports the integrity of the reticular network, presumably allowing the spatial organization necessary for dendritic cell (DC)–T cell interactions and optimal priming. Such

◄ **Figure 7. HNSCC patient survival is correlated with high TNC and expression of the FRC signature.**

(A, B) Kaplan–Meier analysis of a nine-gene FRC signature in tumors with levels of TNC above (A) or below the median (B). $N = 268$ patients with $n = 134$ TNC high and $n = 134$ TNC low levels. Kaplan–Meier survival curves were plotted by using R function. Multivariate Cox regression analysis is provided in Appendix Table S4. (C) Roles of TNC in modulating radiosensitvity: TNC plays a compartmentalized and dual role in regulating tumor radiosensitivity in HNSCC with distinct functions in the tumor (yellow background shading) opposed to the tumor-draining lymph nodes, TdLNs (purple background shading). In the tumor, ionizing irradiation (IR) induces massive cell death which launches an adaptive immune response involving the TdLNs, with activates immune cells returning into the tumor as indicated by a two-sided arrow. In HNSCC, the multiple tumor islets are surrounded by TNC-rich stroma where FRCs are a major producer of TNC (central cartoon showing one stroma-islet unit, image of a tumor stained for TNC (green) and GP38 (red) (upper right), original image in Appendix Fig. 5C)). In the tumor and TdLN compartments, FRCs (green box) are important, and their function is regulated by TNC. Surviving FRCs expand in the tumor stroma, which is linked to a cell-cell crosstalk (indicated by the blue arrow), where tumor cells activate FRCs to increase production of CCL21 and TNC. This amplifies the immunosuppressive tumor microenvironment (IS-TME) where TNC contributes to the generation of stromal tumor matrix tracks, TMT. In addition, the interaction with FRCs (indicated by the green arrow) promotes tumor cell survival and resistance involving DSB repair and plasticity/EMT, which is TNC-dependent (blue box). Conversely, in the TdLNs, FRCs organize the reticular ECM networks that regulate adaptive immunity, where TNC is important (image of a TdLN stained for TNC (green) and GP38 (red) (upper right), original image in Appendix Fig. 1Q). Here, FRC-derived CCL21 attracts DCs and immature CD8 + T cells, generally supporting anti-tumor immunity. Critically, our RNA profiling, flow cytometry, staining and quantification results suggest a role of TNC in the organization of the TdLNs, and in the regulation of the numbers of T and B cells and, DC and T-cell function. Altogether, an altered TdLN function may contribute to an impaired adaptive anti-tumor response and reduced radiotherapy efficacy in TNCKO mice. This dual role of TNC, detrimental in the tumor stroma (TNC expressed in TMTs) but essential in TdLNs (TNC expressed in the reticular networks), determines the overall radiotherapy outcome. Representative microscopy images previously shown in Appendix Figs. S1Q and S5B are used in this summary cartoon. (D) Impact of targeting TNC with a MAREMO peptide: cellular and molecular responses within the tumor compartment (yellow background) are illustrated in the presence of the MAREMO peptide MP5. In addition to blocking the tumor-promoting functions of TNC and counteracting immune evasion (Li et al, 2024), MP5 further enhances radiosensitivity, as demonstrated in this study. However, whether and how MP5 directly influences the TdLNs remains to be investigated. Whereas IR induces cell death in the majority of tumor cells and FRCs (gray shading), here we focus on the surviving FRCs, where IR triggers TNC and TGFβ expression, which form complexes known to potentiate TGFβ activity (Sun et al, 2019; Loustau et al, 2022) (1). This mechanism contributes to establishing an IS-TME that favors tumor regrowth. Crosstalk between surviving tumor cells and FRCs (two-sided gray arrow) promotes FRC activation through TGFβ signaling, resulting in increased secretion of TNC and CCL21. Collectively, these events reinforce an IS-TME where DCs become entrapped and functionally impaired by TNC (Loustau et al, 2022; Spenlé et al, 2020) (2). Analogous to CCR7 inhibition (Spenlé et al, 2020), blockade of TNC function in DCs using the TNC-specific nanobody Nb3 (Dhaouadi et al, 2021) abolishes the immunosuppressive DC-matrix-chemoretention effect enhanced by irradiation, further highlighting the central role of TNC (3). As described in Fig. 7C, TNC also regulates the interplay between OSCC tumor cells and FRCs, increasing tumor cell survival and radioresistance through induction of EMT (via TGFβ) and Rad51, in a TNC-dependent manner (4). Targeting TNC with the MAREMO peptide MP5 enhances tumor radio-responsiveness by promoting tumor cell death (5) and inhibiting irradiation-induced EMT (5). Together, these findings substantiate a pivotal role of TNC in radioresistance and identify its therapeutic targeting as a promising strategy to improve radiotherapy efficacy. Source data are available online for this figure.

roles align with a proposed evolutionary function of TNC in adaptive immunity and in the formation of immune-regulating stromal networks in the TdLNs, an idea supported by prior work (Orend and Tucker 2021; Spenlé et al, 2015; Panocha et al, 2025). In contrast, within the tumor, the same TNC–CCL21 axis presumed to promote immune cell positioning in lymphoid tissue is co-opted to retain CCR7+ immune populations in the stromal matrix, limiting their infiltration into tumor islets as already shown in OSCC tumors (Spenlé et al, 2020). A similar mechanism applies to CD8 + T cells, which are detained in TNC-rich stroma through a TNC–CXCL12 interaction (Loustau et al, 2022; Murdamoothoo et al, 2021) thereby reducing their cytotoxic impact. The net therapeutic outcome of IR therefore likely depends on the balance between TNC's immune-supporting functions in the TdLNs and its immune-restrictive activity in the TME, where IR enhances TGFβ, TNC, and CCL21 expression within the tumor, reinforcing a "sticky" IS-TME for DCs and T cells.

Several IR-activated pathways can mediate the IS-TME. Among them, TGFβ signaling is one of the most prominent regulators of radioresponse (reviewed in Farhood et al, 2020). As shown in this study, TGFβ signaling activation also occurs in the irradiated OSCC cells and FRCs. TGFβ signaling plays a key role as it activates and triggers the secretion of CCL21 in the irradiated FRCs and induces EMT in the OSCC. Our results also indicate that TGFβ signaling is regulated by TNC, playing multiple roles in the observed IR phenotype (Fig. 7D). We demonstrate that TGFβ signaling is specifically activated in TNC-expressing (but not TNC-depleted) FRCs and tumor cells upon IR, and that pharmacological inhibition of TGFβRI attenuates downstream responses. While the structural basis for this TNC-dependent TGFβ activation remains unclear,

previous studies suggest that TNC may facilitate TGFβ signaling through direct binding within the FNIII domains (Loustau et al, 2022) and/or recruitment of latent TGFβ binding protein (Aubert et al, 2021), two mechanisms worthy of further investigation in the HNSCC context. Moreover, TNC-expressing FRCs promote EMT markers through TGFβ, which may also contribute to radio-resistance (this study and Theys et al, 2011; Sun et al, 2019; Zhou et al, 2020).

High intratumoral CCL21 expression in TNC-expressing tumors upon IR may disrupt the normal chemokine gradient toward the TdLNs. As previously proposed (Spenlé et al, 2020), this local enrichment of CCL21 could impair the migration of primed APCs from the tumor to the TdLNs, thereby reducing the effectiveness of adaptive anti-tumor immunity. This mechanism may explain the observed paradox of increased leukocyte accumulation in the tumor yet reduced infiltration into the tumor islets. This mechanism also seems to apply upon IR, as evidenced by an increased accumulation of immature CCR7 + CD4 + T and Treg cells in the tumors of irradiated WT mice, an effect not observed in the TNCKO mice (Fig. 7C,D).

IR is well known to induce immunogenic cell death (ICD), which is important for tumor eradication (Golden and Apetoh, 2015; Lhuillier et al, 2019). IR-induced tumor regression in TNC-expressing hosts was accompanied by increased accumulation of CCR7+ leukocytes, including DCs, macrophages, and CD4 + T cells, though infiltration into tumor islets paradoxically decreased. This compartment-specific immune retention suggests that TNC-mediated stromal sequestration of immune cells *via* CCL21 (and CD8 + T cells *via* CXCL12) may impair direct anti-tumor activity despite heightened systemic immune response. The

interplay between IR-induced ICD, TGFβ-driven stromal remodeling, and TNC-mediated immune immobilization likely shapes this outcome, but the relative contributions of each remain to be fully dissected (Fig. 7C).

The minimal alteration in proliferation, gene, and protein expression which is in contrast to the TNC-expressing tumors that showed a strong reduction in proliferation upon IR and a high induction of an IR-response gene and protein expression signature further supports radioresistance of the TNC-depleted tumors. As FRCs are generating reticular fibers and conduits in the TdLNs and are impaired in producing several ECM and chemoattracting molecules in the absence of TNC, it is conceivable that this may impact the TdLN functions and contribute to the radioresistance phenotype in the TNCKO mice. Indeed, our gene expression data, tissue staining, and quantification of T and B cells suggest a strongly deregulated organization and immunity of the TNC-depleted TdLNs.

We discovered that the identified FRC signature correlates with shorter HNSCC patient survival, in particular when TNC levels are high, suggesting an important role of TNC-expressing FRCs in HNSCC. Although TNC-high FRCs represent a minority of cells within the tumor, their immunosuppressive effects are relevant, suggesting that tumor FRCs might be a promising target for anti-cancer therapy as recently proposed (Onder et al, 2024). However, direct FRCs and carcinoma-associated fibroblasts targeting has not yielded consistent benefits (reviewed in Zhang et al, 2023; Chen, McAndrews, and Kalluri, 2021, Onder et al, 2024), suggesting that some fibroblasts may possess anti-tumor functions that should be preserved. We propose that this also applies to the lymph node-derived FRCs that may remain in the HNSCC. We hypothesize that these FRCs initially fulfill an anti-tumor function, similar to what was seen in lung cancer, where FRCs generated a protective T-cell environment (Onder et al, 2024), but may undergo transformation into pro-tumorigenic cells, where high TNC expression contributes to the phenotype. Supporting this view, our co-engraftment experiments show that LN-derived FRCs markedly enforce IS-TME formation, and FRCs lacking TNC are significantly less capable of promoting tumor growth. We suggest that the depletion of pro-tumorigenic FRCs may reduce immune suppression and enforce IR-induced tumor eradication. Targeting TNC, which is highly expressed by the tumor FRCs, could be a promising therapeutic strategy. Supporting this hypothesis, we have recently demonstrated that targeting TNC with the MAREMO mimetic peptide MP5 (Loustau et al, 2022) induces depletion of FRCs among other fibroblasts and restores anti-tumor immunity (Li et al, 2024). Here, we show that MP5 combined with IR further reduces cell numbers and tumor cell plasticity, hallmarks of improved therapy response. Our study has limitations that should be acknowledged. While the 4NQO mouse model and human coculture systems provide mechanistic insights, validation in patient-derived xenografts, organoids, experimental tumors, and prospective clinical studies is required to fully establish the relevance of targeting TNC-dependent radioresistance in human HNSCC, also including HPV+ tumors. In addition, although TCGA analyses support a link between TNC-high FRCs and poor survival, causality must be confirmed experimentally. Finally, although we provide evidence that immunity in the TdLNs of TNC-depleted tumor mice is deregulated, which likely contributes to radioresistance, the underlying mechanism has to be worked out.

Collectively, our findings shed light on our understanding of the interactions between the tumor and its IR-modulated microenvironment, reveal compartmentalized roles of FRCs and TNC in the tumor and TdLN, and suggest the targeting of TNC in particular in tumor FRCs as a potential strategy to improve radioresistance overcomes in HNSCC, which may warrant investigation in other cancers.

# Methods

**Reagents and tools table**

| Reagent/resource | Reference or source | Identifier or catalog number |
|---|---|---|
| **Experimental models** | | |
| C57BL/6J wild-type (WT) mice | Charles River | C57BL/6J |
| TNCKO mice on C57BL/6J background | Spenlé et al, 2020 | N/A |
| OSCC13 cell line (4NQO-induced mouse tongue tumor) | Spenlé et al, 2021 | N/A |
| Fibroblast reticular cells (FRCs) from mouse lymph nodes | This study and Spenlé et al, 2021 | N/A |
| CAL33 | Bozek et al, 2006 | N/A |
| TIF | Gift from Dr J. Norman (Beatson Institute, Glasgow, UK) | N/A |
| DC2.4 dendritic cell line (mouse) | Merck | SCC142 |
| TCGA-HNSC patient cohort | TCGA | TCGA-HNSC |
| Clinical data for TCGA-HNSC | UCSC Xena | xenabrowser.net |
| **Recombinant DNA** | | |
| Lentiviral shRNA vector targeting TNC (shTNC) | Sigma | shTNC: CCGGGCATCAA-CACAACCAGTC-TAACTCGAGT-TAGACTGGTTG-TGTTGATGCTT-TTTG |
| Non-targeting shRNA lentiviral control (shC) | Sigma | SHC202V |
| Recombinant his-tagged human TNC FNIII 4–5 (TN 4-5) | Dr. Manuel Koch | N/A |
| Recombinant strep-tagged human fibronectin FNIII 4–6 (FN4-6) | Loustau et al, 2022 | N/A |
| **Antibodies** | | |
| Tenascin-C (TNC), mAb mTNC12 (1:100) | Homemade | N/A |
| Tenascin-C (TNC) (IF 1:200, WB 1:1000) | Merck/Millipore | AB19011 |
| Ki67 | Thermo Fisher | MA5-14520 |
| Collagen VI/ERTR7 (1:200) | Santa Cruz | sc73355 |
| CD11c | BD Biosciences | 550275 |
| Laminin Ln6 7 s (1:200) | Simo et al, 1992 | N/A |
| CK8/18 | Progen | GP11 |
| CD31 | BD Pharmingen | AB_39666 |

| Reagent/resource | Reference or source | Identifier or catalog number |
|---|---|---|
| Podoplanin (gp38) (1:200) | Invitrogen | MA5-16113 |
| CCL21 | R&D Systems | AF457 |
| Collagen XII (ColXII), clone Kr33 | Dr M. Koch | N/A |
| Fibronectin | Sigma | F3648 |
| Collagen IV, α2 | Dr Arcangelis 1996 | Col IV 2a |
| VCAM1 | Merck/Millipore | CBL1300 |
| Vimentin (1:50) | Santa Cruz | sc3260 |
| E-cadherin (1:300) | Cell Signaling Technology | 24E10, #3195 |
| Integrin α7 (ITGA7) (1:100) | Bio-Techne | MAB3518 |
| pSMAD3 (Ser423/Ser425) (1:200) | Rockland Immunochemicals | 600-401-919S |
| Rad51 (1:100) | Calbiochem | PC130 |
| CD3e (1:200) | BD Biosciences | 550275 |
| B220 (1:200) | Invitrogen | 14-0452-82 |
| GAPDH (1:2000) | Cell Signaling Technology | 14C10, #2118 |
| Anti-strep-tag antibody | Abcam | ab184224 |
| Biotinylated secondary antibody | Vector Laboratories | PK-4001 |
| ABC detection kit | Vector Laboratories | ABC system (used with PK-4001) |
| CD45-FITC | eBioscience/ Thermo Fisher | 11-0454-85 |
| CD11c-PE | eBioscience/ Thermo Fisher | 12-0114-81 |
| MHCII-APC-eFluor 780 | eBioscience/ Thermo Fisher | 47-5321-80 |
| CCR7-PerCP-Cy5.5 | eBioscience/ Thermo Fisher | 45-1971-82 |
| CK10-Alexa Fluor 700 | Novus Biologicals | NBP2-34752AF700 |
| CD31-FITC | eBioscience/ Thermo Fisher | 11-0311-81 |
| Podoplanin (gp38)-PE | eBioscience/ Thermo Fisher | 12-5381-80 |
| CD11b-Alexa Fluor 700 | eBioscience/ Thermo Fisher | 56-0112-80 |
| F4/80-APC-eFluor 780 | eBioscience/ Thermo Fisher | 47-4801-82 |
| CD3-PE | eBioscience/ Thermo Fisher | 12-0031-81 |
| CD4-APC-eFluor 780 | eBioscience/ Thermo Fisher | 47-0041-80 |
| Foxp3-PE-Cy7 | eBioscience/ Thermo Fisher | 25-5773-80 |
| CD80-Alexa Fluor 700 | eBioscience/ Thermo Fisher | 56-0801-82 |
| CD86-PE-Cy7 | eBioscience/ Thermo Fisher | 25-0862-80 |

| Reagent/resource | Reference or source | Identifier or catalog number |
|---|---|---|
| CD25-Alexa Fluor 700 | eBioscience/ Thermo Fisher | 56-0251-80 |
| CCR7-neutralizing antibody | R&D Systems | MAB3477 |
| GP38 (for MACS 1 µg/sample) | BioLegend | 156202 |
| **Oligonucleotides and other sequence-based reagents** | | |
| MP5 | Loustau et al, 2022 | N/A |
| Cy5-MP5 | Li et al, 2024 | N/A |
| RT qPCR primers | Forward | Reverse |
| Ccl21 | TCCAAGGGCTG-CAAGAGA | TGAAGTTCGTG-GGGGATCT |
| TGF beta | TGACGTCACTG-GAGTTGTACGG | GGTTCATGT-CATG-GATGGTGC |
| Cadh1 | CAGCCTTCTTTT-CGGAAGACT | GGTAGA-CAGCTCCCTAT-GACTG |
| Vimentin | CCAACCTTTTCT-TCCCTGAAC | TTGAGTGGGTG-TCAACCAGA |
| Twist | AGTGTTTGG-CAGGGGACA | CCCATCCCCTG-GGTATCT |
| Tenascin-C (murine) | CAGGGATA-GACTGCTCT-GAGG | CATTGTCC-CATGCCA-GATTT |
| Acta2 | TCCTATGTGGGT-GACGAGGC | TACATGGCTGG-GGTGTTGAA |
| Gp38 | ACAACCA-CAGGTGC-TACTGGAG | GTTGCTGAGGT-GGACAGTTCCT |
| Col1a2 | TTCTGTGGGTC-CTGCTGGGAAA | TTGTCACCTCG-GATGCCTTGAG |
| Ki67 | GAGGA-GAAACGCCAAC-CAAGAG | TTTGTCCTCGG-TGGCGTTATCC |
| Itga7 | TCTGTCAGAG-CAACCTCCAGCT | CTAT-GAACGGCTGCC-CACTCAA |
| Rad51 | TGGAGGCTGTT-GCTTATGCACC | GCTGGTGAAA-CTCAGTTGCCG-T |
| Ccr7 | CTCCTTGTCATT-TTCCAGGTG | TGGTATTCTCG-CCGATGTAGT |
| Il10 | ATC-GATTTCTCCCCT-GTGAA | TGTCAAATT-CATT-CATGGCCT |
| Il17 | Taqman probe: Mm00439618_m1, Thermofisher | |
| **Chemicals, enzymes, and other reagents** | | |
| 4-NQO | Sigma-Aldrich | N8141 |
| Isoflurane (2%) | Minerve system | N/A |
| Ocrygel | Zooplus | N/A |
| RPMI-1640 medium | Dutcher | RPMI-1640 |
| Dispase | Roche | 04942078001 |
| Collagenase P | Roche | 11213857001 |

| Reagent/resource | Reference or source | Identifier or catalog number |
|---|---|---|
| DNase I | Invitrogen | 18068015 |
| Fetal bovine serum (FBS) | Dutscher | S00AC3000C |
| DMEM high-glucose | Dutscher | L0102-500 |
| DMEM/F12 (4.5 g/L glucose) | ThermoFisher | 10565018 |
| Penicillin-Streptomycin (PenStrep) | Dutscher | P06-07100 |
| Gentamicin | ThermoFisher Scientific | 15750060 |
| Puromycin | ThermoFisher | J67236 |
| Hydrocortisone | Sigma | H0888 |
| Trypsin-EDTA | PanBiotech | P10-023100 |
| Recombinant mouse TGFβ1 | Bio-Techne | 7666-MB |
| TGFβR inhibitor GW788388 | Selleckchem | GW788388 |
| TRIzol reagent | Invitrogen | 12044977 |
| MultiScribe reverse transcriptase | Applied Biosystems | 10117254 |
| Sybr Green Master Mix | ThermoFisher Scientific | 4344463 |
| Fast TaqMan Mix | ThermoFisher Scientific | 4444557 |
| Gapdh and Rpl19 TaqMan assays | Life Technologies | 433764 T |
| Phosphatase inhibitor cocktail | Santa Cruz Biotechnology | sc-45045 |
| Protease inhibitor cocktail | Roche | 05892970001 |
| Bradford assay | Bio-Rad | 5000001 |
| Laemmli buffer | Bio-Rad | 1610737 |
| Precast 4–20% SDS-PAGE gels | Bio-Rad | 4561096 |
| Nitrocellulose membranes | Bio-Rad | 1620113 |
| Blocking-Grade Blocker | Bio-Rad | 1706404 |
| Amersham ECL Western Blotting Detection Reagent | GE Healthcare | RPN2106 |
| SuperSignal West Femto substrate | ThermoFisher Scientific | 34095 |
| 6-Ckine (CCL21) ELISA kit | ThermoFisher Scientific | EMCCL21A |
| High-binding 96-well ELISA plates | SARSTEDT | 82.1581.200 |
| Carbonate buffer pH 10 | Thermo Scientific | 258605000 |
| TMB ELISA substrate | SERVA Electrophoresis | 37068.01 |
| Stop reagent for TMB substrate | Sigma | S5814 |
| PolyHEMA | Sigma | 192066 |
| B27 supplement | ThermoFisher | 17504044 |
| EGF | Biotechne | 236-EG |
| DAPI | Sigma | D9542 |
| Ascorbic acid | Sigma | APO456787373 |
| hTNC-specific nanobody Nb3 | Dhaouadi et al, 2021 | N/A |
| Dynabeads | Invitrogen | 11035 |
| Accutase | Gibco | A111-5-1 |

| Reagent/resource | Reference or source | Identifier or catalog number |
|---|---|---|
| DNase I (CDM) | Roche | 4716728001 |
| Eukitt mounting medium | EMS/Sigma | EMS Cat#15320; Eukitt solution |
| **Software** | | |
| FlowJo | BD | FlowJo |
| ImageJ | NIH | ImageJ |
| Axiovision | Zeiss | Axiovision |
| SpectroFlo 3.1 | Cytek Biosciences | SpectroFlo v3.1 |
| FastQC | Babraham Institute | FastQC |
| STAR aligner | Dobin et al | STAR |
| Bowtie2 | Langmead & Salzberg | Bowtie2 |
| HTseq-count | HTSeq Python package | HTSeq |
| DESeq2 | Bioconductor | DESeq2 |
| PANTHER (v11) | PANTHER | PANTHER v11 |
| REACTOME | REACTOME | REACTOME |
| MaxQuant (v1.6.14.0) | Max Planck Institute | MaxQuant v1.6.14.0 |
| Andromeda search engine | MaxQuant package | Andromeda |
| Prostar (v1.18.6) | Wieczorek et al, 2017 | Prostar v1.18.6 |
| R (v4.4.1) | R Foundation | R 4.4.1 |
| GSVA package | Bioconductor | GSVA |
| survival package | R/Bioconductor | survival |
| survminer package | R | survminer |
| GraphPad Prism (v9.1.1) | GraphPad Software | Prism 9.1.1 |
| IBM SPSS Statistics (v30.0) | IBM | SPSS 30.0 |
| **Other** | | |
| X-ray irradiator | Siemens | Primus, 6 MeV |
| Gallios flow cytometer | Beckman Coulter | Gallios |
| MultiSkan EX plate reader | ThermoFisher | MultiSkan EX |
| Varioskan Lux microplate reader | ThermoFisher Scientific | Varioskan Lux |
| Zeiss Axio Imager Z2 microscope | Zeiss | Axio Imager Z2 |
| AxioCam MRm camera | Zeiss | AxioCam MRm |
| Philips/FEI XL 30 FESEM | Philips/FEI | XL 30 |
| Philips EM-410 electron microscope | Philips | EM-410 |
| Leica ultramicrotome | Leica Biosystems | Ultramicrotome |
| AFM instrument ART-1 | ARTIDIS AG | ART-1 |
| Rectangular cantilevers | MikroMasch | HQ:CSC38 B |
| TPP culture dishes (34 mm) | TPP Techno Plastic Products AG | TPP 93040 |
| Lab-Tek coverslips | ThermoScientific NUNC | 154534 |

| Reagent/resource | Reference or source | Identifier or catalog number |
|---|---|---|
| Ultra-low attachment plates | Various | Ultra-low attachment plates for spheroids |
| Polycarbonate transwells, 5 μm | Corning Costar | 3421 |
| SpectroFlo QC beads | Cytek Biosciences | QC beads |
| NextSeq500 system | Illumina | NextSeq 500/550 High Output Kit v2 |
| Q-Exactive Plus Orbitrap mass spectrometer | Thermo Fisher Scientific | Q-Exactive Plus |
| nanoAcquity UPLC system | Waters | nanoAcquity |
| ACQUITY UPLC Peptide BEH C18 column | Waters | Peptide BEH C18 |
| Symmetry C18 trap column | Waters | Symmetry C18 |

## The 4NQO model and irradiation treatment of tumor-bearing mice

Sex as a biological variable was not considered, as mice of both sexes were used, and no differences between males and females were noticed. 4-NQO (Sigma-Aldrich, catalog number N8141) was administered to 8-week-old WT and TNCKO mice, which had been backcrossed with C57BL/6J mice (Charles River) for over 10 generations as previously described (Spenlé et al, 2020). The compound was delivered through drinking water at a final concentration of 100 μg/ml for 16 weeks (stock solution: 5 mg/ml in propylene glycol) (Spenlé et al, 2020). Following this treatment, the mice received a single 2 Gy dose of photon irradiation under anesthesia (isoflurane 2%, Minerve system). Irradiation was delivered using X-rays (Primus, 6 MeV, Siemens) at a dose rate of 200 UM/min. During the procedure, the mice having Ocrygel (Zooplus) protection on the eyes were positioned on a heated mat and beneath a lead shield (Xraystore), with only the tumor region (head) being exposed to the radiation. After irradiation, the animals were provided with regular water for 6 weeks prior to sacrifice. At the time of euthanasia, tongue and mandibular lymph nodes (TdLNs) were harvested for Fluorescence-Activated Cell Sorting (FACS) analysis, cryosectioning, and mRNA or protein extraction, as described below. Tumor incidence and organ appearance were evaluated during tissue collection. All procedures adhered to the ethical guidelines of INSERM and the ethical committee (Cremeas) and the French Ministries of Research and Agriculture (APAFIS#16463), complying with Directive 2010/63/EU for the protection of animals used in scientific research. Mice were housed in a conventional animal facility under a 12 h light/12 h dark cycle, at controlled temperature ($22 \pm 2$ °C) and humidity (50–60%), with ad libitum access to food and water. Environmental enrichment was provided in the form of cotton nesting material, tunnels, and shelters. All animal protocols comply with the ARRIVE guidelines.

## Orthotopic grafting of OSCC13 and FRC cells in the neck of C57BL/6 J mice

WT mice (male, 8 weeks of age) were grafted on the back of the neck with $1.5 \times 10^6$ OSCC13 cells also derived from a male mouse

(Spenlé et al, 2021) alone or in combination with $0.3 \times 10^6$ WT FRCs in 150 μL PBS using an U-100 insulin syringe (BD Micro-Fine) with an endpoint of 15 weeks post engraftment. In a second series of engraftment $3 \times 10^6$ OSCC13 cells were grafted alone or together with $0.6 \times 10^6$ FRCs (WT or TNC KO) with an endpoint of 41 days. Tumors were detectable by palpation and tumor size measurement with a caliper was started 1 or 2 weeks upon engraftment. Mice were sacrificed for analysis when the ethical weight limit was reached, when the tumor length passed 1.5 cm or at the final date of the protocol. At the time of euthanasia, tumors were collected for cryosectioning and mRNA extraction, as described below.

## Isolation of lymph node fibroblast reticular cells (FRC)

FRCs were isolated from the lymph nodes of male WT and TNCKO mice as previously published (Fletcher et al, 2015; Spenlé et al, 2020). Cervical, mandibular, brachial, axillary, popliteal, and inguinal lymph nodes from two WT and two TNCKO (10 weeks old) mice were dissected and placed in 5 mL of Roswell Park Memorial Institute Medium (RPMI-1640) (per isolate) on ice. After all lymph nodes were collected, RPMI-1640 was removed and replaced with 2 mL of freshly made enzyme mix composed of RPMI-1640 containing 0.8 mg/mL dispase (Roche), and 0.2 mg/mL collagenase P (Roche) and 0.1 mg/mL DNase I (Invitrogen). Tubes were incubated at 37 °C and 200 rpm shaking for 30 min. Under sterile condition, lymph nodes were very gently aspirated and expirated using a 1 mL pipette. The mixture was placed again at 37 °C and 200 rpm shaking for 10 min, and then centrifuged ($300 \times g$, 4 min, 4 °C). The supernatant (containing the enzyme mixture) was removed, and cell pellets were resuspended in ice-cold fresh isolation buffer (2% fetal bovine serum (FBS), 5 mM EDTA in sterile PBS). The resulting cell suspensions were filtered through a 70-μm and 40-μm cell strainer and then placed in 6 cm culture dishes containing complete Dulbecco's Modified Eagle's Medium (DMEM) (10% FBS, 1% PenStrep) for 24 h. The day after, floating cells corresponding to dead cells and leukocytes were removed by replacing the culture medium with fresh complete DMEM. The clearance of endothelial cells in the culture dish occurred over the following two weeks with culture of the cells in DMEM supplemented medium. The exclusive presence of fibroblasts was verified by immunofluorescent staining of the cells in LabTek slides as described below for podoplanin (gp38), collagen VI (ERTR7), and CD31.

## Hematoxylin–eosin staining (HE)

Tissue sections (8 μm thick) embedded in OCT were first incubated in ddH$_2$O and then stained with hematoxylin (Surgipath) and eosin (Harris) following a standard protocol (Cardiff et al, 2014). After staining, sections were mounted using the Eukitt solution (Sigma).

## Immunohistochemistry on OCT sections

Cryosections were rehydrated in 1× PBS for 10 min after encircling the tissue with a DakoPen. Endogenous peroxidase activity was blocked using 0.5% H$_2$O$_2$ in methanol (freshly prepared) for 30 min at room temperature (RT), followed by a 5-min wash in distilled water. Antigen retrieval was performed with ice-cold acetone for

30 min, then slides were equilibrated in distilled water and permeabilized with 0.1% Triton X-100 in PBS for 10 min. Sections were blocked with 5% normal goat serum (NGS) in PBS for 1 h at RT, followed by overnight incubation at 4 °C with anti-tenascin-C (TNC) antibody (Millipore, 1:100 in 5% NGS-PBS). The next day, slides were washed in PBS and incubated with a biotinylated secondary antibody (PK-4001, Vector Laboratories) for 1 h at RT. An avidin-biotin complex (ABC) solution was applied for 1 h, followed by DAB substrate (SK-4100, Vector Laboratories) for up to 10 min. Slides were then washed, counterstained with hematoxylin (30 s), rinsed, and dehydrated through graded ethanol and toluene. Coverslips were mounted with Eukitt (EMS, Cat#15320) and dried overnight.

## Flow cytometry

Tongue tumors and cervical lymph nodes were finely chopped with a scalpel and digested in a solution containing 1 mg/mL collagenase D (Roche, catalog number 50-100-3282), 0.2 mg/mL DNase I (Roche, catalog number 4716728001), and 2% heat-inactivated FBS in RPMI medium. The digestion was performed at 37 °C for 2 h. After digestion, 92 μL of 54 mM EDTA was added, and the samples were vortexed at maximum speed for 30 s. Cell suspensions were filtered sequentially through 70-μm and 40-μm cell strainers and treated with flow cytometry buffer (PBS, 2% FBS, 1 mM EDTA). Cell counts were performed, and aliquots of $2 \times 10^6$ cells per lymph node sample or $1 \times 10^6$ cells per tumor sample were stained with Dead Viability Dye-eFluor 450 (Thermo Fisher, catalog number 65-0863-18) following the manufacturer's instructions. The cells were then incubated in a blocking solution containing 2% FcBlock CD16/CD32 (Thermo Fisher, catalog number 14-0161-85) in flow cytometry buffer for 15 min at 4 °C, followed by a 30-min incubation at 4 °C with a standard immunophenotyping antibody panel with solution 1: anti-CD45-FITC, anti-CD11c-PE, anti-B220-APC, anti-CD80-APC EF780, anti-CD86- PE Cy7 and anti-CCR7-Percp Cy5; solution 2: anti-CD45-FITC, anti-CD3e-PE, anti-C8a-APC, anti-CD4-APC EF780, anti-Foxp3-PE Cy7, anti-CCR7-Percp Cy5 and anti-CD25-AF700; solution 3: anti-CD45-FITC, anti-Gp38-PE, anti-CD31-APC, anti-F4/80-APC EF780, anti-CCR7-Percp Cy5 and anti-CD11b-AF700 (Reagents and Tools table). Data acquisition was performed using a Beckman Coulter Gallios flow cytometer, and subsequent adjustments and analyses were carried out with the FlowJo software.

## RNA extraction and real-time quantitative PCR (qRTPCR)

Frozen tongue and neck tumors and cultured cells were lysed in TRIzol reagent (Invitrogen, catalog number 12044977) to extract total RNA. RNA quality was verified by measuring optical density at 260 nm (OD 260 nm). Complementary DNA (cDNA) was synthesized from 1000 μg of total RNA using random primers and Moloney murine leukemia virus reverse transcriptase (MultiScribe, Applied Biosystems, catalog number 10117254). The cDNA was then subjected to qRTPCR using an Mx3005P Real-Time PCR System (ThermoFisher Scientific). Reactions were performed in duplicate for all conditions using either Sybr Green Master Mix (ThermoFisher Scientific, catalog number 4344463) or Fast Taq-Man Mix (ThermoFisher Scientific, catalog number 4444557).

Expression of Gapdh and Rpl19 mRNA (Life Technology, catalog number 433764T) was used as endogenous controls, and the comparative cycle threshold method ($2^{-\Delta\Delta Ct}$) was employed for analysis. Primer sequences are detailed in the Reagents and Tools Table.

## Analysis of protein expression

Tissues or cell lysates were prepared in a lysis buffer containing 50 mM Tris-HCl (pH 7.6), 150 mM NaCl, 1% NP-40, 0.5% sodium deoxycholate, and 0.1% SDS, supplemented with a phosphatase inhibitor cocktail (Santa Cruz, catalog number sc-45045) and protease inhibitors (Roche, catalog number 05892970001). Protein concentration from tissue samples and conditioned media was measured using the Bradford assay (Bio-Rad, catalog number 5000001) following the manufacturer's protocol.

For protein separation, 30 μg of lysate was mixed with Laemmli buffer (Bio-Rad, catalog number 1610737) and loaded onto pre-casted 4–20% gradient gels (Bio-Rad, catalog number 4561096) for SDS-PAGE. Proteins were then transferred to nitrocellulose membranes (Bio-Rad, catalog number 1620113) using the Trans-Blot Turbo Transfer system (Bio-Rad). Membranes were blocked with 5% Blocking-Grade Blocker (Bio-Rad, catalog number 1706404) in PBS with 0.1% Tween20 and incubated with primary antibodies overnight at 4 °C, followed by secondary antibodies for 1 h at room temperature in 1.5% Blocking-Grade Blocker in 0.05% Tween20 PBS. Antibodies used are detailed in the Reagents and Tools table. Protein bands were visualized using Amersham ECL Western Blotting Detection Reagent (GE Healthcare, catalog number RPN2106) or SuperSignal West Femto Substrate (Thermo-Fisher, catalog number 34095). Expression levels of CCL21 were quantified using the 6-Ckine ELISA kit (ThermoFisher Scientific, catalog number EMCCL21A), following the manufacturer's instructions. Absorbance readings for samples and standards were obtained using a MultiSkan EX plate reader (ThermoFisher).

## Cell culture, coculture, and exposure to irradiation

FRCs were cultured in DMEM high-glucose (Dutscher, L0102-500) supplemented with Fetal bovine serum (FBS, Dutcher) 10%, 100 U/mL penicillin, 100 μg/mL streptomycin (PenStrep, Dutscher), 40 U/mL Gentamicin (ThermoFisher Scientific). TNC silencing in FRCs was achieved using short hairpin RNA (shRNA)-mediated gene knockdown. Lentiviral particles carrying shRNA vectors targeting TNC (shTNC: CCGGGCATCAACACAACCAGTCTAACTCGAGTTAGACTGGTTGTGTTGATGCTTTTTG) were obtained from Sigma and used for transduction. Lentiviral particles encoding a non-targeting shRNA vector (SHC202V, Sigma) served as a control. Transduced cells were selected using the standard culture medium supplemented with 10 μg/mL puromycin (ThermoFisher, J67236). The selection pressure was maintained throughout all in vitro experiments by the addition of puromycin. The OSCC13 cell line, derived from a primary 4NQO-induced tongue tumor of a male WT mouse (Spenlé et al, 2021), was cultured in DMEM-F12 containing 4.5 g/L glucose, FBS 10%, Penstrep, gentamicin, and 50 μg/mL hydrocortisone (Sigma). Cell identity was verified by marker expression and morphology. DC-like DC2.4 cells were obtained from Merck (SCC142) and maintained in DMEM high-glucose complemented with 10% FBS, PenStrep, Gentamicin, and

1× HEPES. DC2.4 cells were authenticated by the supplier. The human HNSCC CAL33 cells were established in the Antoine Lacassagne Cancer Centre (Bozec et al, 2006) and maintained in DMEM high glucose complemented with 10% FBS and PenStrep. Normal human skin fibroblasts (TIFs) immortalized with the telomerase reverse transcriptase (hTERT) gene were provided by J. Norman (Beatson Institute, Glasgow, UK) and maintained in DMEM high glucose, complemented with 20% FBS, 20 mM HEPES, and PenStrep, as previously reported (Gopal et al, 2017). Cells were routinely tested for mycoplasma (Venor GeM OneStep kit by Minerva Biolabs). All cell lines were cultured at 37 °C in a humidified atmosphere with 5% $CO_2$. Culture media were replenished every 2–3 days, and cells were subcultured at 80–90% confluency using trypsin-EDTA (PanBiotech, P10-023100). Prior to treatment or irradiation, cells were serum-starved overnight in medium containing 1% FBS. Cells received a single 2 Gy dose of photon irradiation. Cells were treated with 10 ng/mL recombinant mouse TGFβ1 (Biotechne, 7666-MB) for 24 h and 10 µM of TGFβR inhibitor (GW788388, Selleckchem) for 1 h or 24 h. Upon starvation, the conditioned medium (CM) of OSCC13 cells was collected, filtered at 0.22 µm, and stored at −80 °C for future use. FRCs were treated 24 h with filtered CM from OSCC13 and with purified mouse TNC (10 µg/mL) diluted with DMEM medium complemented with 1% FBS, penstrep, and gentamicin. Purified strep-tagged mouse TNC was obtained as previously described (Spenlé et al, 2020). For the establishment of the FACS/MACS sorting protocol, OSCC13 and FRCs were cocultivated at a ratio of 1:1 for 24 h in full medium. Cells were detached mechanically, concentrated by centrifugation, and separated by FACS sorting using the 5 Laser Aurora Full Spectrum Cell sorter (Cytek Biosciences) and the following antibody panel: anti-Gp38-PE, anti-CD29-PE Cy7, anti-ITGA7-APC, anti-FAP-PE CF594, anti-140b-APC EF780. Data were acquired with Sectroflo 3.1. Quality control measures were performed using SpectroFlo QC beads (Cytek Biosciences) prior to acquiring samples. For the irradiation coculture experiments, OSCC13 and FRC cells (shC, shTNC, or TNCKO) were cocultured in a 2:1 ratio in medium containing 50% OSCC13 medium without hydrocortisone and 50% FRC medium without puromycin, for 48 h. Then the cells were irradiated with 2 Gy or 10 Gy, and 48 h later the cells were collected and separated by the magnetic cell sorting (MACS) method.

## Magnetic cell sorting (MACS)

Dynabeads (Invitrogen #11035) at 25 µl per condition were washed two times in isolation buffer (PBS, 0.1% BSA, 2 mM EDTA, pH 7.4). After the washes, the beads were resuspended in isolation buffer at the same initial volume of the Dynabeads and coated with 1.5 µg of antibodies against integrin α7β1 and GP38 per 25 µl of beads (Reagents and Tools table). The coated beads were incubated for 1 h at 4 °C with tilting and rotation. The cocultured cells were harvested using Accutase (GIBCO #A111-5-1), and the cell pellets were resuspended in 1 mL isolation buffer. The coated beads were washed three times with isolation buffer and after the final wash the cells were resuspended in 1 mL isolation buffer/25 µl initial volume. In all, 1 mL of beads was added to the cell suspensions and they were incubated for 2 h at 4 °C with tilting and rotation. After placing the tubes in the magnet for 5 min, the supernatant was collected, containing the OSCC13 cells (negative for integrin α7β1

and GP38). The remaining bead-bound cells were washed two times with isolation buffer and then collected in 2 mL isolation buffer. The cells were then pelleted and resuspended in Trizol reagent for RNA isolation.

## 3D (spheroid) cocultures and peptide treatments

OSCC13 and FRCs were cocultured at a 2:1 ratio and CAL33 with TIF at a 1:1 ratio in DMEM/F12 supplemented with B27 and EGF (20 ng/mL) for 7 days in ultra-low attachment plates pre-coated with polyHEMA (poly(2-hydroxyethyl methacrylate, Sigma #192066), as reported in (Shaw et al, 2012). PolyHEMA was prepared by dissolving 2 g of powder in 166 mL of 95% ethanol, followed by filtration. Plates were coated with the solution and dried in an incubator at 37 °C for 48 h prior to use. On day 7, spheroids were irradiated with 10 Gy. The following day, spheroids were treated with either MP5 peptide (50 µg/mL) or Cy5-MP5 (50 µg/mL) for 72 h. After treatment, MP5-treated spheroids were mechanically dissociated with a syringe for cell counting, while Cy5-MP5-treated spheroids were fixed in 4% paraformaldehyde for whole-mount immunofluorescence analysis. For the fractionated irradiation experiments, the OSCC/FRC spheroids were subjected to 5 consecutive days of 2 Gy irradiation, before fixation and immunofluorescence analysis.

## Competitive ELISA

High-binding 96-well ELISA plates (SARSTEDT, Cat. No. 82.1581.200) were coated overnight at 4 °C, with 1 µg/ml of recombinant His-tagged TNC FNIII 4-5 (TN4-5) molecule (Loustau et al, 2022) in carbonate buffer pH 10 (Cat No: 258605000, Thermo Scientific). The non-coated surface was blocked with 5% skim milk in PBS for 1 h at 37 °C. Peptides MP5 or Cy5-MP5 at different concentrations (0–250 µg/mL) were added to the TN4-5 coated plate, which was immediately followed by the addition of the purified recombinant strep-tagged fibronectin FNIII 4-6 (FN4-6) molecule at 0.5 µg/ml in PBS/1%BSA/0.3% Tween20 and, incubated at 37 °C for 2 h. After 4 times washing with 1× PBS/ 0.05% Tween20 100 µl of an anti-strep-tag antibody (Abcam, ab184224) in a dilution of 1:2,000 (PBS/1%BSA/0.3% Tween20) was added to each well for 2 h at 37 °C. Wells were washed 4 times with 1× PBS/0.05% Tween20, and horseradish peroxidase (HRP)-conjugated secondary antibody (anti-mouse, 1:5000 dilution in PBS/1%BSA/0.3% Tween20) was added. After incubation for 1 h at 37 °C and washing in 1× PBS/0.05% Tween20, the bound HRP conjugate was detected by adding 100 µl of TMB ready-to-use ELISA substrate (SERVA electrophoresis, Ref 37068.01). The peroxidase reaction was stopped after 5 min by the addition of 50 µl stop solution 50 µl/well (Sigma, Stop reagent for TMB substrate, Ref S5814). Optical densities were measured at 450 nm using a microplate reader (Varioskan Lux Microplate reader, Thermo Fisher Scientific).

## Chemoretention assay

Boyden chamber migration assays with DC2.4 cells were performed in 5 µm pore-sized polycarbonate membrane transwells (Corning Costar Co, catalog number 3421). The lower surface of the transwells were coated with purified TN1-5 (dissolved in 0.01%

Tween20 PBS at a final concentration of $1 \mu g/cm^2$) and peptides MP5 or Cy5-MP5 ($25 \mu M$ in PBS) were added and incubated overnight at 4 °C. The bottom chambers of the transwells were filled with 1% FBS RPMI containing mouse CCL21 (200 ng/ml 366-6C-025/CF, R&D Systems). Murine DC2.4 dendritic cells (Merck, SCC142) ($5 \times 10^4$) cells in 1% FBS RPMI were placed on the top of the transwell chamber for 24 h at 37 °C in 5% $CO_2$. Cells on the lower side of the insert were fixed with 4% paraformaldehyde (PFA) and stained with 4′,6-Diamidino-2-phenylindole (DAPI) before cell counting. Images were captured and analyzed using ImageJ software. Between 15 and 20 randomly taken images were analyzed per well.

## Generation of cell-derived matrix (CDM)

CDM of FRCs was done as described for other cells (Rupp et al, 2016). Briefly, Lab-tek coverslips (ThermoScientific NUNC #154534), six-well or 24-well plates were coated with 1% gelatin for 60 min at 37 °C, followed by crosslinking with 1% glutaraldehyde for 20 min at room temperature. Next, the crosslinker was quenched with 1 M glycine for 30 min and gelatin-coated surfaces were incubated with DMEM complete medium before seeding 25,000 FRCs per $cm^2$. On day 2 and during the next 9 days, medium was changed every second day and replaced by medium supplemented with $50 \mu g/mL$ of ascorbic acid. On day 10, matrices were decellularized in 20 mM NH4OH, 0.5% Triton X-100 in PBS followed by $10 \mu M$ DNase I treatment (Roche, 4716728001).

## Modified chemoretention assay on CDM

Boyden chamber assays with DC2.4 cells were performed using 5-μm pore-sized polycarbonate membrane transwells (Corning Costar Co, 3421). The surface of the lower chamber was pre-coated with CDM deposited by WT and TNCKO FRCs over 10 days as described above. Then the lower chamber was filled with DMEM containing mouse CCL21 (200 ng/ml, R&D Systems, 457-6C-025) either alone or supplemented with 500 nM of hTNC-specific nanobody Nb3, as previously described (Dhaouadi et al, 2021). To block DC2.4 chemotaxis toward CCL21, cells were pre-incubated for 6 h with a CCR7 neutralizing antibody ($10 \mu g/mL$, R&D Systems, MAB3477) diluted in 1% FBS complemented DMEM. DC2.4 (50,000 cells) suspended in $150 \mu l$ of 1% FBS complemented DMEM were placed into the upper chamber of the transwell system. After 5 h of migration, the surface of the lower chambers was fixed with 4% PFA, and cells were stained with DAPI and ECM for the indicated molecules. Images were captured and analyzed using ImageJ software. Between 12 and 16, randomly taken images were analyzed per well.

## Immunofluorescence (IF)

Frozen tissue sections (8 μm, unfixed), CDM, or cells fixed with 4% paraformaldehyde were incubated at room temperature for 1 h with blocking serum (5% normal goat or donkey serum in PBS; Jackson ImmunoResearch, 005-000-121 and 017-000-121, respectively). Primary antibodies (detailed in the Reagents and Tools table) were then applied overnight. Bound antibodies were detected using secondary antibodies conjugated to Alexa 488, Cy3, or Cy5 specific for goat, rabbit, guinea pig, hamster, or rat primary antibodies.

DAPI (Sigma, D9542) was used for nuclear staining. Sections were embedded in FluorSave Reagent (Calbiochem, 345789) and analyzed using a Zeiss Axio Imager Z2 microscope. Images were captured with an AxioCam MRm camera (Zeiss) and Axiovision software. Control sections were processed identically, except for the omission of the primary antibodies. Image acquisition settings (microscope, magnification, light intensity, and exposure time) were maintained constant within and across experiments. Immune cell quantification and positive Ki67 cells or P-Smad3 nuclei were analyzed using ImageJ software. At least three sections from four different tumors or mice were analyzed per condition. The number of immune cells was normalized to the total number of DAPI-positive nuclei.

## Scanning electron microscopy

The cell ultrastructure of both non-irradiated and irradiated cell culture samples was analyzed by high-resolution SEM. Cell samples were treated as described above, then fixed with 2.5% glutaraldehyde in cacodylate buffer. Then cells were washed with cacodylate buffer and dehydrated with an ascending ethanol series. Carbon dioxide was used for critical point drying of the specimens, and absolute ethanol was used as an intermediate solvent. Samples were mounted on aluminum holders, sputtered with 30 nm palladium/gold, and examined in a Philips/FEI XL 30 field emission scanning electron microscope (FESEM) operated at 5 kV accelerating voltage.

## Electron microscopy of tumor tissue

Frozen and cryopreserved tissue samples were first thawed and washed with distilled water. Tissue was then fixed in a solution containing 2% formaldehyde (v/v) and 0.25% glutaraldehyde (v/v) in 100 mM cacodylate buffer, pH 7.4, at 4 °C overnight. Following fixation, the samples were rinsed with PBS, dehydrated in ethanol up to 70%, and embedded in LR White embedding medium (London Resin Company, UK), and polymerized under UV light. Ultrathin sections were prepared using an ultramicrotome (Leica Biosystems), collected on copper grids, and stained negatively with 2% uranyl acetate (Sigma-Aldrich) for 15 min. Electron micrographs were captured at 60 kV using a Phillips EM-410 electron microscope and imaging plates (Ditabis, Pforzheim, Germany).

## Atomic force microscopy

Frozen microscopy slides with 14 μm OSCC tumor tissue sections were prepared by using a glass cutter to fit into a 34-mm culture dish (TPP Techno Plastic Products AG, Trasadingen, Switzerland, TPP 93040). The selected sample slides were then affixed to the bottom of the dish using a two-component epoxy adhesive. The dish was filled with degassed and filtered PBS (pH 7.4, Gibco), and left in Custodiol (Dr. Franz Köhler Chemie GmbH, Bensheim, Germany) for 30 min at room temperature to thaw. Samples were gently washed to remove non-adherent cells and debris and refilled with degassed and filtered Custodiol. Rectangular Cantilevers (HQ:CSC38 B, MikroMasch) were used to map the stiffness of the OSCC sections. Each cantilever's spring constant was determined using the thermal tuning method and was $k = 0.05–0.1 \, Nm^{-1}$ (Sader et al, 1995). Deflection sensitivity was

measured in degassed and filtered Custodiol on an empty TPP culture dish to ensure accurate force measurements during AFM analysis (Plodinec et al, 2012). Measurements were performed using the ART-1 machine (ARTIDIS AG, Switzerland). OSCC sample dishes were mounted on the sample tray, and force-displacement curves were recorded to assess cell stiffness (elastic modulus, E) and cell-surface adhesion. A total of 30–40 maps were acquired, each containing 400 force curves within a $20 \times 20\ \mu m$ area, using a relative trigger force of 1.8 nN and an indentation speed of 16 µm/s. Maps were selected based on the device's built-in optical microscopy and compared with previously recorded confocal microscopy images of adjacent frozen sections to ensure accurate targeting of tumor regions. Forward and backward elastic moduli were calculated from force-displacement curves using the Oliver-Pharr model (Oliver and Pharr, 1992). Maps with more than 30% irregular force curves were excluded from the analysis. Force curves influenced by the underlying coverslip were distinguishable from those generated by the sample. Maps with over 30% of these stiff force curves were also excluded. For the remaining maps, a Gaussian function was fitted to determine the distribution values (mean forward elastic modulus [kPa] ± standard deviation). Overall means and standard deviations were then reported, summarizing all maps.

## Gene expression analysis

RNA from WT and TNCKO FRCs (derived from two mice per group) or from the NIR or IR TdLNs of the 4NQO WT and TNC KO mice was extracted (in duplicate from different passages) using the RNeasy Mini Kit (Qiagen), and RNA integrity was assessed on an Agilent Bioanalyzer 2100 (Pico Kit, Agilent Technologies). RNA Sequencing libraries were constructed using the SMARTer® Stranded Total RNA-Seq Kit v2 - Pico Input Mammalian (TaKaRa) following the manufacturer's protocol. Libraries were pooled and sequenced in paired-end mode (2×75 bp) on a NextSeq500 system with the NextSeq 500/550 High Output Kit v2 (Illumina). Quality control for each sample was performed using the FastQC tool from the NGS Core Tools suite. Sequence reads were aligned to the reference genome using STAR and Bowtie2 aligners, with results output in BAM (Binary Alignment Map) format for subsequent raw read count extraction. Read counts were determined with HTseq-count, part of the HTSeq Python package, using default parameters to generate an abundance matrix. Differential expression analysis was conducted using the DESeq2 package within the Bioconductor framework. Genes were considered significantly upregulated or downregulated based on an adjusted P value threshold of <0.10 and a fold-change cutoff of >±0.8 (detailed in Appendix Table S2). Further functional analysis of deregulated genes was performed using the PANTHER version 11 and REACTOME software tools. To assess which FRC subclusters were represented in the model established in this study, the corresponding gene signatures of each cluster were used to evaluate enrichment through gene set variation analysis (GSVA). Specifically, based on the droplet-based single-cell RNA sequencing by (Rodda et al, 2018), there are nine peripheral stromal cell (SC) clusters in the lymph node, several of which were classified as FRC subclusters. Included are Ccl19 T-zone reticular cells (TRCs), marginal reticular cells, follicular dendritic cells (FDCs), and perivascular cells. The expression level of each gene set

was normalized by z-score, and the Wilcox test was used to estimate the difference between the WT and TNC KO groups. Differential expression analysis of the TdLNs was performed on 12 samples using the limma package. Differentially expressed genes (DEGs) between non-irradiated TNCKO and WT samples were identified using a threshold of |log2FC| = 2.5 and an adjusted P value = 0.05. Heatmaps and volcano plots of the DEGs were generated using the ggplot2 and ComplexHeatmap packages, respectively. Subsequently, the most significant enrichment pathways and biological processes of DEGs were investigated using the Gene Ontology (GO) analyses using the R software "clusterProfiler" package. See Appendix Table S1 for details.

Total RNA from MP5 and control NT193M cells, as well as from NT193 shC and shTNC cells, was extracted following the same procedure used for qRT-PCR and subsequently processed for RNA-seq, as previously described (Li et al, 2024 Murdamoothoo et al, 2021). Differential gene expression analyses were carried out using the DESeq2 package within the Bioconductor framework, applying internal normalization and shrinkage estimation for dispersion and fold-change calculations. Genes with nominal P values < 0.05 and false discovery rate (FDR) < 0.10 were considered significantly regulated. Principal component analysis (PCA) was performed on variance-stabilized read counts generated by DESeq2. For this analysis, we selected 7000 genes with the highest variance across all samples and detectable expression in every dataset, ensuring robust comparative clustering. The overall transcriptomic patterns referenced here correspond to datasets published in (Li et al, 2024) and Murdamoothoo et al (2021). Heatmaps illustrating gene expression patterns were generated using the Heatmapper online platform (https://heatmapper.ca/), applying default hierarchical clustering parameters.

## NanoLC-MS/MS analysis

WT and TNCKO FRCs ($n = 5$ per genotype) were extracted in Laemmli-like buffer. Protein concentration was determined using the DC protein assay (Bio-Rad) following the manufacturer's protocol. For each sample, 15 µg of protein lysate was heated at 95 °C for 5 min. Proteins were loaded onto a 5% acrylamide SDS-PAGE stacking gel prepared in-house. Gel bands were reduced, alkylated, and digested overnight at 37 °C with modified porcine trypsin (Promega, Madison, USA) at an enzyme-to-protein ratio of 1:50. Extracted peptides were analyzed by nanoLC-MS/MS using a nanoAcquity UPLC system coupled to a Q-Exactive Plus Orbitrap mass spectrometer with a Nanospray Flex™ ion source. Peptides were separated on an ACQUITY UPLC® Peptide BEH C18 analytical column and a Symmetry® C18 trap column. A total of 800 ng of peptides was loaded onto the trap column and eluted with a gradient of acetonitrile in 0.1% formic acid at 400 nL/minute. The mass spectrometer was operated in Data-Dependent Acquisition (DDA) mode with automatic switching between MS and MS/MS. The ten most abundant ions with a charge state ≥2 were selected on each MS spectrum for further isolation and higher energy collision dissociation (HCD) fragmentation, excluding unassigned and monocharged ions. The dynamic exclusion time was set to 60 s. To avoid contamination between two successive samples, a "blank" sample without peptide was injected between each sample. NanoLC-MS/MS data were interpreted to do label-free extracted

ion chromatogram-based differential analysis using MaxQuant (version 1.6.14.0, Tyanova et al, 2016). Peaks were assigned with the Andromeda search engine against a custom mouse protein database (UniProtKB-SwissProt, Taxonomy ID: 10,090; 19-10-2020). MaxQuant parameters were set as follows: Maximum number of missed cleavages set to 1, Carbamidomethyl (C) set as fixed modification, Acetyl (Protein N-term) and Oxidation (M) set as variable modifications. False discovery rates (FDR) were estimated based on the number of hits after searching a reverse database and were set to 1% for both peptide spectrum matches and proteins. Protein quantification was performed using the LFQ intensities without "Match between runs". All other MaxQuant parameters were set as default. Differential analysis of normalized LFQ intensities was conducted using Prostar software (v1.18.6) applying a Limma $t$ test (Wieczorek et al, 2017). The complete proteomics dataset has been deposited into the ProteomeXchange Consortium via the PRIDE partner repository with the dataset identifier PXD060164 (Perez-Riverol et al, 2022). Deregulated proteins (adjusted $P$ value threshold of <0.05) were further analyzed via PANTHER and REACTOME. See Appendix Table S3 for details.

## Patient survival analysis

Sex as a biological variable was not considered, as both men and women were included. Patient data were retrieved from the TCGA public database (TCGA-HNSC), and clinical data were obtained from the University of California Santa Cruz (UCSC) Xena platform (https://xenabrowser.net/) and analyzed using the R language. The cohort was stratified into "High" and "Low" expression groups based on the median expression level of the corresponding gene. Survival analysis was conducted specifically for HNSCC patients who had undergone radiotherapy (RT). Cox regression analysis was performed using the patient cohort TCGA-HNSCC. Analyses were performed using R version 4.4.1. The GSVA package was utilized to calculate the enrichment scores for each patient based on the FRC signature expression, which were then used to classify patients into high-FRC and low-FRC groups. Survival and Survminer packages were used to perform the Kaplan–Meier analysis. Cox regression analysis was conducted using IBM SPSS Statistics version 30.0. (Appendix Table S4).

## Statistical analysis

For all datasets, the Gaussian distribution was assessed using the Shapiro-Wilk normality test. If the data followed a Gaussian distribution, statistical differences were evaluated using an unpaired $t$ test (with Welch's correction for unequal variances), and one- or two-way ANOVA followed by Tukey's post-test. For non-Gaussian distributions, the Mann–Whitney test or Kruskal–Wallis with Dunn's post-test was employed to determine statistical significance. All analyses were conducted using GraphPad Prism software (version 9.1.1), and results are expressed as mean ± SEM or mean ± STDEV. Statistical significance was set at $P < 0.05$, with levels denoted as *$P < 0.05$, **$P < 0.01$, ***$P < 0.001$, ****$P < 0.0001$. The exact $P$ values are listed on Appendix Table S5. No statistical methods were used to predetermine sample size. Sample sizes were chosen based on prior experience with similar

### The paper explained

#### Problem

Ionizing radiation is the standard treatment for head and neck tumors and effectively reduces tumor burden. However, it simultaneously induces an immunosuppressive tumor microenvironment, particularly marked by increased expression of the tumor-promoting extracellular matrix glycoprotein tenascin-C (TNC). In this study, we sought to determine the specific contribution of TNC to tumor response following irradiation, using models with genetic or functional loss of TNC.

#### Results

Our work reveals that tenascin-C plays multiple regulatory roles in tumor radiosensitivity. In addition to directly influencing tumor cells, irradiation profoundly alters the phenotype of fibroblastic reticular cells (FRCs), a major source of TNC both within the tumor and in tumor-draining lymph nodes (TdLNs). Irradiation-surviving FRCs expand and acquire enhanced immunosuppressive properties, thereby promoting tumor progression. We demonstrate that physical interactions between tumor cells and FRCs drive FRC expansion and activation, increased tumor-cell survival and radioresistance, all in a TNC-dependent manner. Irradiation also remodels the TdLNs, where FRCs maintain the structural and functional networks required for adaptive immunity. In TNC-deficient tumor-bearing mice, TdLN immunity is deregulated, which likely contributes to the observed radioresistance.

#### Impact

TNC supports tumor regrowth after irradiation, in contrast to its protective role within the TdLNs, where TNC maintains immune-supportive functions. Our findings indicate that the balance between these opposing roles critically shapes radiotherapy outcomes. This identifies both FRCs and TNC as promising targets to enhance radiotherapeutic efficacy. Importantly, a high intratumoral abundance of highly TNC-expressing FRCs correlates with reduced survival in irradiated head and neck cancer patients. Finally, we demonstrate that therapeutic targeting of TNC using the MAREMO peptide increases tumor radiosensitivity, underscoring its clinical potential.

experimental models and are reported in the figure legends. Animals and samples were allocated to experimental groups based on genotype and treatment; no additional randomization was applied. Investigators were blinded to group allocation during experiments or outcome assessment. No data were excluded from the analyses unless predefined technical issues occurred.

## Data availability

Raw RNA-seq data are deposited at the EMBL-EBI ArrayExrpress archive (accession no E-MTAB-14801 and E-MTAB-16360. The complete proteomics dataset has been deposited into the ProteomeXchange Consortium via the PRIDE partner repository with the dataset identifier PXD060164. All other data are available in the main text or the supplementary information. The engineered cells are available upon request.

The source data of this paper are collected in the following database record: biostudies:S-SCDT-10_1038-S44321-026-00406-8.

## Peer review information

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

## Acknowledgements

We like to thank the staff of the animal facility and the staff of the GENOMAX facility for their assistance. Work in this article is supported by Worldwide Cancer Research (WCR) 24-0331/Association contre le cancer (ARC) (GN/HB and GO), Aviesan ITMO Cancer project Radio-3R (GN/HB and GO), INCa FITMANET (GN/HB and GO), and funding to GO: INSERM, INCa/Ligue contre le cancer LNCC ECMPACT, Ligue Régional contre le cancer Grand Est, and INCa PLBIO TENMAX.

## Author contributions

**Thomas Loustau**: Conceptualization; Data curation; Formal analysis; Supervision; Investigation; Methodology; Writing—original draft; Writing—review and editing. **Ioanna Mitrentsi**: Conceptualization; Data curation; Formal analysis; Supervision; Investigation; Visualization; Methodology; Writing—original draft; Writing—review and editing. **Nuohan Wang**: Formal analysis; Investigation; Methodology. **Caroline Spenlé**: Conceptualization; Data curation; Formal analysis; Investigation; Visualization; Methodology. **Alexia Pavlidaki**: Data curation; Writing—review and editing. **Thibaud Tranchant**: Data curation; Formal analysis; Investigation; Methodology. **Gilles Riegel**: Formal analysis; Investigation; Methodology. **Akhil Venu**: Formal analysis; Investigation. **Rime Oueidat**: Investigation; Methodology. **Manuel Koch**: Resources. **Marion Dumas**: Formal analysis; Investigation; Methodology. **Fanny Wack**: Investigation. **Aurelie Hirschler**: Formal analysis; Investigation; Methodology. **Christine**

Carapito: Formal analysis; Supervision; Investigation. **Nicodème Paul**: Data curation; Formal analysis; Investigation; Methodology. **Raphael Carapito**: Supervision; Investigation; Methodology. **Matthias Mörgelin**: Formal analysis; Investigation; Methodology. **Uwe Hansen**: Formal analysis; Investigation. **Joyce Azzi**: Investigation; Methodology. **Lucie Aubergeon**: Investigation; Methodology. **Nathalie Salomé**: Supervision. **Sayda Dhaouadi**: Resources. **Pierre Grenot**: Investigation; Methodology. **Balkiss Bouhaouala-Zahar**: Resources; Supervision. **Simona La Cioppa**: Formal analysis; Investigation; Methodology. **Philipp Oertle**: Formal analysis; Supervision; Investigation; Methodology. **Valerio Izzi**: Data curation; Supervision. **Marija Plodinec**: Formal analysis; Supervision; Investigation; Methodology. **Georges Noel**: Supervision; Methodology. **Hélène Burckel**: Supervision; Methodology. **Gertraud Orend**: Conceptualization; Supervision; Funding acquisition; Writing—original draft; Project administration; Writing—review and editing.

Source data underlying figure panels in this paper may have individual authorship assigned. Where available, figure panel/source data authorship is listed in the following database record: biostudies:S-SCDT-10_1038-S44321-026-00406-8.

## Disclosure and competing interests statement

Matthias Mörgelin is employed by Colzyx. Simona La Cioppa and Philippe Oertle are employed by Artidis. The remaining authors declare no competing interests.

# Expanded View Figures

**Figure EV1. Characterization of the 4NQO tumors and TdLNs after irradiation in WT and TNCKO mice.**

(**A**) Cartoon summarizing the 4NQO protocol combined with a single dose of 2 Gy. (**B**) Quantification of tongue tumor number in NIR and IR WT and TNCKO mice, $N = 7$–10 mice per group. (**C, D**) Representative IF images (of at least 25) for the indicated molecules in IR 4NQO WT and TNCKO tumors (**C**) and NIR and IR WT and TNCKO tumors (**D**). White arrows indicate the CD11c+ cells either trapped in the stroma (**C**) or reaching the tumor nests (**D**). Scale bar, 50 μm (**C**), 200 μm (**D**). (**E–H**) FACS analysis in TdLNs (LN) of dendritic cells (**E**), CCR7 + DC (**F**) CD86/80 + DC (**G**), and Treg (**H**) in NIR and IR 4NQO-induced WT and TNCKO tumor mice. $N = 5$ mice per group. (**I**) Gene Ontology analysis of the most highly deregulated Biological Processes in the TdLNs of WT and TNCKO mice. Mean ± SEM; Kruskal–Wallis test and Dunn post-test, *$P < 0.05$, **$P < 0.01$, ***$P < 0.005$. The exact $P$ values are listed in Appendix Table S5.

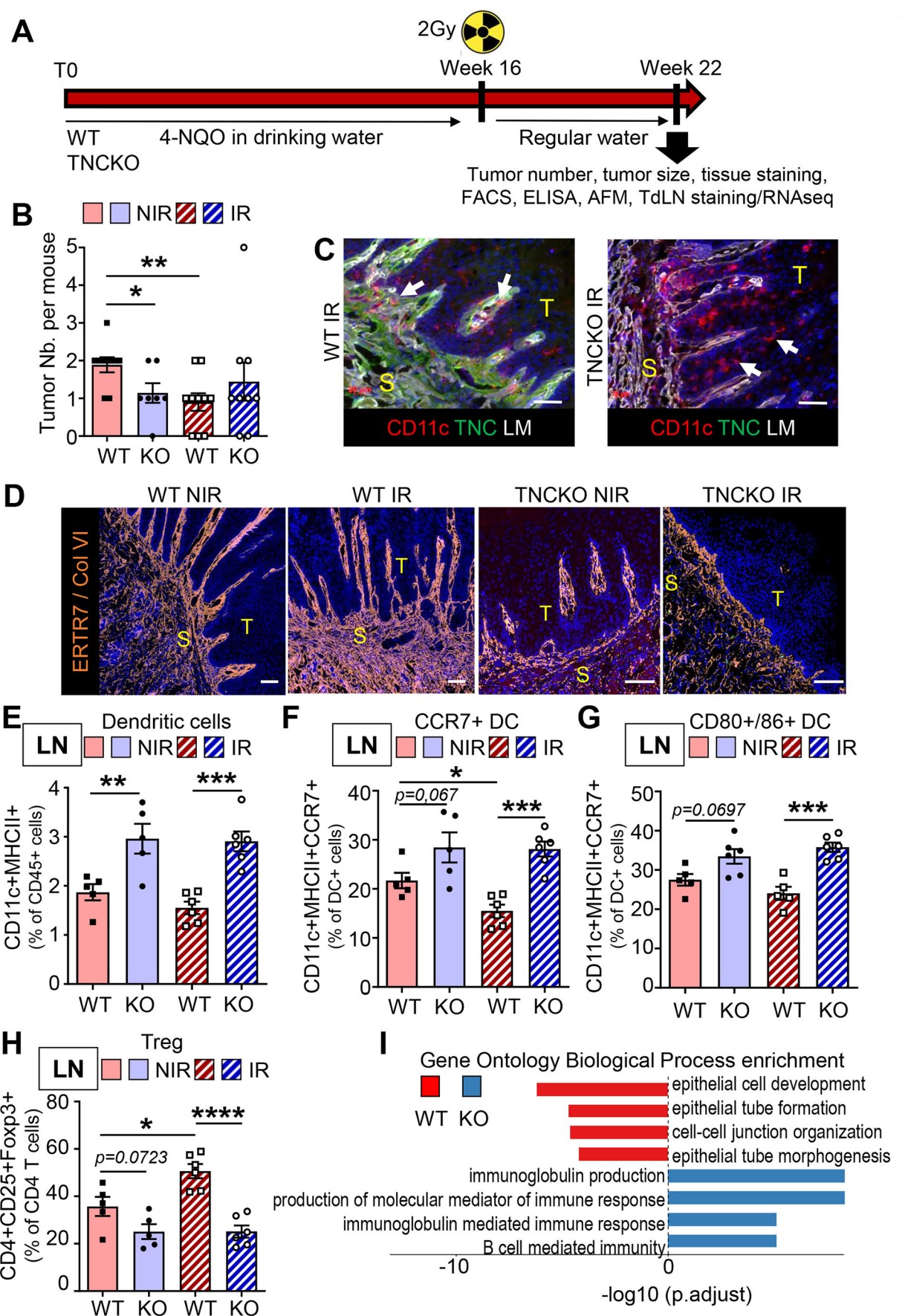

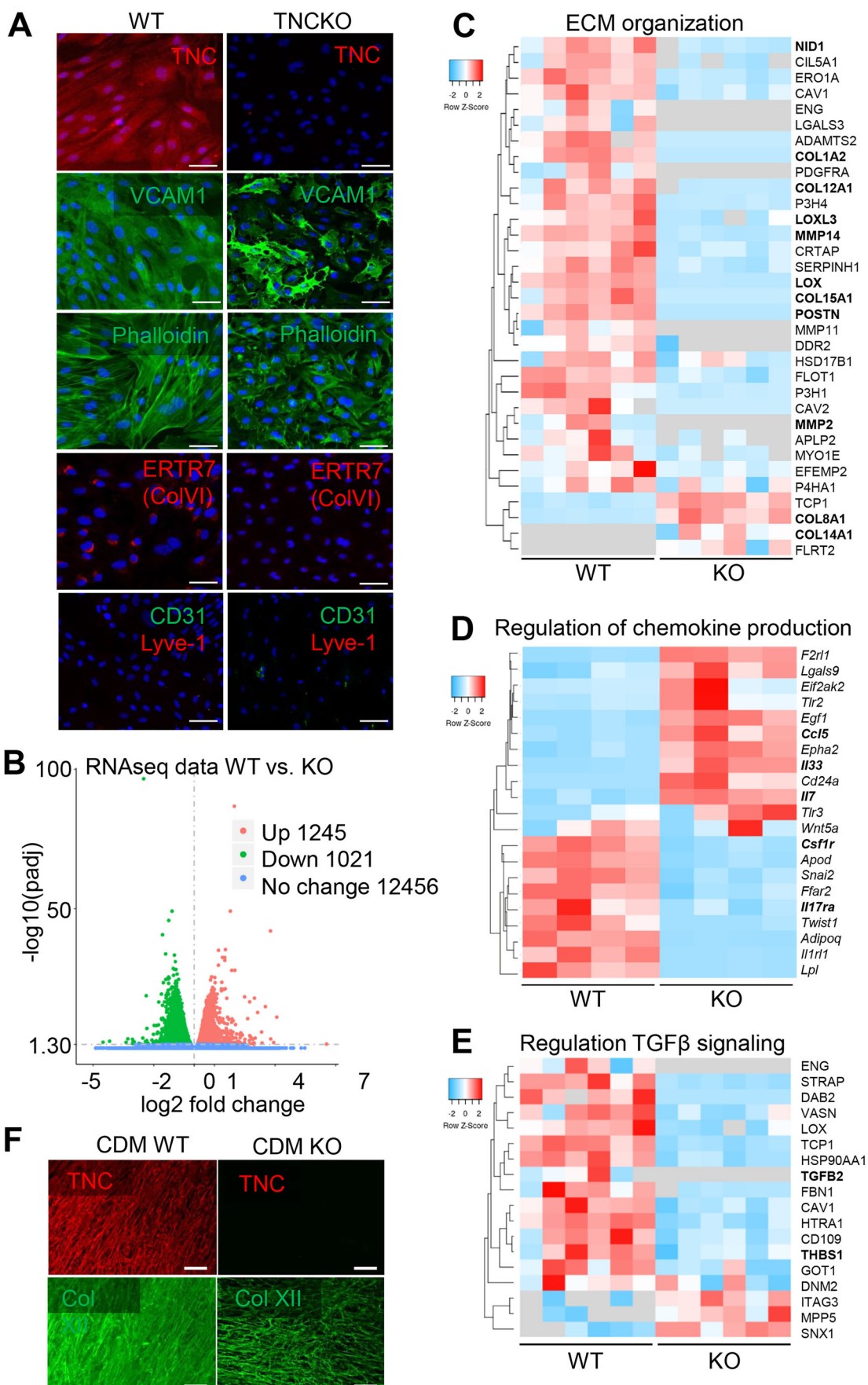

◀ **Figure EV2.   TNC expression plays a pivotal role in determining the FRC identity.**

(A) Representative IF images for TNC, VCAM-1, polymerized actin (phalloidin staining), collagen VI (ERTR7), CD31, and Lyve-1 in FRCs isolated from naïve lymph nodes of WT or TNCKO mice. For VCAM-1, the same microscopic field is shown that has been co-stained for GP38 in Fig. 2A. Scale bar, 50 µm. (B) Volcano plot of deregulated genes (DEGs) obtained after RNA sequencing of FRC WT and TNCKO cells. Volcano plot showing the fold change and the adjusted $P$ value for the 14,722 genes expressed with overexpression (red dots) and downregulation (green dots) in FRC WT compared to TNCKO cells. (C–E) Proteomics data analysis represented as heatmaps for the most deregulated proteins that belong to ECM organization (C), Regulation of chemokine production (D), and regulation of the TGFβ signaling (E). Bold text indicates proteins with established roles in the respective categories that are discussed in the text. (F) Representative IF staining images of Col XII and TNC expression in the CDM obtained after 10 days of FRC WT and TNCKO cell cultures. Scale bar, 200 µm.

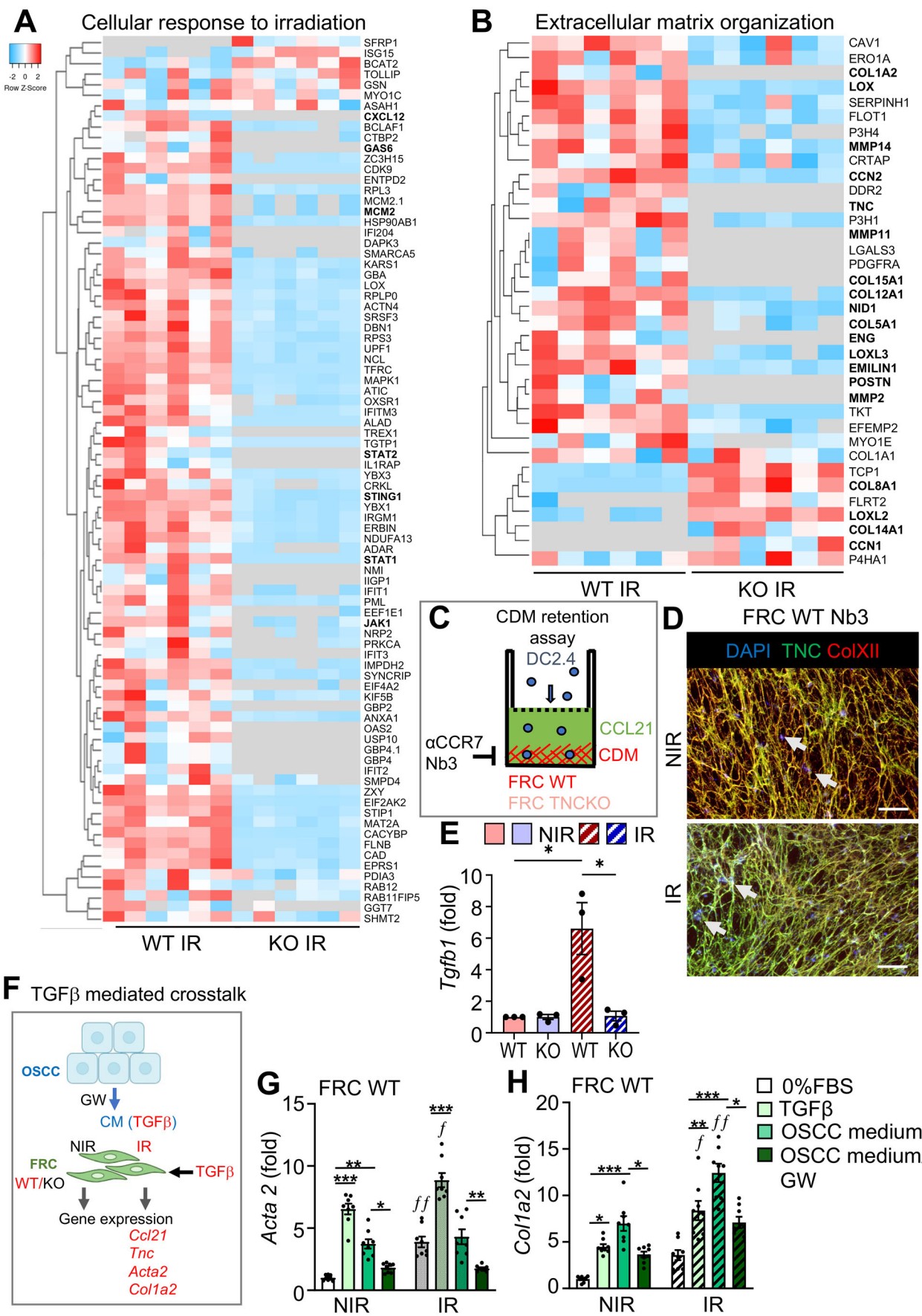

**Figure EV3. The FRC cell response to irradiation is determined by TNC expression.**

(A, B) Heatmap representation of the indicated proteome categories of irradiated FRCs (WT vs. TNCKO), represented as heatmaps for deregulated proteins involved in the cellular response to irradiation (A) and ECM organization (B). Bold text indicates proteins with established roles in the respective categories and/or that are discussed in the text. (C) Schematic representation of the DC retention assay with CCL21 (200 ng/ml) as chemoattractant and anti-CCR7 and Nb3 as TNC inhibitors. Note the less binding of cells to the CDM from TNCKO FRCs (light red) and upon inhibition of TNC. (D) Representative IF images of DC2.4 adherent on the CDM after the chemoretention assay in the presence of Nb3. Staining is as indicated. Arrows indicate the nuclei. Scale bar, 200 μm. (E) Gene expression (qRTPCR) of *Tgfb1* in WT and TNCKO FRCs, NIR or IR. $N = 3$. Mean ± SD, two-way ANOVA with *$P < 0.05$. (F) Schematic representation of the experimental setup to assess a TGFβ-mediated cellular crosstalk between OSCC13 cells (OSCC) providing Conditioned Medium (CM) that was added to FRCs (WT, KO, NIR, IR) in the presence or absence of GW788388 (GW) in comparison to TGFβ (10 ng/ml). Gene expression of the indicated molecules was assessed by qRTPCR. Note a significant stimulating effect of the CM of the OSCC13 cells on irradiated WT FRCs in a TGFβ signaling depenndent manner (red). (G, H) Gene expression as determined by qRTPCR in FRC WT for *Acta2* and *Col1a2*. $N = 6–9$ per condition. Error bars represent mean ± SEM, two-way ANOVA with *$P < 0.05$, **$P < 0.01$, ***$P < 0.005$. The exact $P$ values are listed in Appendix Table S5.

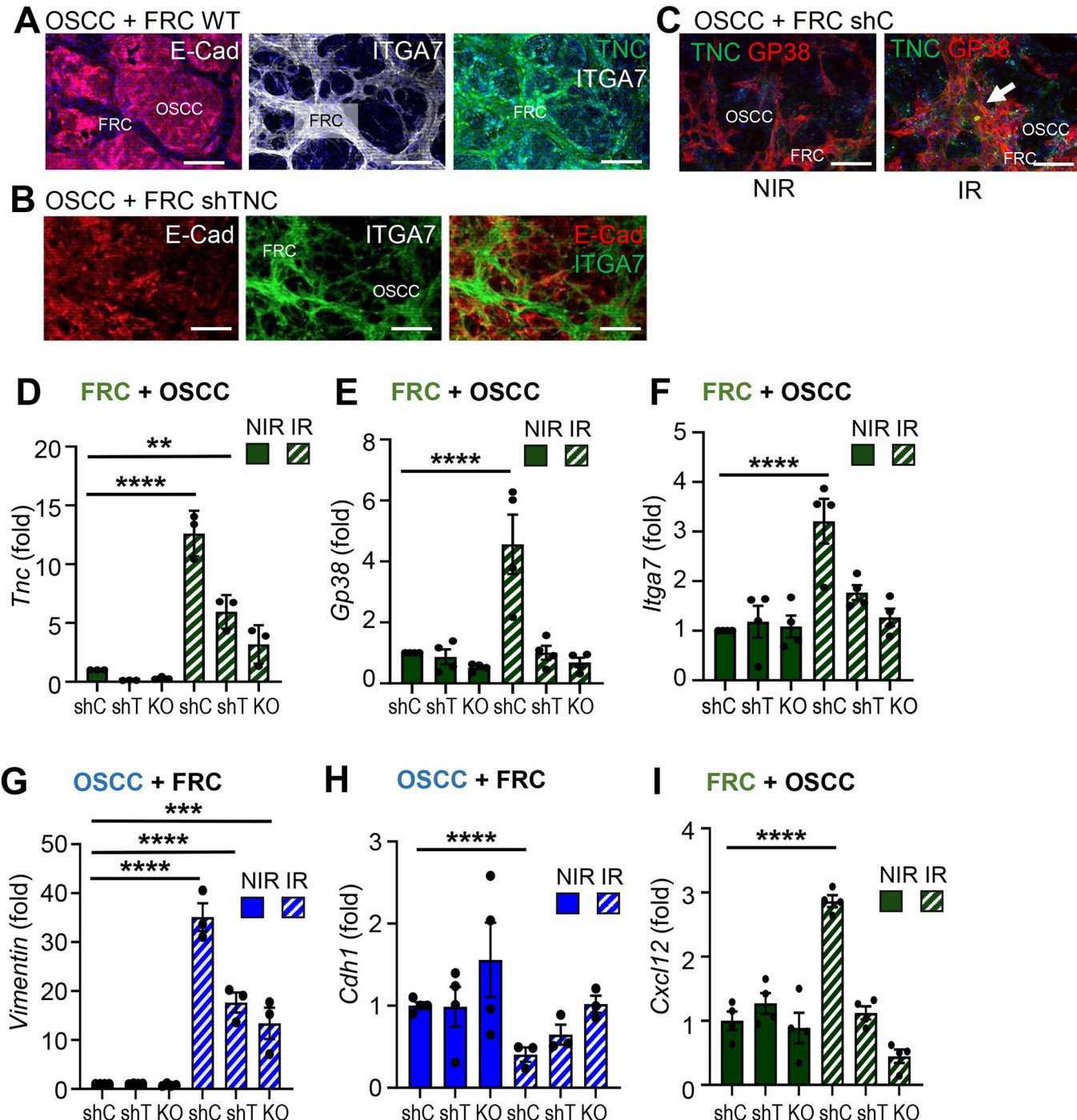

**Figure EV4.  TNC impact on the OSCC-FRC crosstalk upon irradiation.**

(A–C) Immunofluorescence images after staining for the indicated molecules in cocultures of OSCC13 (OSCC) with FRCs at a 2.1 ratio for 4 days. FRCs and OSCC are labeled. Arrow points at increased GP38 and TNC levels in the IR condition (C). Scale bars, 200 μm. (D–I) Gene expression (qRTPCR) in FRCs (WT, shTNC (shT), TNCKO (KO)) (D–F, I) or OSCC13 (OSCC) (G, H) upon coculture, IR or NIR, and isolation by MACS for the indicated molecules. $N = 3$ experiments; Error bars represent mean ± SEM; ordinary one-way ANOVA test, ***$P < 0.005$. ****$P < 0.0001$. The exact $P$ values are listed in Appendix Table S5.

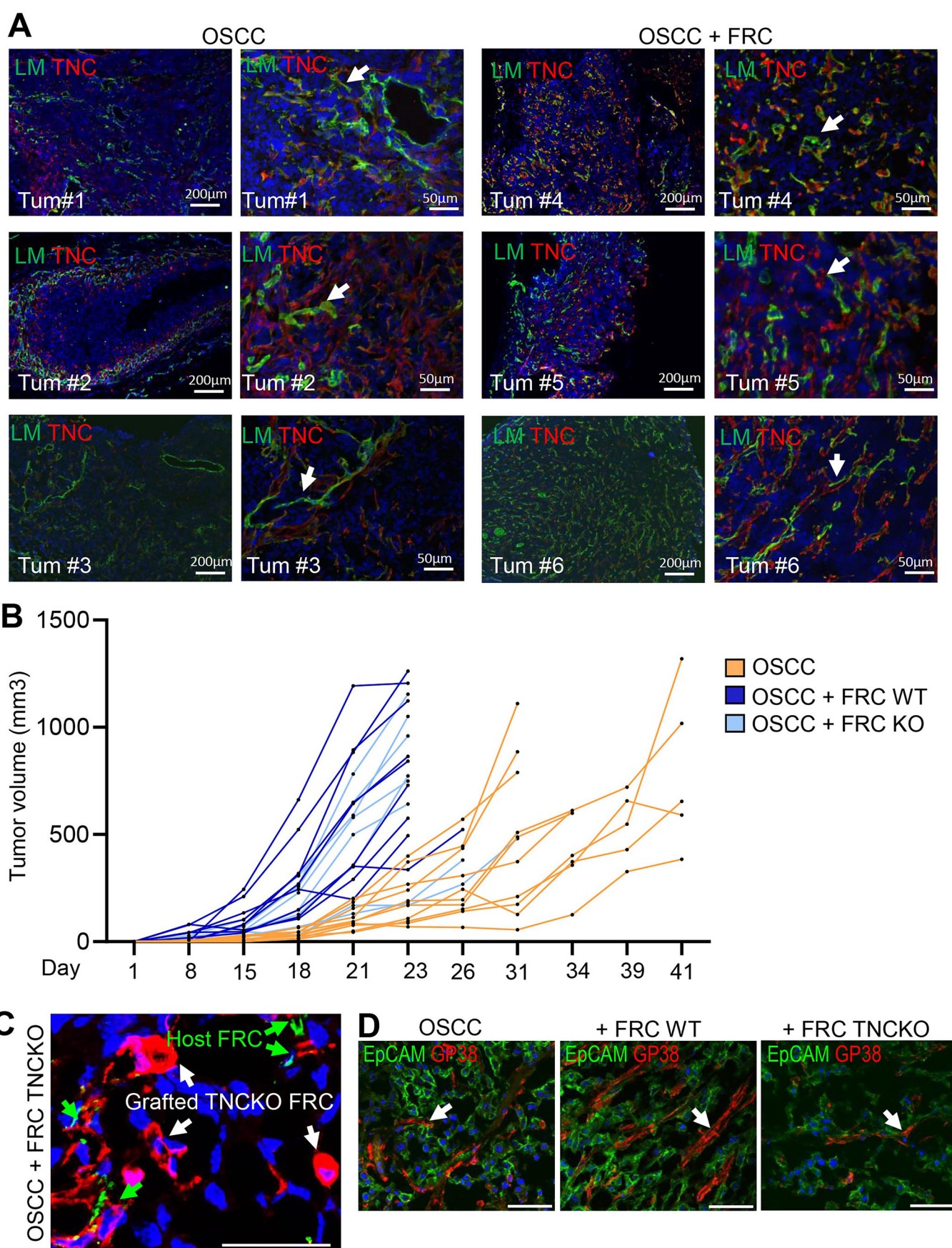

◄ **Figure EV5. Characterization of neck coengrafted tumors in C57Bl6 mice.**

(A) Representative IF images of neck tumors displayed in Fig. 5A, obtained after OSCC grafting ($3 \times 10^6$ cells) or OSCC/FRC (5:1 ratio) co-graftments in WT mice for the indicated markers. The LM/TNC costained panel of the OSCC/FRC tumor #6 represents an uncropped version of the image shown in Fig. 5B. Scale bar, 200 μm. (B) Monitoring of individual tumor growth (volume) of the engrafted OSCC13 (OSCC) cells alone ($3 \times 10^6$ cells) (orange) or combined with WT FRC (dark blue), or FRC TNCKO cells (light blue), in a 5:1 ratio, into the neck of WT mice during the indicated time frame (up to 41 days). (C) Enlarged image showing FRCs (GP38 + ) in a OSCC/ TNCKO FRC-engrafted tumor where the engrafted FRCs are negative for TNC while the host FRCs are positive as indicated with the colored arrows. Scale bar, 50 μm. (D) Representative immunofluorescence images for EpCAM and GP38 in the tumors displayed in Fig. 5D derived from engraftments of the OSCC cells alone ($3 \times 10^6$ cells) (orange) or in combination with WT FRC (dark blue), or FRC TNCKO cells (light blue), in a 5:1 ratio, into the neck of WT mice during the indicated time frame (up to 41 days). Scale bars, 50 μm. Arrow points at FRCs.

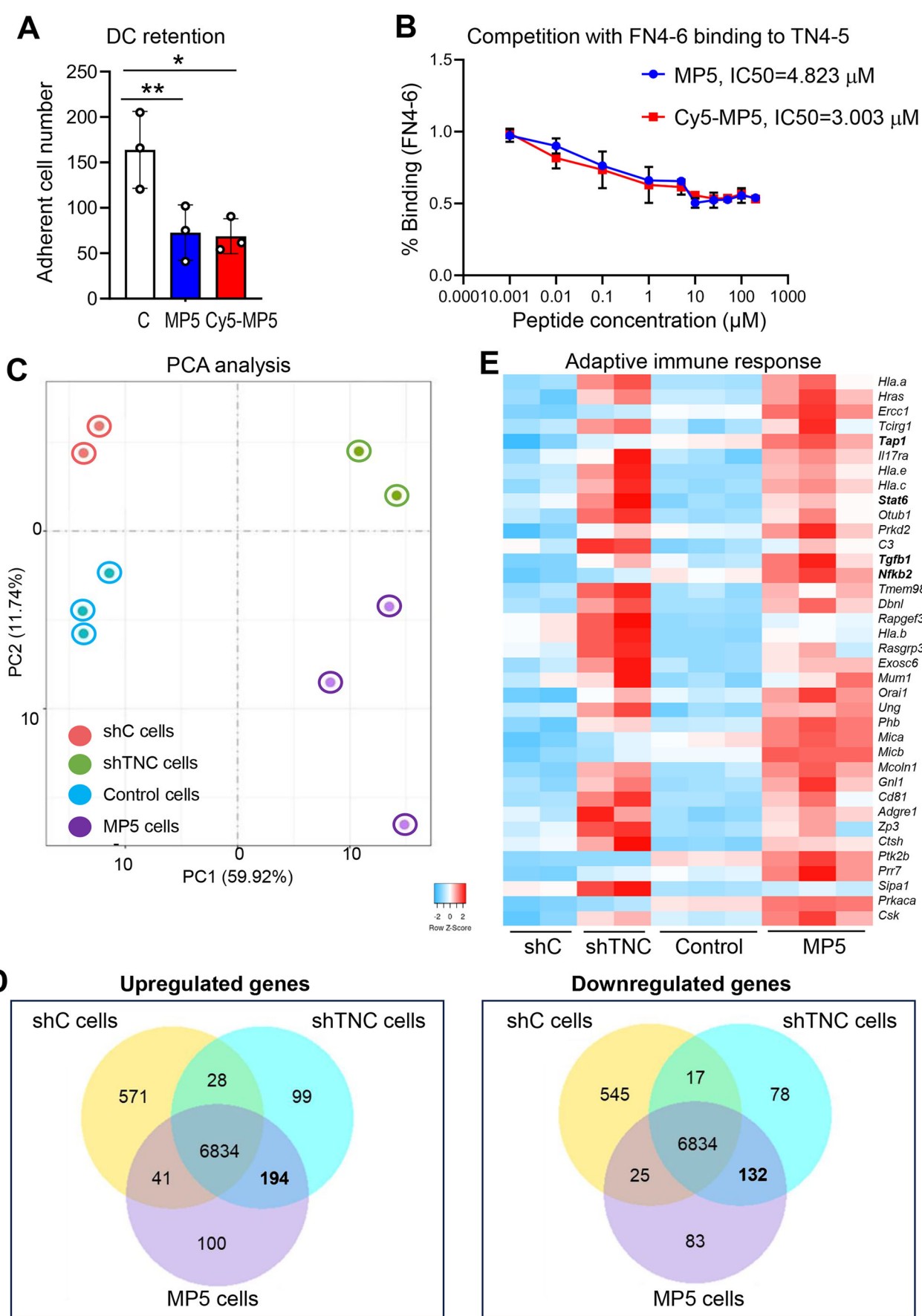

◀ **Figure EV6.   Targeting TNC with MAREMO peptide MP5 reduces tumor cell proliferation and plasticity upon IR.**

(A) Chemoretention assay with DC2.4 cells measuring adhesion on the TNC-coated lower surface of the insert with 200 ng/mL CCL21 as chemoattractant. Note, both peptides (25 nM) cause release of the cells from TNC. $N = 3$ experiments; Mean ± SD; Ordinary one-way ANOVA test, $*P < 0.05$. (B) Competitive ELISA measuring binding of FN4-6 to surface-adsorbed TN4-5 which is blocked in a dose-dependent manner by both MP5 and Cy5-MP5. Dose–response inhibition curves were fitted using nonlinear regression with a four-parameter logistic model. $IC_{50}$ values were compared using an extra sum-of-squares F test. $N = 3$ experiments, Mean ± SD. (C) PCA analysis of the RNA sequencing data from NT193 breast cancer cells (shC, shTNC, Control (PBS-treated) and MP5-treated), indicating a similar distribution of the shTNC and MP5 samples. (D) RNA sequencing analysis of the upregulated and downregulated genes in NT193 breast cancer cells shC, shTNC or MP5-treated. In the centre, the commonly expressed genes between all categories are shown. 194 genes are commonly upregulated in the MP5 and shTNC conditions compared to the shC condition, and 132 genes are commonly downregulated in the MP5 and shTNC conditions compared to the shC condition. (E) Heatmap representation of gene expression data (derived from RNA sequencing) linked to adaptive immune response between groups ($P < 0.05$). The exact $P$ values are listed in Appendix Table S5.

