## [Peer Review File · EMBO Molecular Medicine]

Tenascin-C orchestrates radiotherapy-induced head and neck tumor regression

Thomas Loustau, Ioanna Mitrentsi, Nuohan Wang, Caroline Spenlé, Alexia Pavlidaki, Thibaud Tranchant, Gilles Riegel, Akhil Venu, Rime Oueidat, Manuel Koch, Marion Dumas, Fanny Wack, Aurelie Hirschler, Christine Carapito, Nicodème Paul, Raphael Carapito, Matthias Mörgelin, Uwe Hansen, Joyce Azzi, Lucie Aubergeon, Nathalie Salomé, Sayda Dhaouadi, Pierre Grenot, Balkiss Bouhaouala, Simona La Cioppa, Philipp Oertle, Valerio Izzi, Marija Plodinec, Georges Noel, Hélène Burckel, and Gertraud Orend

Corresponding authors: Gertraud Orend (gertraud.orend@inserm.fr) , Thomas Loustau (thomas.loustau@unistra.fr)

Review Timeline:

Submission Date:	4th Apr 25
Editorial Decision:	8th Apr 25
Revision Received:	4th Aug 25
Editorial Decision:	5th Sep 25
Revision Received:	11th Dec 25
Editorial Decision:	13th Jan 26
Revision Received:	11th Feb 26
Accepted:	2nd Mar 26

Editor: Lise Roth

Transaction Report:

8th Apr 2025

Decision on your manuscript EMM-2025-21740

Dear Gertraud,

Thank you for submitting your full manuscript to EMBO Molecular Medicine following your presubmission inquiry. I have now read your article carefully and discussed it with the other members of our editorial team. I am sorry to inform you that we find that the manuscript is not well suited for publication in EMM, and that we therefore have decided not to proceed with peer review. However, our colleagues from our sister journal Life Science Alliance would be happy to offer peer review if you were to transfer your manuscript there. Life Science Alliance is an open access journal launched in partnership between EMBO Press, Rockefeller Press, and Cold Spring Harbor Laboratory Press, and publishes work that is of high value to the respective communities across all areas in the life sciences.

We appreciate that your study investigates the role of tenascin-C (TNC) in ionizing irradiation (IR)-induced tumor regression in the context of head and neck squamous cell carcinoma (HNSCC), using a 4-NQO-induced tongue oral squamous cell carcinoma (OSCC) mouse model in wild-type of TNC-knockout mice. Your findings identify TNC as a critical regulator of radiosensitivity in HNSCC and suggest that targeting TNC specifically in tumor reticular fibroblast cells may represent a novel therapeutic avenue. However, this is not further demonstrated here, which limits the translational angle of your work. Therefore, I am afraid that we cannot offer further consideration in EMBO Molecular Medicine.

However, given the potential interest of the findings, we would have no objection to considering a new manuscript on the same topic if you were to obtain data in the near future that would significantly strengthen the translational aspect of the manuscript.

Alternatively - and as mentioned above - my colleagues would be pleased to send your manuscript out for formal peer-review at Life Science Alliance in its current form. You can take advantage of this offer by transferring your study to Life Science Alliance using the link below; no re-formatting is required.

I am very sorry to disappoint you on this occasion and would like to reassure you that this is not a judgment of the quality or interest of your work, but a decision based on the scope requirement of our journal.

With my best wishes,

Lise

=====

As a service to authors, EMBO provides authors with the possibility to transfer a manuscript that one journal cannot offer to publish to another EMBO publication. The full manuscript and if applicable, reviewers reports are automatically sent to the receiving journal to allow for fast handling and a prompt decision on your manuscript. For more details of this service, and to transfer your manuscript to another EMBO title please click on Link Not Available

The authors submitted a revised version of the manuscript.

5th Sep 2025

Dear Dr. Orend, Dear Gertraud,

Thank you for submitting your manuscript to EMBO Molecular Medicine, and please accept my apologies for the delay in getting back to you, which was due to the annual leave of referees and editorial staff. We have now received feedback from the three reviewers who agreed to evaluate your manuscript. As you will see from the reports below, the referees acknowledge the interest of the study and are overall supporting publication of your work pending appropriate revisions.

Addressing the reviewers' concerns in full will be necessary for further considering the manuscript in our journal, and acceptance of the manuscript will entail a second round of review. EMBO Molecular Medicine encourages a single round of revision only and therefore, acceptance or rejection of the manuscript will depend on the completeness of your responses included in the next, final version of the manuscript. For this reason, and to save you from any frustrations in the end, I would strongly advise against returning an incomplete revision.

We are expecting your revised manuscript within three months, if you anticipate any delay, please contact us.

We require:

Additional information on source data and instruction on how to label the files are available

4) A .docx formatted letter INCLUDING the reviewers' reports and your detailed point-by-point responses to their comments. As part of the EMBO Press transparent editorial process, the point-by-point response is part of the Review Process File (RPF), which will be published alongside your paper.

5) A complete author checklist, which you can download from our author guidelines (<https://www.embopress.org/page/journal/17574684/authorguide#submissionofrevisions>). Please insert information in the checklist that is also reflected in the manuscript. The completed author checklist will also be part of the RPF.

6) All Materials and Methods need to be described in the main text using our 'Structured Methods' format. According to this format, the Methods section includes a Reagents and Tools Table (listing key reagents, experimental models, software and relevant equipment and including their sources and relevant identifiers) followed by a Methods and Protocols section describing the methods, ideally using a step-by-step protocol format. The aim is to facilitate adoption of the methodologies across labs. Please download and fill our Reagents and Tools Table template (.docx), which you can find in our author guidelines: <https://www.embopress.org/page/journal/14693178/authorguide#structuredmethods>.

7) Please note that all corresponding authors are required to supply an ORCID ID for their name upon submission of a revised manuscript.

8) It is mandatory to include a 'Data Availability' section after the Materials and Methods. Before submitting your revision, primary datasets produced in this study need to be deposited in an appropriate public database, and the accession numbers and database listed under 'Data Availability'. Please remember to provide a reviewer password if the datasets are not yet public (see <https://www.embopress.org/page/journal/17574684/authorguide#dataavailability>).

9) For data quantification: please specify the name of the statistical test used to generate error bars and P values, the number (n) of independent experiments (specify technical or biological replicates) underlying each data point and the test used to calculate p-values in each figure legend. The figure legends should contain a basic description of n, P and the test applied. Graphs must include a description of the bars and the error bars (s.d., s.e.m.). Please provide exact p values.

10) Our journal encourages inclusion of *data citations in the reference list* to directly cite datasets that were re-used and obtained from public databases. Data citations in the article text are distinct from normal bibliographical citations and should directly link to the database records from which the data can be accessed. In the main text, data citations are formatted as follows: "Data ref: Smith et al, 2001" or "Data ref: NCBI Sequence Read Archive PRJNA342805, 2017". In the Reference list, data citations must be labeled with "[DATASET]". A data reference must provide the database name, accession number/identifiers and a resolvable link to the landing page from which the data can be accessed at the end of the reference. Further instructions are available at .

11) We replaced Supplementary Information with Expanded View (EV) Figures and Tables that are collapsible/expandable online. EV Figures should be cited as 'Figure EV1, Figure EV2' etc... in the text and their respective legends should be included in the main text after the legends of regular figures.

12) The paper explained: EMBO Molecular Medicine articles are accompanied by a summary of the articles to emphasize the major findings in the paper and their medical implications for the non-specialist reader. Please provide a draft summary of your article highlighting

13) Author contributions: CRedit has replaced the traditional author contributions section because it offers a systematic machine readable author contributions format that allows for more effective research assessment. Please remove the Authors Contributions from the manuscript and use the free text boxes beneath each contributing author's name in our system to add specific details on the author's contribution. More information is available in our guide to authors.

Please also suggest a visual abstract to illustrate your article as a PNG file 550 px wide x 300-600 px high. A cropped portion of this image will serve as thumbnail for the table of content on our webpage.

16) As part of the EMBO Publications transparent editorial process initiative (see our Editorial at <http://embomolmed.embopress.org/content/2/9/329>), EMBO Molecular Medicine will publish online a Review Process File (RPF) to accompany accepted manuscripts.

In the event of acceptance, this file will be published in conjunction with your paper and will include the anonymous referee reports, your point-by-point response and all pertinent correspondence relating to the manuscript. Let us know whether you agree with the publication of the RPF and as here, if you want to remove or not any figures from it prior to publication. Please note that the Authors checklist will be published at the end of the RPF.

I look forward to receiving your revised manuscript.

Yours sincerely,

Lise Roth

**** Reviewer's comments ****

Referee #1 (Comments on Novelty/Model System for Author):

Technical Quality justification: The study integrates in vivo (4NQO-induced OSCC in immunocompetent WT vs TNC-KO mice), ex vivo (primary FRC cultures), in vitro (co-culture, spheroid systems), and multi-omics (RNA-seq, proteomics) approaches. Controls are appropriate. Multiparameter flow cytometry, AFM stiffness measurements, electron microscopy, and patient dataset analysis strengthen the rigor. The main technical limitation is the irradiation regimen: single low-dose 2 Gy IR is well-controlled but not optimally representative of clinical fractionated dosing for HNSCC, which could limit translational inference for therapy response.

Novelty justification: The identification of FRC-derived Tenascin-C as a central determinant of radiotherapy outcome in HNSCC, with dual tumor-promoting and tumor-regressing roles, is novel. The link between TNC-CCL21-mediated immune cell retention and IR-induced immune suppression is well-developed mechanistically. Introducing MAREMO peptide as a targeted approach against TNC in the IR setting is new. The patient survival stratification using a combined TNC + FRC signature is an original translational application.

Medical impact justification: This work has a strong clinical relevance. TNC is an ECM protein with existing antibody and peptide-targeting possibilities. FRC signatures are assessable in patient samples and linked to survival. The concept of targeting pro-tumorigenic fibroblast subtypes aligns with ongoing stromal immunotherapy research. There are some limitations to be pointed out: the efficacy of the MP5 is only shown in vitro (no in vivo IR + MP5 combination therapy data are provided). The IR regimen differs from standard clinical dosing, which may affect predictive value. There is no direct validation in human primary tumor-derived fibroblasts.

Adequacy of the model justification: 4NQO-induced OSCC mimics histopathology, mutational profile, and immune landscape of human tobacco-associated HNSCC. The choice of using an immunocompetent background allows study of tumor-stroma-immune interactions, which is essential for ECM/immune biology. However, there are some limitations to be pointed out: The model represents HPV-negative, tobacco-associated OSCC, so findings may not generalize to HPV-positive disease. The 2 Gy single dose IR does not model fractionated clinical therapy and may select for different stromal/immune dynamics.

Referee #1 (Remarks for Author):

This manuscript from Loustau et al. investigates TNC as a key regulator of radiotherapy response in HNSCC, with a focus on fibroblast reticular cells as mediators of both anti- and pro-tumorigenic effects following irradiation. Using 4NQO-induced oral squamous cell carcinoma models in wild-type and TNC-knockout mice, the authors integrate in vivo tumor models, primary FRC cultures, co-culture systems, transcriptomics, proteomics, and patient dataset analysis. The work also explores therapeutic targeting of TNC with a MAREMO peptide to mitigate tumor cell plasticity and proliferation post-irradiation.

Strengths of the work: The dual role of TNC in modulating radiosensitivity and immune suppression is well framed and adds mechanistic insight into ECM-immune-tumor crosstalk. The identification of FRC-derived TNC as a prognostic marker in irradiated HNSCC patients is a novel and translationally relevant finding. The therapeutic proof-of-concept with the MAREMO peptide is timely. The integration of in vivo, ex vivo, and in vitro systems, plus omics datasets, provides multi-layered mechanistic evidence for TNC's role in FRC phenotype regulation, ECM remodeling, immune cell retention, and TGF β -driven plasticity.

Analysis of TCGA-HNSCC radiotherapy cohorts and stratification by TNC/FRC signatures directly links findings to patient outcome, enhancing translational impact.

Major concerns to be addressed: 1/Clarity on the dual role of TNC in radiosensitivity : The current narrative oscillates between TNC facilitating radiosensitivity (in WT tumors) and promoting radioresistance (via immune suppression and EMT). While biologically plausible, the manuscript needs a clearer conceptual model reconciling these effects, possibly as a function of spatial context (tumor vs lymph node FRCs) or temporal sequence post-IR. 2/Dose and fractionation in the IR model: The choice of a single low-dose (2 Gy) irradiation is unusual for clinical translation, as HNSCC radiotherapy is typically fractionated at higher cumulative doses. The authors should justify this choice, discuss limitations, and consider whether the TNC-dependent effects persist under a fractionated regimen. 3/ 3/Mechanistic specificity of MAREMO peptide : While MP5 reduces proliferation and plasticity post-IR, the specificity of the observed effects to TNC-fibronectin FNIII domain interaction should be further validated. For example: Does MP5 phenocopy TNC depletion in co-culture transcriptomics? t would strengthen the translational case to test MP5 in vivo in combination with IR. 4/Functional validation of FRC-tumor crosstalk : The conclusion that TGF β is the dominant mediator of IR-induced FRC-tumor crosstalk is well supported, but other cytokines/pathways upregulated in the proteomics could be contributing. The authors should either experimentally rule out key alternatives (such as IL-6, CXCL12) or explicitly acknowledge them in discussion.

Minor concerns: Some p-values are reported without effect sizes. Several figures are data-dense and would benefit from schematic additions summarizing the take-home message. The methods note that both sexes were used but do not state whether sex-specific effects were analyzed.

The manuscript is of high scientific quality, addresses an important question, and has strong translational potential. However, clarification of the dual role of TNC, justification of IR regimen, deeper mechanistic and validation of MP5 are necessary to make the conclusions compelling and clinically meaningful.

Referee #2 (Remarks for Author):

Loustau et al, in their manuscript "Tenascin-C orchestrates radiotherapy-induced head and neck tumor regression" demonstrate the role of tenascin C (TNC) in radiotherapy-induced tumor regression in HNSCC models. They found that fibroblast reticular cells (FRC) regulate the response upon IR and create immunosuppressive tumor microenvironment and thereby resistance to radiotherapy. They also show that TNC is a key mediator in this process which they proved with TNC knockdown and using an inhibitory peptide in 3D spheroid model. Overall the study is highly interesting and innovative. The manuscript is well written, and data is clearly presented. Here are some comments to improve the study.

1. IR induces FRC which may secrete TNC to establish TNC/CCL21-mediated TIL-matrix retention. This process leads to induction of IS-TME. However, these data do not show how the immunosuppression is induced. Please explain.
2. In isolated FRC from TNCKO, there was less ECM expression, collagen production, proliferation.
3. Can authors specifically show which exact features of FRC are controlled by TNC and how?
4. Have authors tried to add TNC to FRC isolated from TNCKO to study the effect on the reversal of features? Is there TNC-FRC interaction?
5. The quality of staining in Fig 2E is a bit fuzzy and should be improved.
6. In line 278, Authors state "this matrix was impaired in retaining the DCs (Fig. 3F)." However, these data show the levels of CCL21. Please clarify.
7. Is there heterogeneity in FRC as only one type of FRC are used during the whole study?
8. In Fig. 5c, the expression of Foxp3 and CD206 is not significantly induced. Therefore, one cannot claim their abundance. One should show the p value even if it is not below 0.05.
9. The staining of CD45 in the co-injected tumors (Fig. 5B) seems to be not discussed in the text.
10. The effect of the MP5 peptide in 3D spheroids is interesting as it blocks the binding of TNC-FNIII to otherFN4-6. Is there a study with the OSCC + FRC TNC-KO in 3D spheroids?
11. Also, it would be interesting to study the effect of the peptide in vivo in co-injection model with IR treatment in vivo.

Referee #3 (Remarks for Author):

This manuscript addresses an important clinical challenge, the mechanisms that convey resistance to tumor cells upon radiation therapy. The study focuses on understanding the role of Tenascin C (TNC), an extracellular matrix protein previously reported to promote immune evasion in tumors. While previous studies also reported that TNC is induced by radiation (IR) therapy, this

manuscript addresses if TNC plays a role in the outcome of IR therapy in HNSCC using a TNC KO mouse model.

First, the authors describe that TNC depletion reduces carcinogen-induced tumor growth (as previously shown) but also reduces the tumor responsiveness to IR. This radioresistance coincides with higher immune cell infiltration but lower fraction of fibroblast reticular cells (FRC) in the tumors of the TNC KO mice.

The authors then isolated FRC from the lymph nodes of healthy(?) WT and TNC KO mice and analysed them by proteomics and RNA sequencing, where FRC of TNC KO mice demonstrate differences in multiple pathways relevant to the tumor microenvironment. The isolated FRCs from WT and TNC KO mice were then treated with IR in vitro, and TNC was found to regulate FRC effects on the ECM and cytokine expression. Importantly, TNC depletion promoted FRCs survival from IR. When TNC depleted FRC were co-cultured with OSCC in vitro, reduced DNA repair response to IR was observed in both FRCs and OSCCs.

FRCs promoted OSCC tumor growth when inoculated together as xenografts, which was delayed if TNC was depleted from the FRCs. In addition, the introduction of a peptide targeting TNC reduced the growth of OSCC+FRC spheroids in vitro. Lastly, the authors demonstrate that FRC gene signature predicts poor survival in TNC high HNSCC.

Although interesting and relevant data are presented, the manuscript is overall very hard to follow and the conclusions have not been well formulated. In all, it is not clear how the authors see the large amount of data coming together coherently into an informative report of their findings.

Major:

1. Together, the data support a regulatory role of TNC in FRC where it mediates increased growth of the adjacent cancer cells and promotes DNA damage response, pro-tumorigenic TME changes and cell survival in response to IR. However, what is strongly conflicting, and complicates the understanding of the study, is the claim of lower tumor responsiveness to IR in TNC KO mouse (Fig. 1A). How do the authors accommodate this finding that sets the stage for the whole paper with the rest of the data that, instead, point to a key role of TNC in FRC in promoting radioresistance?

2. The immunofluorescence images are generally low resolution, and many appear to be of poor quality throughout the manuscript. In several panels showing multiple markers, some images represent the same cell population while others do not, which complicates the interpretation and is somewhat confusing for the reader. The authors should consistently show markers from the same cell population. In addition, the fluorescence signals should be quantified where possible, as not all results are clearly discernible from the IF images alone.

3. The extensive and detailed presentation of altered gene expression pathways appears exaggerated, particularly since the pathways lack subsequent validation and discussion. These data could be presented in a more concise and sophisticated way.

4. The claim that IR induces EMT is not sufficiently supported by the data, as the conclusion is based on enhanced expression of only one marker. Given the complexity of EMT, multiple markers should be assessed to justify such a conclusion.

Minor comments:

1. Some of the figure legends are rather vague and non-informative. They should be revised to reflect the findings more clearly.

2. The introduction part is a mix of introduction, results and discussion. Please revise.

3. There are several structural errors in the abstract that make it difficult to understand the authors' intended message.

4. The figure panels are not clearly constructed: some are placed so closely together that it is unclear which labels belong to which image, and certain images appear to be of low quality or stretched.

5. Tables S1 and S2 are also inconsistently prepared, with mixed use of decimal commas and points, uneven decimal places, and values presented without a logical order to aid the reader in interpretation.

6. The schematic presentations in Fig. 7C-D are not helpful in summarizing the findings in a coherent manner. Please revise thoughtfully.

***** Reviewer's comments *****

Referee #1 (Comments on Novelty/Model System for Author):

Technical Quality justification: The study integrates in vivo (4NQO-induced OSCC in immunocompetent WT vs TNC-KO mice), ex vivo (primary FRC cultures), in vitro (co-culture, spheroid systems), and multi-omics (RNA-seq, proteomics) approaches. Controls are appropriate. Multiparameter flow cytometry, AFM stiffness measurements, electron microscopy, and patient dataset analysis strengthen the rigor. The main technical limitation is the irradiation regimen: single low-dose 2 Gy IR is well-controlled but not optimally representative of clinical fractionated dosing for HNSCC, which could limit translational inference for therapy response.

Authors response: We agree with the reviewer that the use of single-dose irradiation, although experimentally well-controlled, does not adequately reflect the fractionated regimens used in clinical radiotherapy for HNSCC. In our in vitro co-culture experiments, we used a single 10 Gy irradiation, which provides a reproducible model to induce tumor-stroma interactions in vitro. However, to increase translational relevance, we extended our experimental design by implementing a fractionated irradiation protocol on 3D spheroid co-cultures. Specifically, a regimen of 5 × 2 Gy fractions (on consecutive week days) would approximate clinical practice, while limiting cumulative toxicity in vitro compared to a single 10 Gy exposure. This allows sublethal damage repair between fractions and models the biological compromise observed in clinical radiotherapy. A similar reduction in cell numbers upon irradiation between 10 Gy and 5 × 2 Gy (now shown in **Appendix Fig. S6D-E**) revealed a similar response justifying the use of 10 Gy in the cell culture experiments.

Keeping in mind that murine tumors do not necessarily phenocopy the radio responses of human cancers we believe that the use of a single 2 Gy dose is justified as we see profound responses of the irradiated tumors that are described in detail in **Fig. 1 (Fig EV1 and S1)**.

The most appropriate experimentation to address the reviewer's concern is to implement a fractionated irradiation protocol on 3D spheroid co-cultures, especially those using human HNSCC cell lines combined with fibroblasts to mimic clinical radiotherapy regimens more closely as we now did and show the results in **Fig. 6C-D**. This change will significantly increase translational relevance and allow robust comparison with single-dose models. A single 10 Gy fraction (BED = 20 Gy) delivers the same total physical dose as 5×2 Gy (BED = 12 Gy), but carries a high risk of toxicity to surrounding normal tissues. Using 5 fractions of 2 Gy each allows normal tissues to repair sublethal damage between sessions, improving safety and tolerability. While the biological effect on the tumor is lower than a single 10 Gy dose, this fractionation is a clinically accepted compromise that balances tumor control with reduced risk of adverse effects.

Novelty justification: The identification of FRC-derived Tenascin-C as a central determinant of radiotherapy outcome in HNSCC, with dual tumor-promoting and tumor-regressing roles, is novel. The link between TNC-CCL21-mediated immune cell retention and IR-induced immune suppression is well-developed mechanistically. Introducing MAREMO peptide as a targeted approach against TNC in the IR setting is new. The patient survival stratification using a combined TNC + FRC signature is an original translational application.

Medical impact justification: This work has a strong clinical relevance. TNC is an ECM protein with existing antibody and peptide-targeting possibilities. FRC signatures are assessable in patient samples and linked to survival. The concept of targeting pro-tumorigenic fibroblast subtypes aligns with ongoing stromal immunotherapy research. There are some limitations to be pointed out: the efficacy of the MP5 is only shown in vitro (no in vivo IR + MP5 combination therapy data are provided). The IR regimen differs from standard clinical dosing, which may affect predictive value. There is no direct validation in human primary tumor-derived fibroblasts.

Authors response:

As stated above, concerning the irradiation regimen, we acknowledge that our use of single-dose 2 Gy in vivo does not fully mirror standard fractionated clinical protocols. However, this dose was selected because it reliably reduced tumor incidence and volume in the 4NQO-induced OSCC model, providing a controlled context to study tumor–stroma–immune interactions.

In our in vitro co-culture experiments, we employed 10 Gy single-dose irradiation as the cells were shown to tolerate this dose (Spel  et al., 2021). To further address this concern, we performed additional experiments using OSCC13/FRC spheroid co-cultures exposed to fractionated irradiation (5 × 2 Gy over consecutive weekdays) and compared them to a single 10 Gy protocol. We assessed spheroid cell number with and without MP5, allowing us to evaluate the effect of fractionation on both tumor biology and the peptide’s therapeutic efficacy. These results suggest a similar effect of 10 Gy versus 5 x 2 Gy on reducing the cell numbers, now displayed in **Appendix Fig S6C-E**.

To enhance translational value and in line with the reviewer’s suggestion, we extended our studies to human HNSCC spheroid models by combining CAL33 cells with telomerase immortalized human fibroblasts (TIF). The results show a similar effect on reducing the cell numbers as in the murine model and with a single dose of 10 Gy and 5 x 2 Gy, now displayed in **Fig. 6C-D, S6C**. This model will allow us to test fractionated irradiation protocols in human systems in the future and to explore the impact of MP5 in a context more directly relevant to the clinical setting which was not a focus of this paper.

It is important to note that the efficacy of MP5 in vivo has already been demonstrated in our previous work (Li et al., 2024), where it caused significant tumor regression. In the present manuscript, our contribution is to show how MP5 can specifically modulate radiotherapy-induced processes such as reducing cell numbers and EMT in stromal-rich tumor models. While we agree that combining MP5 with radiotherapy in an in vivo model would be highly informative, designing and completing such an extensive study, particularly in the context of fractionated protocols, would require several additional months and goes beyond the scope of the current manuscript. We see our work as a foundation that justifies such future in vivo investigations.

Adequacy of the model justification: 4NQO-induced OSCC mimics histopathology, mutational profile, and immune landscape of human tobacco-associated HNSCC. The choice of using an immunocompetent background allows study of tumor-stroma-immune interactions, which is essential for ECM/immune biology. However, there are some limitations to be pointed out: The model represents HPV-negative, tobacco-associated OSCC, so findings may not generalize to HPV-positive disease. The 2 Gy single dose IR does not model fractionated clinical therapy and may select for different stromal/immune dynamics.

Authors response:

We acknowledge the limitation that HPV-positive HNSCC, which is typically associated with a better response to treatment and improved patient survival, is not adequately represented by the 4NQO and OSCC13 models. However, HPV-negative, tobacco-associated HNSCC remains more frequent worldwide and presents with markedly worse prognosis. For this reason, we consider the use of the chosen model to be highly relevant to the central objectives of our study. This limitation has been explicitly stated in the manuscript (page 17, line 555).

In addition, we recognize that our use of a single 2 Gy dose of irradiation does not replicate fractionated clinical regimens. Our rationale for employing this approach was to generate a controlled and reproducible perturbation of the tumor–stroma–immune ecosystem, while minimizing confounding variables introduced by fractionation. We agree that different irradiation schedules may elicit distinct stromal and immune dynamics, and we now clarify this limitation in the revised text (page 5: line 133).

Referee #1 (Remarks for Author):

This manuscript from Loustau et al. investigates TNC as a key regulator of radiotherapy response in HNSCC, with a focus on fibroblast reticular cells as mediators of both anti- and pro-tumorigenic effects following irradiation. Using 4NQO-induced oral squamous cell carcinoma models in wild-type and TNC-knockout mice, the authors integrate in vivo tumor models, primary FRC cultures, co-culture systems, transcriptomics, proteomics, and patient dataset analysis. The work also explores therapeutic targeting of TNC with a MAREMO peptide to mitigate tumor cell plasticity and proliferation post-irradiation.

Strengths of the work: The dual role of TNC in modulating radiosensitivity and immune suppression is well framed and adds mechanistic insight into ECM-immune-tumor crosstalk. The identification of FRC-derived TNC as a prognostic marker in irradiated HNSCC patients is a novel and translationally relevant finding. The therapeutic proof-of-concept with the MAREMO peptide is timely. The integration of in vivo, ex vivo, and in vitro systems, plus omics datasets, provides multi-layered mechanistic evidence for TNC's role in FRC phenotype regulation, ECM remodeling, immune cell retention, and TGF β -driven plasticity. Analysis of TCGA-HNSCC radiotherapy cohorts and stratification by TNC/FRC signatures directly links findings to patient outcome, enhancing translational impact.

Major concerns to be addressed:

1/Clarity on the dual role of TNC in radiosensitivity : The current narrative oscillates between TNC facilitating radiosensitivity (in WT tumors) and promoting radioresistance (via immune suppression and EMT). While biologically plausible, the manuscript needs a clearer conceptual model reconciling these effects, possibly as a function of spatial context (tumor vs lymph node FRCs) or temporal sequence post-IR.

Authors response: We agree with the reviewer's comment and have now clarified the dual effects of TNC by revising both the summary figure **Fig. 7C** and the corresponding text (see page 14, line 454, page 15, line 470). We also generated a **graphical abstract** that illustrates the context-dependent roles of TNC and updated Fig. 7C accordingly. We also explained the molecular events in more detail as shown in **Fig. 7D**, with more information in the legend.

We also provide novel data that allow to clarify the dual roles of TNC better. Our FACS results already indicated an impaired function of the TdLN. We substantiated this results by doing RNA seq analysis of TdLNs that we had collected from the 4NQO-exposed WT and TNCKO tumor-bearing mice. This analysis revealed substantial gene expression differences between the two genotypes. Notably, TdLNs from WT mice exhibited higher expression of genes associated with epithelial tube and structural programs, whereas TdLNs from TNCKO mice showed a prominent enrichment of B cell-related and adaptive immunity-associated gene signatures (**Fig. 1I**; **Fig. EV1I**; **Appendix Fig. S1R-T**).

These findings were further validated at the tissue level: immunostaining demonstrated markedly increased numbers of CD3⁺ T cells and B220⁺ B cells in the TdLNs of TNCKO tumor-bearing mice compared with their WT counterparts. These results are now presented in **Fig. 1J** and described in the manuscript on page 6, line 185 – page 7 line 202.

2/Dose and fractionation in the IR model: The choice of a single low-dose (2 Gy) irradiation is unusual for clinical translation, as HNSCC radiotherapy is typically fractionated at higher cumulative doses. The authors should justify this choice, discuss limitations, and consider whether the TNC-dependent effects persist under a fractionated regimen.

Authors response: We acknowledge the reviewer's important point regarding the clinical relevance of dose and fractionation. In our in vivo model, we chose a single irradiation dose of 2 Gy. This choice was based on several considerations. First, 2 Gy represents the clinically relevant fraction size typically used in standard fractionated radiotherapy protocols for HNSCC. While patients receive multiple fractions totaling higher cumulative doses, the

biological response to each individual 2 Gy fraction is clinically meaningful and forms the basis of radiotherapy fractionation schedules.

Second, the experimental question we aimed to address was the identification of early TNC-dependent mechanisms triggered by irradiation within the tumor and its microenvironment. For this purpose, a single 2 Gy exposure was sufficient to induce significant and measurable effects, including a marked reduction in tumor number and size, as well as profound alterations in the tumor microenvironment. These effects provided us with adequate tumor material for mechanistic analyses, which would have been more difficult to obtain with higher or multiple doses that risk excessive tumor regression and loss of analyzable tissue.

Third, the molecular pathways we investigated, particularly the induction of TNC and TGF- β , have been shown to be activated both after single dose irradiation and under fractionated regimens (Meyer et al., 2017). On this basis, we conclude that a single 2 Gy dose is appropriate to model the fundamental biology of radiation-induced TNC responses, even if it does not fully replicate clinical treatment schedules. While additional fractions would likely enhance and prolong the observed effects, we do not anticipate that the underlying mechanisms would differ substantially between single-dose and fractionated irradiation.

To further strengthen the translational relevance of our findings, we performed additional in vitro experiments under fractionated conditions. Specifically, we exposed OSCC13 spheroids co-cultured with fibroblastic reticular cells (FRCs) to 5 \times 2 Gy, and compared their response, such as cell number, EMT induction and modulation by the TNC-targeting peptide MP5, to those obtained after a single 10 Gy exposure. These results confirmed that the TNC-dependent responses observed after single-dose irradiation are also preserved under a fractionated regimen, supporting the robustness of our conclusions and the biological validity of our model. These results are now shown in **Appendix Fig S6C-E** and mentioned in the manuscript on page 13, line 411.

3/Mechanistic specificity of MAREMO peptide : While MP5 reduces proliferation and plasticity post-IR, the specificity of the observed effects to TNC-fibronectin FNIII domain interaction should be further validated. For example: Does MP5 phenocopy TNC depletion in co-culture transcriptomics? It would strengthen the translational case to test MP5 in vivo in combination with IR.

Authors response: We had published results on NT193 breast tumors and cultured cells lacking TNC (TNCKO, Murdamoothoo et al., 2021) and cells that were treated with MP5 (Li et al., 2024). We now compared the gene expression profiles, in these cells. Principal component analysis (PCA) confirmed that MP5-treated cells clustered together with shTNC cells, whereas PBS-treated controls clustered with shC cells. Consistently, MP5-treated tumor cells displayed gene-expression patterns more similar to shTNC cells than to control cells. We further showed that genes associated with the GO categories “adaptive immune response,” “antigen presentation,” and “cytokine response” were similarly upregulated following TNC depletion and MP5 treatment. These results are now shown in **Fig EV6D** and **Appendix Fig S6E-G** and mentioned in the manuscript on page 13, line 414.

4/Functional validation of FRC-tumor crosstalk : The conclusion that TGF β is the dominant mediator of IR-induced FRC-tumor crosstalk is well supported, but other cytokines/pathways upregulated in the proteomics could be contributing. The authors should either experimentally rule out key alternatives (such as IL-6, CXCL12) or explicitly acknowledge them in discussion.

Authors response: We are well aware that presumably several signaling pathways are regulated by TNC and the chosen treatment. Here, we focused on TGF β as it is well established that irradiation induces TGF β and associated signaling (Massagué, 2008), Fig

EV3E). As requested, we investigated whether IL6 and CXCL12 were regulated by TNC and the treatment in the cocultures by qRT-PCR and observed that *Cxcl12* was increased in the co-cultures after irradiation in a TNC dependent manner, but *Il6* remained unchanged in the WT FRCs that are of importance. In future experiments CXCL12/CXCR4 inhibitors could be used to determine the relevance of pathway activation/repression. These results are now shown in **Fig EV4I** and **Appendix Fig S4U-W** and mentioned in the manuscript on page 12, line 358. We also added a sentence in the discussion on page 15, line 491 that likely other pathways are involved.

Minor concerns: Some p-values are reported without effect sizes. Several figures are data-dense and would benefit from schematic additions summarizing the take-home message. The methods note that both sexes were used but do not state whether sex-specific effects were analyzed.

Authors response: we like to thank this reviewer for noticing this negligence. We have provided the requested information. We now provide a take home message for some of the figures in **Fig. EV3C and F** and **Figs 7C and D**.

Eighteen female mice and 19 male mice were used for the 4NQO experiments and were divided equally between the IR and non-IR conditions. Across all parameters measured, such as tumor size and immune cell profile, no differences mediated by mouse sex were detected. However, in the grafting experiments we only used male mice because the OSCC13 cells and the FRCs derived from a male mouse. This information has been added in the Material & Methods part on page 15, 564, page 19, line 622.

The manuscript is of high scientific quality, addresses an important question, and has strong translational potential. However, clarification of the dual role of TNC, justification of IR regimen, deeper mechanistic and validation of MP5 are necessary to make the conclusions compelling and clinically meaningful.

Authors response: we like to thank this reviewer for all critical remarks, suggestions for improvement and appreciation of the quality and relevance of our results and followed the advice to clarify the dual role of TNC in irradiation responses.

Referee #2 (Remarks for Author):

Loustau et al, in their manuscript "Tenascin-C orchestrates radiotherapy-induced head and neck tumor regression" demonstrate the role of tenascin C (TNC) in radiotherapy-induced tumor regression in HNSCC models. They found that fibroblast reticular cells (FRC) regulate the response upon IR and create immunosuppressive tumor microenvironment and thereby resistance to radiotherapy. They also show that TNC is a key mediator in this process which they proved with TNC knockdown and using an inhibitory peptide in 3D spheroid model. Overall the study is highly interesting and innovative. The manuscript is well written, and data is clearly presented. Here are some comments to improve the study.

1. IR induces FRC which may secrete TNC to establish TNC/CCL21-mediated TIL-matrix retention. This process leads to induction of IS-TME. However, these data do not show how the immunosuppression is induced. Please explain.

Authors response: Our published data derived from RNA seq analysis demonstrated several immunity associated molecules to be expressed in a TNC dependent manner in the 4NQO tumors that altogether contribute to TNC orchestrating an IS-TME as previously published (Spel   et al., 2020). Based on these results that comprised information that FRCs express TNC and CCL21 in a TNC dependent manner and that TNC binds CCL21 to form an adhesive substratum for dendritic cells and macrophages (Loustau et al., 2022; Spel   et al., 2020), we followed up this particular aspect upon radiotherapy. Now we show that irradiation is increasing the abundance of FRCs where TNC is a driver as this is not the case

in the TNCKO mouse. However, it remains to be determined by which molecular mechanism irradiation enforces the FRC abundance apart from TGF β . Here, we identified TGF β as a key player that is well known to be induced by irradiation. We show that TGF β is a target downstream of TNC that gets induced in the FRCs in the cocultures upon irradiation. TGF β plays likely multiple roles that were not exhaustively investigated here (as this is not the focus of this study). Binding of TGF β may enforce its signaling as seen in a breast cancer model (Sun et al., 2019). By using a TGF β RII inhibitor we demonstrate that TGF β is activating the FRCs (inducing higher TNC and CCL21 levels) and causing EMT in the OSCC cells upon irradiation. These responses require the coculture of the two cell types, however the exact mechanism remains to be determined and is not the focus of this article. This has been discussed on page 16, line 520.

2. In isolated FRC from TNCKO, there was less ECM expression, collagen production, proliferation.

Authors response: we do not understand what the question is.

3. Can authors specifically show which exact features of FRC are controlled by TNC and how?

Authors response: The RNA seq and proteomics results demonstrate that TNC largely impacts the cell phenotype, rendering cells sensitive to irradiation when they express TNC. Why TNCKO FRCs are insensitive towards irradiation remains to be determined.

4. Have authors tried to add TNC to FRC isolated from TNCKO to study the effect on the reversal of features? Is there TNC-FRC interaction?

Authors response: We have performed an experiment where we added TNC as soluble agent to FRCs lacking TNC (KO) and compared the expression of *Col1a1* and *Itga7*, two of the signature FRC genes that we had identified, with FRCs expressing TNC or with the FRC TNC KO. We observed no difference after the addition of TNC and the results are displayed below for the attention of the reviewers. Due to the high number of results we did not add this result to the manuscript.

FRCs express several integrins that could bind to TNC, however depending on the integrin the responses might be different as previously published for other cells. Cell rounding on TNC through integrin $\alpha9\beta1$ blocked YAP and MKL1 signaling (Sun et al., 2018). However, binding to other integrins (e.g. $\alpha v\beta3$) inducing cell spreading may have the opposite effects and induce YAP and MKL1 signaling. However, this is not known for FRCs and is beyond the scope of this article.

5. The quality of staining in Fig 2E is a bit fuzzy and should be improved.

Authors response: We have provided high quality images that reveal fibrillar matrix patterns in the CDM of WT FRCs. This pattern is less prominent in the CDM from TNCKO cells as described in the manuscript page 8, line 235.

6. In line 278, Authors state "this matrix was impaired in retaining the DCs (Fig. 3F)." However, these data show the levels of CCL21. Please clarify.

Authors response: Sorry for this mistake. Indeed, **Fig 3F** shows the DC retention assay results. The *Ccl21* levels are displayed in **Fig 3G**. We have corrected the text accordingly.

7. Is there heterogeneity in FRC as only one type of FRC are used during the whole study?

Authors response: The FRCs were isolated from all lymph nodes of naïve age-matched WT or KO mice, thus it is expected that different FRC subtypes are present in the FRC isolates. Previous studies identified nine peripheral stromal cell (SC) clusters in the lymph node, several of which correspond to fibroblastic reticular cell (FRC) subpopulations. These include *Ccl19*⁺ T-zone reticular cells (TRCs), marginal reticular cells, follicular dendritic cells (FDCs), and perivascular cells, illustrating the transcriptional diversity of SC populations and their niche-specific immune functions. To determine which of these stromal subsets were represented in our model, we applied the published gene signatures of each cluster and assessed their enrichment. Our RNA seq analysis indeed reveals such signatures that are present in both WT and TNC KO FRCs. Interestingly, the expression of subtype specific indicator genes is not different between genotypes. What is different is the expression of general FRC markers (9 gene signature). These data are shown in **Appendix Fig S2A** and mentioned in the manuscript, on page 7, line 221.

8. In Fig. 5c, the expression of *Foxp3* and *CD206* is not significantly induced. Therefore, one cannot claim their abundance. One should show the p value even if it is not below 0.05.

Authors response: Non-significant expression levels have been removed from the Fig 5C.

9. The staining of *CD45* in the co-injected tumors (Fig. 5B) seems to be not discussed in the text.

Authors response: this mistake has now been rectified on page 12, line 378.

10. The effect of the MP5 peptide in 3D spheroids is interesting as it blocks the binding of TNC-FNIII to otherFN4-6. Is there a study with the OSCC + FRC TNC-KO in 3D spheroids?

Authors response: To assess the TNC specificity of MP5, we generated cocultured OSCC13 (shTNC)/FRC (TNC-KO) spheroids that lack TNC expression and subjected them to IR and MP5 treatment using the same protocol applied to TNC-expressing spheroids. Unlike their TNC-positive counterparts, these TNC-deficient spheroids showed no decrease in cell numbers following MP5 treatment, demonstrating that the MP5 response requires the presence of TNC. This TNC-dependent effect is presented in **Appendix Fig. S6F** and described in the manuscript on page 13, line 414.

11. Also, it would be interesting to study the effect of the peptide in vivo in co-injection model with IR treatment in vivo.

Authors response: We agree that it will be important to study the effect of the peptide in vivo in combination with IR treatment. We have already shown though that the peptide has an effect on breast cancer tumors in vivo, in our previous publication by Li et al. 2024. An in vivo experiment in the OSCC/FRC model is however out of the scope of this article as it requires the establishment of a reasonable protocol, that may require ip injection of the peptide as the OSCC13 tumors develop a strong fibrotic capsule that impairs delivery of MP5 via a peritumoral application as has been done in the past (Li et al., 2024). Also the irradiation and treatment protocol has to be determines as e.g. which dose of irradiation? how often? peptide treatment before or after irradiation?

Referee #3 (Remarks for Author):

This manuscript addresses an important clinical challenge, the mechanisms that convey resistance to tumor cells upon radiation therapy. The study focuses on understanding the role of Tenascin C (TNC), an extracellular matrix protein previously reported to promote immune evasion in tumors. While previous studies also reported that TNC is induced by

radiation (IR) therapy, this manuscript addresses if TNC plays a role in the outcome of IR therapy in HNSCC using a TNC KO mouse model.

First, the authors describe that TNC depletion reduces carcinogen-induced tumor growth (as previously shown) but also reduces the tumor responsiveness to IR. This radioresistance coincides with higher immune cell infiltration but lower fraction of fibroblast reticular cells (FRC) in the tumors of the TNC KO mice.

The authors then isolated FRC from the lymph nodes of healthy WT and TNC KO mice and analysed them by proteomics and RNA sequencing, where FRC of TNC KO mice demonstrate differences in multiple pathways relevant to the tumor microenvironment. The isolated FRCs from WT and TNC KO mice were then treated with IR in vitro, and TNC was found to regulate FRC effects on the ECM and cytokine expression. Importantly, TNC depletion promoted FRCs survival from IR. When TNC depleted FRC were co-cultured with OSCC in vitro, reduced DNA repair response to IR was observed in both FRCs and OSCCs.

FRCs promoted OSCC tumor growth when inoculated together as xenografts, which was delayed if TNC was depleted from the FRCs. In addition, the introduction of a peptide targeting TNC reduced the growth of OSCC+FRC spheroids in vitro. Lastly, the authors demonstrate that FRC gene signature predicts poor survival in TNC high HNSCC.

Although interesting and relevant data are presented, the manuscript is overall very hard to follow and the conclusions have not been well formulated. In all, it is not clear how the authors see the large amount of data coming together coherently into an informative report of their findings.

Authors response: We agree and in order to overcome this problem have now reorganized the supplementary figures into **Expanded View** and **Appendix Supplementary figures**. The most important results of the previous Supplementary figures are now presented in the Expanded View, and are convinced that this will help the reader to follow the conclusions of the manuscript.

Major:

1. Together, the data support a regulatory role of TNC in FRC where it mediates increased growth of the adjacent cancer cells and promotes DNA damage response, pro-tumorigenic TME changes and cell survival in response to IR. However, what is strongly conflicting, and complicates the understanding of the study, is the claim of lower tumor responsiveness to IR in TNC KO mouse (Fig. 1A). How do the authors accommodate this finding that sets the stage for the whole paper with the rest of the data that, instead, point to a key role of TNC in FRC in promoting radioresistance?

Authors response: We appreciate the reviewer's comment and have better described the dual role of TNC. Therefore we had collected the TdLNs from 4NQO-treated WT and TNC KO tumor-bearing mice, isolated RNA, and performed bulk RNA sequencing. This analysis revealed pronounced transcriptional differences between the two genotypes. Specifically, WT TdLNs showed higher expression of genes associated with epithelial tube structural programs, whereas TdLNs from TNC-deficient mice exhibited stronger expression of B cell-related and adaptive immune pathways (**Fig. 1I, EV1I**). We also demonstrated markedly increased numbers of CD3⁺ T cells and B220⁺ B cells in TdLNs of TNC KO tumor mice compared with WT controls (**Fig. 1J**). We reason that TNC exerts opposing functions: in TdLNs it supports reticular networks required for effective T cell priming, while in the TME the same TNC–CCL21 axis retains immune cells in the stroma and limits their entry into tumor islets. Thus, therapeutic outcomes likely reflect the balance between these opposing effects, which may shift following IR. These results are now presented in **Fig. 1J Fig. 1I, Fig. EV1I and Appendix Fig. S1R–T** and described on the manuscript at page 6, line 185 and discussed on page 15, line 470.

2. The immunofluorescence images are generally low resolution, and many appear to be of poor quality throughout the manuscript. In several panels showing multiple markers, some images represent the same cell population while others do not, which complicates the interpretation and is somewhat confusing for the reader. The authors should consistently show markers from the same cell population. In addition, the fluorescence signals should be quantified where possible, as not all results are clearly discernible from the IF images alone.

Authors response: all stainings have been well labeled and are of high resolution which should allow the reader to get the information. In **Fig. 4** we have indeed quantified expression of some markers (Rad51, P-Smad and vimentin). For an improved understanding we have labeled inside the image which marker recognizes which cell type (e.g. **Fig. 4C, G, EV4A-C**).

3. The extensive and detailed presentation of altered gene expression pathways appears exaggerated, particularly since the pathways lack subsequent validation and discussion. These data could be presented in a more concise and sophisticated way.

Authors response: We have now reorganized the presentation of the figures and included only a few heatmaps in the EV figures. We kept the other heatmaps, now presented in the Appendix supplemental figure files, as they clearly show the extent of gene/protein expression differences between conditions and thus enhance credibility of the data. As we cannot comment on most of these molecule we have marked some genes/proteins of interest in bold as they either are well known markers of the respective biological category and/or are referred to in the manuscript. This is explained in the figure legend.

4. The claim that IR induces EMT is not sufficiently supported by the data, as the conclusion is based on enhanced expression of only one marker. Given the complexity of EMT, multiple markers should be assessed to justify such a conclusion.

Authors response: Our conclusion is supported by convergent evidence from both qRTPCR and immunofluorescence analyses across multiple EMT-associated markers. In the initial manuscript, qRTPCR demonstrated a consistent IR-induced upregulation of *Tgfb1*, Vimentin, and *Twist*, accompanied by a decrease in *Cdh1* expression, indicative of a shift towards a mesenchymal transcriptional program. Complementing these findings, immunofluorescence analysis confirmed increased Vimentin protein levels and enhanced pSMAD signaling, reflecting activation of the TGF β –SMAD pathway, a canonical driver of EMT. Taken together, these results, obtained using two independent methodologies and a coherent panel of epithelial and mesenchymal markers, robustly support our conclusion that IR promotes EMT in our model.

Minor comments:

1. Some of the figure legends are rather vague and non-informative. They should be revised to reflect the findings more clearly.

Authors response: We agree with the reviewer, and we have now modified the figure legends in order to reflect the findings more clearly.

2. The introduction part is a mix of introduction, results and discussion. Please revise.

Authors response: We have considerably shortened the introduction and discussion and, have removed one paragraph from the discussion considering IR-induced mechanical properties in context of TNC.

3. There are several structural errors in the abstract that make it difficult to understand the authors' intended message.

Authors response: We apologize and have modified the abstract. After the addition of the new results concerning the role of TNC in the TdLNs we believe that the message is now clear.

4. The figure panels are not clearly constructed: some are placed so closely together that it is unclear which labels belong to which image, and certain images appear to be of low quality or stretched.

Authors response: We have rearranged the figure panel structure and provided them as single TIFF files, which would improve their quality.

5. Tables S1 and S2 are also inconsistently prepared, with mixed use of decimal commas and points, uneven decimal places, and values presented without a logical order to aid the reader in interpretation.

Authors response: The supplemental tables are now revised and the commas have been replaced by points. The order that we have presented the values, is according to the order of the heatmaps that appear in the text. In these tables we show the p values of all the genes or proteins that appear throughout the manuscript.

6. The schematic presentations in Fig. 7C-D are not helpful in summarizing the findings in a coherent manner. Please revise thoughtfully

Authors response: The schematic representations have been updated, and our new results in the TdLNs supporting the original conclusions are now fully integrated. In addition, the figure legends accompanying these schematics have been revised to more clearly explain the underlying results and to highlight the central message of the study.

References

- Li, C., Kaur, A., Pavlidaki, A., Spenlé, C., Rajnpreht, I., Donnadieu, E., Salomé, N., Molitor, A., Carapito, R., Wack, F., Erne, W., Lefebvre, O., Averous, G., Mitrentsi, I., Loustau, T., Orend, G., 2024. Targeting the MAtRix REgulating MOTif abolishes several hallmarks of cancer, triggering antitumor immunity. *Proc. Natl. Acad. Sci. U. S. A.* 121, e2404485121. <https://doi.org/10.1073/pnas.2404485121>
- Loustau, T., Abou-Faycal, C., Erne, W., Zur Wiesch, P.A., Ksouri, A., Imhof, T., Mörgelin, M., Li, C., Mathieu, M., Salomé, N., Crémel, G., Dhaouadi, S., Bouhaouala-Zahar, B., Koch, M., Orend, G., 2022. Modulating tenascin-C functions by targeting the MAtRix REgulating MOTif, "MAREMO." *Matrix Biol. J. Int. Soc. Matrix Biol.* 108, 20–38. <https://doi.org/10.1016/j.matbio.2022.02.007>
- Massagué, J., 2008. TGF β in Cancer. *Cell* 134, 215–230. <https://doi.org/10.1016/j.cell.2008.07.001>
- Meyer, J.E., Finnberg, N.K., Chen, L., Cvetkovic, D., Wang, B., Zhou, L., Dong, Y., Hallman, M.A., Ma, C.-M.C., El-Deiry, W.S., 2017. Tissue TGF- β expression following conventional radiotherapy and pulsed low-dose-rate radiation. *Cell Cycle Georget. Tex* 16, 1171–1174. <https://doi.org/10.1080/15384101.2017.1317418>
- Murdamoothoo, D., Sun, Z., Yilmaz, A., Riegel, G., Abou-Faycal, C., Deligne, C., Velazquez-Quesada, I., Erne, W., Nascimento, M., Mörgelin, M., Cremel, G., Paul, N., Carapito, R., Veber, R., Dumortier, H., Yuan, J., Midwood, K.S., Loustau, T., Orend, G., 2021. Tenascin-C immobilizes infiltrating T lymphocytes through CXCL12 promoting breast cancer progression. *EMBO Mol. Med.* n/a, e13270. <https://doi.org/10.15252/emmm.202013270>
- Spenlé, C., Loustau, T., Burckel, H., Riegel, G., Abou Faycal, C., Li, C., Yilmaz, A., Petti, L., Steinbach, F., Ahowesso, C., Jost, C., Paul, N., Carapito, R., Noël, G., Anjuère, F., Salomé, N., Orend, G., 2021. Impact of Tenascin-C on Radiotherapy in a Novel Syngeneic Oral Squamous Cell Carcinoma Model With Spontaneous Dissemination to the Lymph Nodes. *Front. Immunol.* 12, 636108. <https://doi.org/10.3389/fimmu.2021.636108>
- Spenlé, C., Loustau, T., Murdamoothoo, D., Erne, W., Beghelli-de la Forest Divonne, S., Veber, R., Petti, L., Bourdely, P., Mörgelin, M., Brauchle, E.-M., Cremel, G.,

- Randrianarisoa, V., Camara, A., Rekima, S., Schaub, S., Nouhen, K., Imhof, T., Hansen, U., Paul, N., Carapito, R., Pythoud, N., Hirschler, A., Carapito, C., Dumortier, H., Mueller, C.G., Koch, M., Schenke-Layland, K., Kon, S., Sudaka, A., Anjuère, F., Van Obberghen-Schilling, E., Orend, G., 2020. Tenascin-C Orchestrates an Immune-Suppressive Tumor Microenvironment in Oral Squamous Cell Carcinoma. *Cancer Immunol. Res.* 8, 1122–1138. <https://doi.org/10.1158/2326-6066.CIR-20-0074>
- Sun, Z., Schwenzer, A., Rupp, T., Murdamoothoo, D., Vegliante, R., Lefebvre, O., Klein, A., Hussenet, T., Orend, G., 2018. Tenascin-C promotes tumor cell migration and metastasis through integrin $\alpha 9\beta 1$ -mediated YAP inhibition. *Cancer Res.* 78, 950–961. <https://doi.org/10.1158/0008-5472.CAN-17-1597>
- Sun, Z., Velázquez-Quesada, I., Murdamoothoo, D., Ahowesso, C., Yilmaz, A., Spénlé, C., Averous, G., Erne, W., Oberndorfer, F., Oszwald, A., Kain, R., Bourdon, C., Mangin, P., Deligne, C., Midwood, K., Abou-Faycal, C., Lefebvre, O., Klein, A., van der Heyden, M., Chenard, M.-P., Christofori, G., Mathelin, C., Loustau, T., Hussenet, T., Orend, G., 2019. Tenascin-C increases lung metastasis by impacting blood vessel invasions. *Matrix Biol. J. Int. Soc. Matrix Biol.* <https://doi.org/10.1016/j.matbio.2019.07.001>

13th Jan 2026

Dear Dr. Orend, Dear Gertraud,

Thank you for submitting your revised study, and please accept my apologies for the delay in getting back to you, which is due to the reduced activity of our office at this time of the year, and to the large number of submissions and revisions we have recently received. We have now received the reports from the two referees who were asked to evaluate your revised manuscript. As you will see below, they are overall satisfied with the revisions pending minor revisions, and I will therefore be able to accept your manuscript once the following concerns are addressed:

1/ Referee #3' concerns:

Please address the remaining concerns from this referee through text revisions. Regarding point #2, please either add quantification, or tone down the claims. Please provide a point-by-point rebuttal letter.

2/ Manuscript text:

- Please indicate in track changes mode any new modification.
- Please provide current email address for Sayda Dhaouadi (sayda.dhaouadi@hotmail.fr bounced)
- Remove the abbreviations list and incorporate it into the manuscript text.
- Correct the headings and order of the manuscript sections to: Abstract / Keywords / The Paper Explained / Introduction / Results / Discussion / Methods / Data Availability / Acknowledgements / Disclosure and Competing Interests Statement / References / Figure Legends / Expanded View Figure Legends.
- "Materials and Methods" should be renamed "Methods".
 - o Cells: Please indicate whether the cells were authenticated and tested for mycoplasma contamination. Please indicate the origin of all cells.
 - o Animals: Please provide housing and husbandry conditions.
 - o Statistics: please provide a statement on sample size, randomization, blinding and inclusion/exclusion criteria
 - o Please provide dilutions/concentrations for all antibodies used in the study.
- Data availability: please provide URLs for the deposited datasets, and note that they must be public before acceptance of the manuscript.
- Funding should be merged with Acknowledgments. Please ensure that the funders list in our system is complete and accurate (it should match the information provided in the manuscript), with project numbers where available, as it will be directly linked to the donor database in the published article .
- Competing Interests should be changed to Disclosure and Competing Interests Statement. Please add author company employment (Matthias Mörgelin - Colzyx, Simona La Cioppa and Philippe Oertle - Artidis).
- Author contributions: please remove from the manuscript text. Please provide CRedit (Contributor Role Taxonomy) terms in the submission system instead.
- References: please correct formatting to 10 authors listed before et al, and remove the DOIs of published articles. Please check that references are not duplicated (i.e. Murdamoothoo, Devadarssen, Zhen Sun, Alev Yilmaz, et al. 2021b. "Tenascin-C Immobilizes Infiltrating T Lymphocytes through CXCL12 Promoting Breast Cancer Progression." EMBO Molecular Medicine n/a (n/a): e13270. <https://doi.org/10.15252/emmm.202013270>.)

3/ Figures:

- Please make sure that all figures/figure panels are referenced in the text (currently, a callout is missing for Fig. 4D). The callout for Table S2 should be corrected to Appendix Table S2. Appendix Table S3 is called out before Appendix Table S1.
- Appendix: please correct the nomenclature in the legends to Appendix Figure S1, etc. and Appendix Table S1, etc.
- During our routine image checks, similarities were found between panels from different figures. Please note that figure reuse is allowed, but should be indicated in the figure legends. Moreover, we noticed that the images across the figure set appear pixelated under analysis. This is a common result of converting original 16-bit TIFF images to RGB format for publication, and while not a cause for concern, it can sometimes give the impression of image alteration to critical readers. To resolve this please upload the figure set and the corresponding source data at the original captured resolution.
- Please address the queries from our data editors in the figure legends:
 1. Please note that the legends for figure 1 is not provided in the sequential manner (legend for figure 1F is provided before legend of figure 1E). This needs to be rectified.
 2. Please note that the legends for figures 4, 5 are not provided in the sequential manner. This needs to be rectified.
 3. Please define the annotated p values ****/***/*/* as well as provide the exact p-values for the same in the legend of figure EV6 A, B as appropriate.
 4. Please indicate the statistical test used for data analysis in the legends of figures 2B, 7A, B; EV3 E, G, H; EV6 A, B
 5. Please note that information related to n is missing in the legends of figures 1D, E, F, G, H, I; 2F, 3D, 5C, D; EV1 E-H; EV3 E, G, H; EV6 A, B
 6. Please note that the error bars are not defined in the legends of figures 2F, 3D, EV3 E, G, H; EV6 A, B
 7. Please note that the scale bar needs to be defined for figures 3E, 4A, C, G; EV3 D, EV4 A-C
 8. Please note that the white dotted borders are not defined in the legend of figure 6A, C. This needs to be rectified.

9. Please note that the white arrows are not defined in the legend of figures 5B, E; EV1 C. This needs to be rectified.

4/ Checklist:

- please clarify what are the restrictions indicated in the section "Newly created materials".
- please fill in the section on Cell materials/cell lines.
- please fill in the section Experimental animals/housing and husbandry conditions.
- please fill in the entire section Experimental study design and statistics.
- please fill in the section Adherence to community standards/ARRIVE guideline.

5/ Source data: all image data is very low resolution and pixelated, please provide higher quality figures. Please provide source data for Figure 1H. Please check the source data for figure 2F.

6/ Synopsis:

Please note that the synopsis should include a short stand first (maximum of 300 characters, including space) as well as 2-5 one-sentences bullet points that summarizes the paper. Please write the bullet points to summarize the key NEW findings. They should be designed to be complementary to the abstract - i.e. not repeat the same text. We encourage inclusion of key acronyms and quantitative information (maximum of 30 words / bullet point). Please use the passive voice.

7/ As part of the EMBO Publications transparent editorial process initiative (see our Editorial at <http://embomolmed.embopress.org/content/2/9/329>), EMBO Molecular Medicine will publish online a Review Process File (RPF) to accompany accepted manuscripts.

This file will be published in conjunction with your paper and will include the anonymous referee reports, your point-by-point response and all pertinent correspondence relating to the manuscript. Let us know whether you agree with the publication of the RPF and as here, if you want to remove or not any figures from it prior to publication.

I look forward to receiving your revised manuscript.

With kind regards,

Lise

***** Reviewer's comments *****

Referee #2 (Comments on Novelty/Model System for Author):

Authors have used 3D spheroids in vitro models and knockout mouse in vivo models which are optimal for this study. The in vivo study was approved by the ethical committee, therefore there are no ethical issues are seen.

Referee #2 (Remarks for Author):

Authors investigate a new mechanism for radio-resistance in head and neck squamous cell carcinoma by demonstrating the role of tenascin C-expressing fibroblast reticular cells. These cells promote resistance by developing immunosuppressive microenvironment and represent an excellent target to improve efficacy of radiotherapy. Authors have now addressed the raised comments in a convincing manner with clarification or additional analyses.

Referee #3 (Remarks for Author):

Together, the presented data demonstrate a compartmentalized role for TNC and FRCs in regulating HNSCC tumor radiotherapy outcomes. In the tumor, TNC-expression in FRCs supports the growth and survival of cancer cells upon IR. However, in the tumor draining lymph nodes (TdLNs), TNC-expression in FRCs supports anti-tumorigenic functions through

induction of adaptive immunity. The net therapeutic outcome of IR therefore likely depends on the balance between TNC's immune-supporting functions in the TdLNs and its immune-restrictive activity in the TME where IR enhances TGFb, TNC, and CCL21 expression within the tumor, reinforcing a "sticky" ECM environment for DCs and T cells.

The authors have clarified the presentation and interpretation of the data. However, some aspects remain to be addressed.

1. To clarify the dual role of TNC in tumor response to radiation therapy, the authors have isolated and sequenced the RNA of TdLN from TNC WT and KO mice. The data are presented as a volcano plot (Fig. 1I), bar graph (Fig EV1I) and Table S3. In the graphs, WT enriched genes/pathways are depicted with red and KO enriched genes/pathways in blue. However, in the text the authors state that they detected "differences in 493 genes to be downregulated and 62 genes to be upregulated in WT in comparison to the TNCKO TdLNs. Gene Set Enrichment Analysis (GSEA) demonstrated that the most deregulated categories showed downregulation of processes linked to the structural organization of the reticular networks (epithelial tube formation, cell-cell junctions, namely Cdh1, Epcam) in the TNC depleted TdLNs while processes involved in adaptive immunity (immunoglobulin production, B cell mediated immunity, e.g. genes encoding several IgK molecules) were the most upregulated categories in the same TdLNs (Figs 1I, EV1I, Appendix Table S3). Moreover, markers for lymphatic endothelial cells (LEC) such as Lyve 1 and Acta for activated fibroblasts/FRCs were higher in the WT than TNCKO TdLNs indicating an altered cellular composition and function in the TdLNs (Appendix Table S3)." In the volcano plot, Acta and Lyve1 are depicted in blue as 'KO genes' but the text above states the opposite. Please check that the data and their interpretation are correct and clearly presented in these figures.

2. The authors state in their point-by-point letter that "We also demonstrated markedly increased numbers of CD3⁺ T cells and B220⁺ B cells in TdLNs of TNC KO tumor mice compared with WT controls (Fig. 1J)." In the manuscript they interpret this result to suggest a deregulated and impaired immunity. In order to state that the accumulation of adaptive immune cells in the LNs is indicative of defective T cell priming, the authors should present evidence of reduced activation of the accumulating immune cells. Overall, this data is based on single microscopy images with no quantification of the result or even mention of the number of samples that were analysed. The data should be quantified from several mice and the interpretation carefully evaluated.

3. In the abstract of the revised manuscript, the authors state that "While tumors in a TNC-expressing host were radiosensitive, they were radioresistant in genetically depleted TNC mice due to impaired tumor draining lymph nodes (TdLNs) immunity." Although they detected signs of impaired TdLN immunity in TNC mice, the data are correlative at this stage. More mechanistic studies should be conducted to make this statement, and it should be rephrased to reflect the data.

4. The schematic Figures 7C-D explaining the authors' conceptual model of the data were updated only marginally. Interpretation of the main framework of data into only one schematic figure would be very helpful for the reader as the compartmentalized and dual role of TNC in this paper is rather complicated.

5. The figure 1 title 'Expression of TNC impacts tumor radiosensitivity, and immune cell infiltration in the TdLNs of a murine model of OSCC' is somewhat misleading. The data in Figure 1 shows mostly infiltration in the tumor, not in TdLNs. Please consider revising the title.

**** Reviewer's comments ****

Referee #2 (Comments on Novelty/Model System for Author):

Authors have used 3D spheroids in vitro models and knockout mouse in vivo models which are optimal for this study. The in vivo study was approved by the ethical committee, therefore there are no ethical issues are seen.

Referee #2 (Remarks for Author):

Authors investigate a new mechanism for radio-resistance in head and neck squamous cell carcinoma by demonstrating the role of tenascin C-expressing fibroblast reticular cells.

These cells promote resistance by developing immunosuppressive microenvironment and represent an excellent target to improve efficacy of radiotherapy. Authors have now addressed the raised comments in a convincing manner with clarification or additional analyses.

Author's reply: we like to thank this reviewer for his approval of the resubmitted manuscript file.

Referee #3 (Remarks for Author):

Together, the presented data demonstrate a compartmentalized role for TNC and FRCs in regulating HNSCC tumor radiotherapy outcomes. In the tumor, TNC-expression in FRCs supports the growth and survival of cancer cells upon IR. However, in the tumor draining lymph nodes (TdLNs), TNC-expression in FRCs supports anti-tumorigenic functions through induction of adaptive immunity. The net therapeutic outcome of IR therefore likely depends on the balance between TNC's immune-supporting functions in the TdLNs and its immune-restrictive activity in the TME where IR enhances TGF β , TNC, and CCL21 expression within the tumor, reinforcing a "sticky" ECM environment for DCs and T cells.

The authors have clarified the presentation and interpretation of the data. However, some aspects remain to be addressed.

1. To clarify the dual role of TNC in tumor response to radiation therapy, the authors have isolated and sequenced the RNA of TdLN from TNC WT and KO mice. The data are presented as a volcano plot (Fig. 1I), bar graph (Fig EV1I) and Table S3. In the graphs, WT enriched genes/pathways are depicted with red and KO enriched genes/pathways in blue. However, in the text the authors state that they detected "differences in 493 genes to be downregulated and 62 genes to be upregulated in WT in comparison to the TNCKO TdLNs. Gene Set Enrichment Analysis (GSEA) demonstrated that the most deregulated categories showed downregulation of processes linked to the structural organization of the reticular networks (epithelial tube formation, cell-cell junctions, namely Cdh1, Epcam) in the TNC depleted TdLNs while processes involved in adaptive immunity (immunoglobulin production, B cell mediated immunity, e.g. genes encoding several IgK molecules) were the most upregulated categories in the same TdLNs (Figs 1I, EV1I, Appendix Table S3). Moreover, markers for lymphatic endothelial cells (LEC) such as Lyve 1 and Acta for activated fibroblasts/FRCs were higher in the WT than TNCKO TdLNs indicating an altered cellular composition and function in the TdLNs (Appendix Table S3)." In the volcano plot, Acta and Lyve1 are depicted in blue as 'KO genes' but the text above states the opposite. Please check that the data and their interpretation are correct and clearly presented in these figures.

Author's reply: we apologize for the mistake and are grateful to this reviewer for a critical reading! We have made the correction accordingly and specify that Acta 1 is a cytoskeletal protein (line 248) that together with Lyve-1 is more expressed in the TNC-depleted TdLNs.

2. The authors state in their point-by-point letter that "We also demonstrated markedly increased numbers of CD3⁺ T cells and B220⁺ B cells in TdLNs of TNC KO tumor mice compared with WT controls (Fig. 1J)." In the manuscript they interpret this result to suggest a deregulated and impaired immunity. In order to state that the accumulation of adaptive immune cells in the LNs is indicative of defective T cell priming, the authors should present evidence of reduced activation of the accumulating immune cells. Overall, this data is based on single microscopy images with no quantification of the result or even mention of the number of samples that were analysed. The data should be quantified from several mice and the interpretation carefully evaluated.

Author's reply: We have done an additional experiment where we stained and quantified the signal of CD3⁺ and B220⁺ cells as readout for a higher abundance of T and B cells in the TdLNs of the TNCKO tumor mice and show the result in NEW Fig. 1 J and K. In the legend it is mentioned that these results derive from 2-4 sections of TdLNs of 5 WT and 5 TNCKO tumor mice.

The already shown flow cytometry data of the TdLNs (**Fig EV1**) providing evidence for a higher number of dendritic cells and less Treg in the TdLNs from the TNCKO tumor mice further support a cellular imbalance that likely renders these TdLNs poorly functional. In support we further investigated the RNA seq data that show a significantly deregulated expression of around 200 genes comprising GO terms related to inflammatory and adaptive immune responses with some genes mentioned in the main text such as several highly expressed Igk and Igh molecules. These results are shown in a heatmap representation in **Appendix Fig S1V, W** and **Table S1**. Together, the results clearly show a deregulated immunity which may contribute to the radioresistance of the TNCKO mice, however how exactly, is beyond the scope of this article and the lack of this information is mentioned as a limitation at the end of the discussion.

Line 292: "A heatmap representation reveals a significantly deregulated expression of around 200 genes comprising GO terms related to inflammatory and adaptive immune responses including *Ccl28*, *Cx3cl1* and *Il34* downregulated and, *Ifng*, *Granzyme B*, *Ifnar*, *Il17*, *Ctla4* and *Cd4* as well as several *Igh* and *Igk* genes upregulated in the TdLNs from TNCKO mice (**Appendix S1V, W, Table S1**)."

3. In the abstract of the revised manuscript, the authors state that "While tumors in a TNC-expressing host were radiosensitive, they were radioresistant in genetically depleted TNC mice due to impaired tumor draining lymph nodes (TdLNs) immunity." Although they detected signs of impaired TdLN immunity in TNC mice, the data are correlative at this stage. More mechanistic studies should be conducted to make this statement, and it should be rephrased to reflect the data.

Author's reply: We have provided more data that support some presumed impairment of the TdLN immunity as there are more T and B cells (New **Figure 1J, K**) and DC (Fig. EV1E-G) and less Treg in the TNC-depleted TdLNs and an altered gene expression of immunity associated genes. We also changed the abstract to: "TNC plays a compartmentalized and dual role in regulating tumor radiosensitivity with a detrimental role in the tumor stroma opposed to an essential role in the tumor draining lymph nodes." We are convinced that the provided analysis of the TdLNs by RNA seq analysis, flow cytometry, tissue staining and signal quantification allows to conclude that the immunity function of the TNC-depleted TdLNs is at least partially impaired which likely contributes to radioresistance. Whether this is the only reason for radioresistance of the tumors and what the exact mechanism is has to be further explored in the future. This is mentioned in line 749: "Finally, although we provide evidence that immunity in the TdLNs of TNC-depleted tumor mice is deregulated which likely contributes to radioresistance, the underlying mechanism has to be worked out."

4. The schematic Figures 7C-D explaining the authors' conceptual model of the data were updated only marginally. Interpretation of the main framework of data into only one schematic figure would be very helpful for the reader as the compartmentalized and dual role of TNC in this paper is rather complicated.

Author's reply: We do not agree that one summary figure would allow to integrate the different effects in the tumor and the TdLNs and would allow to link these effects to the identified molecular mechanisms. We have provided a detailed description of the summary cartoons and are convinced that the legends provides now sufficient information to fully understand both figure panels.

Explicitly, we have done the following changes: by using a different color code we now better emphasize the compartmentalized effects of TNC on the tumor (yellow shading) versus the TdLNs (purple shading) upon radiotherapy and the sub-compartments of stroma (green) versus tumor cell islet (blue). This is clearly explained in the figure legend where we explicitly mention TNC expression in the tumor matrix tracks, TMT, inside the tumor and in the reticular networks in the TdLNs, respectively. This is now also documented in immunofluorescence images that cover the whole area of a tumor and a TdLN, respectively and show that FRCs express TNC in both compartments. The original data are provided in

Appendix Fig 1Q and **Appendix Fig 5C**. In **Fig. 7C** we focus on the crosstalk within the different compartments whereas in **Fig. 7D** the focus is on the molecular events inside the tumor following irradiation and how MP5 impacts some of these effects. We believe that the more expanded legend now fully explains the results.

5. The figure 1 title 'Expression of TNC impacts tumor radiosensitivity, and immune cell infiltration in the TdLNs of a murine model of OSCC' is somewhat misleading. The data in Figure 1 shows mostly infiltration in the tumor, not in TdLNs. Please consider revising the title.

Author's reply: With the newly provided data of more T and B cells in the TdLNs (NEW Figure 1 J, K) we consider the title valid.

2nd Mar 2026

Dear Dr. Orend, Dear Gertraud,

Thank you for submitting your revised files, and for your patience during the editorial checks. We have now received the report from referee #3, who is satisfied with the revisions (but has a small suggestion regarding figure 7).

I am pleased to inform you that your manuscript is thus accepted for publication and is now being sent to our publisher to be included in the next available issue of EMBO Molecular Medicine.

Please check the suggestion from referee #3, and send me your revised figure and synopsis via email (if applicable), I'll upload them in the system before export to our publisher. Please note that in agreement with our policy, I have removed the following: "All other data are available in the main text or the supplementary information. The engineered cells are available upon request."

You may qualify for financial assistance for your publication charges - either via a Springer Nature fully open access agreement or an EMBO initiative. Check your eligibility: <https://link.springer.com/journal/44321/how-to-publish-with-us>

With kind regards,

Lise

Referee #3 (Remarks for Author):

Authors have addressed my remaining concerns in the revised manuscript.

One small suggestion: the blue tumor cell with a blue arrow is now pointing to the green box depicting features of tumor stroma and vice versa in Figure 7C. If this is not intentional, consider reversing them. :)

>>> Please note that it is EMBO Molecular Medicine policy for the transcript of the editorial process (containing referee reports and your response letter) to be published as an online supplement to each paper. If you do NOT want this, you will need to inform the Editorial Office via email immediately. More information is available here: <https://link.springer.com/partners/embo-press/editorial-policies#Peer%20review>